# ON LEARNING LINEAR DYNAMICAL SYSTEMS IN CONTEXT WITH ATTENTION LAYERS

**Maria-Luiza Vladarean**[* † ‡]
maria.vladarean@tum.de

**Xuhui Zhang**[* † ‡]
xuhui.zhang@tum.de

**Suvrit Sra** [† ‡]
s.sra@tum.de

## ABSTRACT

This paper studies the expressive power of linear attention layers for in-context learning (ICL) of linear dynamical systems (LDS). We consider training on sequences of inexact observations produced by noise-corrupted LDSs, with all perturbations being Gaussian. Importantly, this non-i.i.d. data setting is a significant step towards modeling real-world scenarios. We provide the optimal weight construction for a single linear-attention layer and show its equivalence to one step of Gradient Descent relative to an autoregression objective of window size one. Guided by experiments, we uncover a connection to a generalization of the Preconditioned Conjugate Gradient method for larger window sizes. We back our findings with numerical evidence. These results add to the existing understanding of transformers' expressivity as in-context learners and offer plausible hypotheses for recent observations that place their performance on par with that of the Kalman Filter — the optimal model-dependent learner for this setting.

## 1 INTRODUCTION

This paper contributes towards understanding transformers' expressive power when learning from *non-i.i.d.* data. In particular, we consider sequences produced by a time-invariant linear dynamical system (LDS) doubly-corrupted by Gaussian noise

$$\begin{cases} \boldsymbol{x}_{t+1} = \boldsymbol{A}\boldsymbol{x}_t + \boldsymbol{w}_t, \\ y_t = \boldsymbol{c}^\top \boldsymbol{x}_t + v_t, \end{cases} \tag{1}$$

where $\boldsymbol{w}_t \overset{\text{i.i.d.}}{\sim} \mathcal{N}(\boldsymbol{0}, \Sigma_{\boldsymbol{w}})$ and $v_t \overset{\text{i.i.d.}}{\sim} \mathcal{N}(\boldsymbol{0}, \sigma_v^2)$ with mutually independent $\boldsymbol{w}_t$ and $v_t$. Our starting point is the well-known ability of transformer-based Large Language Models to perform in-context learning (ICL) (Brown et al., 2020).

ICL boils down to accurately answering a query based on a set of examples given as a textual prefix ("in context") (Brown et al., 2020). This behaviour is desirable, as it dampens the requirement for expensive data collection and fine-tuning stages (Liu et al., 2023). Present research on ICL spans the spectrum of directions between enhancing it through specialized training and prompt engineering, and building a mechanistic understanding of it — see the comprehensive review of Dong et al. (2022). Our work is aligned with the latter direction.

Broadly, there are two theoretical perspectives on ICL mechanics: a Bayesian view in which transformers recover latent concepts from prompts, thereby performing implicit Bayesian inference (Wang et al., 2023; Jiang, 2023; Wies et al., 2023; Xie et al., 2021, and subsequent works in this direction), and an algorithmic view in which they implement iterative optimization methods in context (Von Oswald et al., 2023a; Giannou et al., 2023; Akyürek et al., 2022; Garg et al., 2022; Ahn et al., 2023; Mahankali et al., 2023; Sander & Peyré, 2024; Von Oswald et al., 2023b; Sander et al., 2024). While the latter view does not account for the "emergent" aspect of "in-the-wild" ICL (Shen et al., 2023), it provides concrete expressions for transformers' modelling power and identifies the minimal functional unit wielding it — a single, causally-masked, linear attention layer, without positional encoding.

---

[*]Equal contribution.
[†]School of Computation, Information and Technology, Technical University of Munich
[‡]Munich Center for Machine Learning (MCML)

Works adopting the algorithmic view investigate transformers' ability to perform linear regression in the forward pass, confirming this through both empirical evidence and formal analysis. The i.i.d. setting is well understood, with results showing that for specific tokenizations, data distributions, and architectural choices, the optimal transformer weights implement gradient-based optimization relative to a context-dependent least-squares loss (Von Oswald et al., 2023a; Mahankali et al., 2023; Ahn et al., 2023; Von Oswald et al., 2023b; Sander et al., 2024). In contrast to the i.i.d. case, our theoretical grasp of the more realistic non-i.i.d. setting is weak. The main hurdle in analyzing this scenario is handling tokens' statistical dependence on their preceding context. Our work takes the first steps towards unraveling this difficulty.

Specifically, we study the ability of a single linear attention layer to predict observation $y_T$ based on a context of past observations $\{y_t\}_{t=1}^{T-1}$ generated by LDS (1). This setting is well-motivated: firstly, sequence $\{y_t\}_{t=1}^{T}$ is built on a temporal scaffold similar to that of language-induced tokens, which stands in stark contrast to the i.i.d. setup predominantly addressed by prior works (with few exceptions discussed shortly). Notably, dynamical systems have already been proposed as conveniently flexible models for grammatical sentence formation (Elman, 1995; Tabor et al., 1996; Beim Graben et al., 2004; Belanger & Kakade, 2015), thus making setting (1) particularly relevant. Secondly, considering LDS-produced data places the algorithmic and Bayesian views on closer footing, since LDSs are a subclass of the Hidden Markov Models (Minka, 1999) used in the Bayesian formulation of ICL (Xie et al., 2021). Finally, empirical observations highlight the strong performance of transformers relative to the Kalman Filter (KF) (Kalman, 1960) for predicting $y_T$, even in regimes where KF is provably optimal (Du et al., 2023). To our knowledge, the underlying mechanism is yet to be understood.

Our goal is to characterize the optimal single linear self-attention layer trained to predict $y_T$ based on the context $\{y_t\}_{t=1}^{T-1}$. We begin by defining a context-dependent loss for time-series data via the improper learning approach to system identification, whereby processes of type (1) are well approximated by autoregressive models. We then identify and interpret the structure of optimally trained linear attention layers as algorithmic steps on the corresponding autoregressive loss. Our contributions are the following.

**C1.** In Theorem 4.1, we prove that for an order-one autoregressive approximation of (1), the optimal linear attention layer implements a step of Gradient Descent on the associated least-squares loss. To our knowledge, this is the first optimality result for LDS data.

**C2.** In Lemma 4.1, we identify a salient banded pattern of the matrices involved in the stationarity condition for generic order-$s$ approximations of (1). We further define a class of parameters that satisfy this structural constraint and empirically observe that minimizers obey it, thereby narrowing the search for the provably optimal linear attention layer when $s \geq 2$.

**C3.** In Section 5, we provide numerical experiments verifying our theory for order-one autoregressive approximations. Furthermore, we connect the tiling pattern of empirically determined minimizers for order-$s$ approximations, $s \geq 2$, with a generalization of the Preconditioned Conjugate Gradient method, thus further highlighting the view of ICL as on-the-fly optimization. To our knowledge, this is the first interpretation of the in-context algorithm for general order-$s$ autoregression.

**C4.** Conceptually, we provide theoretical grounding for interpreting ICL as implicit optimization in the LDS setting, bridging system identification theory and empirical observations of transformer-KF parity.

## 2 RELATED LITERATURE

We review the studies treating ICL as in-context optimization, together with works on filtering and system identification. Further comparisons are discussed in Section 4.1.

**ICL for linear regression with i.i.d data.** This line of work studies whether transformers trained on a few-shot learning objective can perform linear regression in-context, and how. Garg et al. (2022); Akyürek et al. (2022); Von Oswald et al. (2023a) provide empirical results in the affirmative, along with possible architecture constructions implementing Gradient Descent (GD) steps relative to a context-induced least squares loss. Through this lens, ICL reduces to on-the-fly optimization

executed in the transformer's forward pass. Mahankali et al. (2023); Zhang et al. (2024); Ahn et al. (2023) complement these findings by proving that the one-layer linear self-attention implementing a (preconditioned) GD step is a global minimizer of the pretraining loss when covariates are Gaussian and i.i.d. Finally, Zhang et al. (2024) complete the picture by proving that Gradient Flow converges to these global minimizers. Our results extend this line of work to the non-i.i.d. setting.

**ICL and system identification.**   Different from the prior section, the following papers use the standard causal pretraining objective (minimizing prediction error over all sequence positions) and, unless stated otherwise, the results concern a single layer of linear self-attention. Von Oswald et al. (2023b) give a construction implementing a GD step on $\mathcal{L}(\boldsymbol{W}) \;:=\; \sum_{i=1}^{t-1} \| \boldsymbol{W} \boldsymbol{y}_i - \boldsymbol{y}_{i+1} \|^2$ in parallel for all positions $t$, for a specific token augmentation. Sander et al. (2024), further characterize the global minimizers of this causal pretraining loss relative to the noiseless data $\boldsymbol{y}_{t+1} = \boldsymbol{A}\boldsymbol{y}_t$, with $\boldsymbol{A}$ sampled from the set of commuting orthogonal matrices. Notably, under the same token augmentation, this characterization coincides with the construction of Von Oswald et al. (2023b). Sander et al. (2024) also describe minimizers for architectures using positional encoding, albeit under the restriction of diagonal weight matrices. Zheng et al. (2024) complement these results by showing that, for a diagonal weight initialization and a controlled distribution of $\boldsymbol{y}_0$, Gradient Flow (GF) recovers the aforementioned GD-implementing optimum. Finally, Sander & Peyré (2024) extend these results to arbitrary orthogonal $\boldsymbol{A}$s via an infinite-depth attention-only transformer that correctly predicts $\boldsymbol{y}_T$ in the limit $T \to \infty$. This result also extends to the standard softmax activation.

Moving away from the noiseless settings above, Cole et al. (2025) establish approximation theoretic results for deep attention-only transformers predicting the sequence $\boldsymbol{y}_{t+1} = \boldsymbol{A}\boldsymbol{y}_t + \boldsymbol{w}_t$, with $\boldsymbol{w}_t \sim \mathcal{N}(\boldsymbol{0}, \sigma_{\boldsymbol{w}}^2 \boldsymbol{I})$ and $\boldsymbol{A} \in \mathbb{S}_{++}^d$. They establish the existence of a $\log(T)$-depth transformer that achieves uniform-over-$\boldsymbol{A}$ error of order $\frac{\log(T)}{T}$ for predicting $\mathbb{E}[\boldsymbol{x}_{T+1} \mid \boldsymbol{x}_t, \boldsymbol{A}]$, and derive a lower bound on the prediction accuracy attainable by a single linear attention layer. Regarding transformers' capacity, Ziemann et al. (2024) establish that achieving a uniform-in-time error bound for next-step prediction requires a number of parameters at least quadratic in the algebraic multiplicities of $\boldsymbol{A}$'s unstable eigenvalues, as well as a context length at least logarithmic in $T$.

In summary, these works either study transformers' ICL performance relative to simplified LDSs or do not address the question of weight optimality. In contrast, we study fully-fledged systems (1) with the aim of characterizing the pretraining loss minimizers in the few-shot training setting.

**Transformers and linear filtering.**   The classical model-based prediction tool for systems of type (1) is the Kalman Filter (KF) (Kalman, 1960). Using knowledge of system parameters, the KF gives the minimum expected squared error estimates $\hat{\boldsymbol{x}}_i$ of the hidden states $\boldsymbol{x}_i$ as linear combinations of the past observations $y_i$. Transformers as potential implementers of KF were studied by Goel & Bartlett (2024), who prove that a softmax causal attention layer is an arbitrarily good approximator for a given type (1) system. Akram & Vikalo (2024) further construct a transformer emulating the KF. Finally, Du et al. (2023) provide empirical evidence that a GPT-2 architecture (Radford et al., 2019) competes in accuracy with the KF for predicting the next observation in-context, though the mechanism remains unstudied. We partially fill this gap with our present work.

## 3   PROBLEM FORMULATION & ASSUMPTIONS

**Notation.**   Vectors and matrices are denoted by bold, lowercase and uppercase letters, respectively, with lowercase letters reserved for scalars. Symbols $\boldsymbol{1}_d$ and $\boldsymbol{0}_d$ are the all-ones and all-zeros vectors of dimension $d$, and $\boldsymbol{1}_{d \times m}$ and $\boldsymbol{0}_{d \times m}$ are the analogous matrices. Subscripts of the form $\boldsymbol{A}_{i:j,q:k}$ identify the submatrix formed by rows from $i$ to $j$ and columns from $p$ to $q$. Unless stated otherwise, $\| \cdot \|$ denotes the Euclidean norm of vectors and the spectral norm of matrices. We denote by $\mathrm{Tr}\,(\cdot)$ the trace of a matrix, $\langle \cdot, \cdot \rangle$ the inner product, by $\| \cdot \|_F$ its Frobenius norm, and by $\rho(\cdot)$ its spectral radius. We use $\boldsymbol{e}_i$ for the $i^{\text{th}}$ canonical basis vector and $\boldsymbol{I}$ for the identity matrix. The notation $\mathbb{S}_{+(+)}^d$ defines the cone of symmetric positive-semidefinite (-definite) matrices in $\mathbb{R}^{d \times d}$, respectively. We use $\mathbb{S}^{d-1}$ to denote the unit sphere in $\mathbb{R}^d$. We use $\odot$ to denote the Hadamard product. Finally, we use $[n]$ when referencing the set of integers $\{1, 2, \ldots n\}$. We write w.p. to mean "with probability".

**The big picture: filtering, system identification, and linear regression.** The KF (Kalman, 1960) computes the optimal estimates $\hat{\boldsymbol{x}}_i$ of $\boldsymbol{x}_i$ through the system of recursions

$$
\begin{cases}
\textbf{Predict: } \hat{\boldsymbol{x}}_{t+1|t} = \boldsymbol{A}\hat{\boldsymbol{x}}_t \\
\qquad\qquad \boldsymbol{P}_{t+1|t} = \boldsymbol{A}\boldsymbol{P}_t\boldsymbol{A}^\top + \Sigma_{\boldsymbol{w}} \\[4pt]
\textbf{Gain: } \quad \boldsymbol{k}_{t+1} = \boldsymbol{P}_{t+1|t}\boldsymbol{c}\left(\boldsymbol{c}^\top\boldsymbol{P}_{t+1|t}\boldsymbol{c} + \sigma_v\right)^{-1} \\[4pt]
\textbf{Update: } \hat{\boldsymbol{x}}_{t+1} = \hat{\boldsymbol{x}}_{t+1|t} + \boldsymbol{k}_{t+1}(y_{t+1} - \boldsymbol{c}^\top\hat{\boldsymbol{x}}_{t+1|t}) \\
\qquad\qquad \boldsymbol{P}_{t+1} = (\boldsymbol{I}_d - \boldsymbol{k}_{t+1}\boldsymbol{c}^\top)\boldsymbol{P}_{t+1|t},
\end{cases}
\tag{2}
$$

where $\hat{\boldsymbol{x}}_0$ and the error covariance estimate $\boldsymbol{P}_0$ are inputs. Under the Gaussian errors assumption, the state prediction satisfies $\hat{\boldsymbol{x}}_t = \mathbb{E}[\boldsymbol{x}_t \,|\, y_t, \dots y_1]$ and, consequently, the forward observation prediction follows $\hat{y}_{t+1} := \boldsymbol{c}^\top\boldsymbol{A}\hat{\boldsymbol{x}}_t = \mathbb{E}[y_{t+1} \,|\, y_t, \dots y_1]$. The fast, constant-time KF predictions, however, require knowing the LDS parameters — a condition generally not satisfied in practice.

Consequently, "proper learning" approaches seek to reconstruct the underlying model by first estimating $\boldsymbol{A}, \boldsymbol{c}, \Sigma_w, \sigma_v$ through costly parameter identification techniques, and then producing forward observation predictions using the KF (Hamilton, 1995). In contrast, "improper learning" methods eschew structural constraints and solely seek to achieve low error relative to the underlying data distribution and the learning objective (Kozdoba et al., 2019, and references therein). For LDSs, this boils down to expressing the next observation as a linear function of the recent past. Not only does the latter approach avoid parameter estimation, but it also benefits from convex formulations, thus being amenable to classical optimization techniques. Most importantly, for certain LDS classes, improper learning methods can closely track $\mathbb{E}[y_{t+1} \,|\, y_t, \dots y_1]$, as we describe next.

The data-generating process (1) can be rewritten via the KF quantities in expression (2). In particular, Tsiamis & Pappas (2019) express future observations as a function of their past $s$ predecessors,

$$
\begin{aligned}
[y_{s+1}, \dots y_{T-1}] = \boldsymbol{c}^\top[(\boldsymbol{A} - \boldsymbol{k}\boldsymbol{c}^\top)^{s-1}\boldsymbol{k}, \dots \boldsymbol{k}] \, [\bar{\boldsymbol{y}}_1, \dots \bar{\boldsymbol{y}}_{T-s-1}] \\
+ \; \boldsymbol{c}^\top(\boldsymbol{A} - \boldsymbol{k}\boldsymbol{c}^\top)^s[\hat{\boldsymbol{x}}_1, \dots \hat{\boldsymbol{x}}_{T-s+1}] \; + \; [\varepsilon_{s+1}, \dots \varepsilon_{T-1}],
\end{aligned}
\tag{3}
$$

where $\bar{\boldsymbol{y}}_t := [y_t, y_{t+1}, \dots y_{t+s-1}]^\top$, $\boldsymbol{k}$ is the steady-state gain, and $e_i \in \mathbb{R}$ are i.i.d, zero-mean Gaussian errors. Under KF convergence conditions, quantity $\rho(\boldsymbol{A} - \boldsymbol{k}\boldsymbol{c}^\top) < 1$ makes the second term vanish exponentially in $s$ and thus renders it negligible. We are now in the familiar setting of noisy linear regression, albeit with non-i.i.d. data. The resulting order-$s$ autoregressive process (AR($s$)) is associated with the optimization objective

$$
\min_{\boldsymbol{w} \in \mathbb{R}^s} \mathcal{L}_{AR(s)}(\boldsymbol{w}) := \frac{1}{2(T-s-1)} \sum_{t=1}^{T-s-1} (y_{t+s} - \boldsymbol{w}^\top\bar{\boldsymbol{y}}_t)^2.
\tag{4}
$$

This simplification is the crux of improper learning approaches to system identification (Kozdoba et al., 2019) and becomes relevant in conjunction with the view of transformers as optimizers of a context-induced least-squares objective. Should this view withstand scrutiny in the non-i.i.d. setting, it would suggest that transformers can learn LDS-based time series in context to arbitrary accuracy as a function of $s$. This is our incentive for characterizing the few-shot pretraining loss minimizers.

To ensure the validity of the autoregressive process approximation, we make the following assumption.

**Assumption 3.1** (System assumptions). *LDS* (1) *has strictly positive definite noise covariances* $\Sigma_{\boldsymbol{w}}$ *and* $\sigma_v > 0$. *The transition matrix* $\boldsymbol{A} \in \mathbb{R}^{d \times d}$ *is marginally stable, with* $\rho(\boldsymbol{A}) \leq 1$, *and the pair* $(\boldsymbol{A}, \boldsymbol{c})$ *is observable, meaning that*

$$
\boldsymbol{O} = \left[\boldsymbol{c}, \, \boldsymbol{A}^\top\boldsymbol{c}, \, \dots, \, (\boldsymbol{A}^{d-1})^\top\boldsymbol{c}\right]^\top
\tag{5}
$$

*has a column rank of* $d$.

Assumption 3.1 is standard in the literature, and ensures KF convergence (Harrison, 1997) along with the exponential vanishing of the bias term in (3). Furthermore, it ensures the closeness of forward observation predictions given by the KF with those produced by a linear autoregressive predictor determined by expression (4) (Kozdoba et al., 2019).

**Transformer architecture.** Transformers (Vaswani et al., 2017) are neural architectures performing sequence-to-sequence mapping. For input tokens $\boldsymbol{S}_N = [\boldsymbol{s}_1, \ldots \boldsymbol{s}_N]^\top \in \mathbb{R}^{N \times p}$, the transformer produces a corresponding $\hat{\boldsymbol{S}}_N = [\hat{\boldsymbol{s}}_1, \ldots \hat{\boldsymbol{s}}_N]^\top \in \mathbb{R}^{N \times p}$ by dynamically mixing tokens via its attention mechanism. An $L$-layer transformer $\mathcal{T}_\theta : \mathbb{R}^{N \times p} \to \mathbb{R}^{N \times p}$ parametrized by $\theta = [\theta_i]_{i=1}^L$ is a composition of blocks $\mathcal{T}_L = \mathcal{T}_{\theta_1} \circ \ldots \mathcal{T}_{\theta_L}$. Each $\mathcal{T}_{\theta_i}$ is a sequence-to-sequence function given by

$$\mathcal{T}_{\theta_i}(\boldsymbol{S}) := (\mathrm{MLP}_{\theta_i^{\mathrm{MLP}}} \circ \mathcal{A}_{\theta_i^{\mathrm{att}}})(\boldsymbol{S}),$$

where $\mathrm{MLP}_{\theta_i^{\mathrm{MLP}}}$ is a multilayer perceptron and $\mathcal{A}_{\theta_i^{\mathrm{att}}}$ is the attention mapping. This paper studies the simplified block $\mathcal{T}_\theta(\boldsymbol{S}) := \mathcal{A}_\theta(\boldsymbol{S})$, with $L = 1$ and $\mathrm{MLP}_{\theta_1^{\mathrm{MLP}}} = \mathrm{Id}$.

The causal $h$-headed attention block with residual connections is given by

$$\mathcal{A}_\theta(\boldsymbol{S}) := \boldsymbol{S} + \sum_{h=1}^H \sigma\left(\boldsymbol{M} \odot \frac{1}{\tau}\boldsymbol{S}\boldsymbol{W}_Q^h(\boldsymbol{W}_K^h)^\top \boldsymbol{S}^\top\right)\boldsymbol{S}\boldsymbol{W}_V^h\boldsymbol{W}_O^h, \tag{6}$$

where the parameters $\theta = [\boldsymbol{W}_Q^h, \boldsymbol{W}_K^h, \boldsymbol{W}_V^h, \boldsymbol{W}_O^h]_{h=1}^H$ are the query, key, value, and projection matrices, respectively; $\tau > 0$ is a scaling constant; $\sigma$ is the softmax normalizing function applied row-wise; and $\boldsymbol{M} \in \mathbb{R}^{N \times N}$, with $\boldsymbol{M}_{i,j} = 1$ if $i \geq j$ and $-\infty$ otherwise, is a causal mask.

Similar to prior works (Von Oswald et al., 2023a; Ahn et al., 2023; Mahankali et al., 2023), we restrict our study to the analytically tractable setting of single-headed linear attention (Katharopoulos et al., 2020). Without loss of expressivity, we drop the projection matrix $\boldsymbol{W}_O$ and consider the $\boldsymbol{W}_Q\boldsymbol{W}_K^\top$ as a single matrix $\boldsymbol{W}_{QK} \in \mathbb{R}^{p \times p}$. Since we're working in the few-shot scenario, we're concerned solely with predicting the final position as

$$\hat{\boldsymbol{s}}_N := \mathcal{T}_\theta(\boldsymbol{S})_N = \boldsymbol{s}_N + \frac{1}{N-1}\boldsymbol{W}_V^\top \sum_{i=1}^{N-1} \boldsymbol{s}_i\boldsymbol{s}_i^\top \boldsymbol{W}_{QK}^\top \boldsymbol{s}_N, \tag{7}$$

where we set $\tau = N - 1$ and omit the last sum element due to a token asymmetry discussed next.

**Token construction.** We use the token construction approach of Von Oswald et al. (2023a); Ahn et al. (2023); Mahankali et al. (2023). The input matrix $\boldsymbol{Y}_0$ built for AR($s$) data (4) is

$$\boldsymbol{Y}_0 := \begin{bmatrix} \bar{\boldsymbol{y}}_1 & \bar{\boldsymbol{y}}_2 & \cdots & \bar{\boldsymbol{y}}_{T-s-1} & \bar{\boldsymbol{y}}_{T-s} \\ y_{s+1} & y_{s+2} & \cdots & y_{T-1} & 0 \end{bmatrix} \tag{8}$$

where $s \geq 1$ is the window size of the AR process and $Y_0 \in \mathbb{R}^{(s+1) \times (T-s)}$. The corresponding scaling constant $\tau$ in (6) becomes $T - s - 1$. The last column represents the "test" token, whose last entry is filled in the transformer's forward pass by the estimate $\hat{y}_T$. This asymmetry motivates the last term's removal in (7).

Lemma 3.1 below ensures the existence (by construction) of a linear attention layer producing $\boldsymbol{Y}_0$ from the raw sequence $\{y_t\}_{t=1}^T$. Its proof is deferred to Appendix C.1 due to space constraints.

**Lemma 3.1.** *For a given $s >= 1$, there exists an $s + 1$-headed linear attention layer with positional encoding which transforms input sequences $[y_1, y_2, \ldots, y_T]^\top$ into*

$$\begin{bmatrix} \bar{\boldsymbol{y}}_1 & \cdots & \bar{\boldsymbol{y}}_{T-s} & \boldsymbol{0}_{(s-1) \times (T-s-1)} \\ y_{s+1} & \cdots & 0 & \boldsymbol{0}_{T-s-1}^\top \end{bmatrix}^\top.$$

*The latter quantity is essentially equivalent to $\boldsymbol{Y}_0$ as defined in equation* (8).

**Data distribution, loss function, and training paradigm.** We consider trajectories $\{y_i\}_{i=1}^T$ sampled from system (1), where each trajectory corresponds to different parameters $\boldsymbol{A}$, $\boldsymbol{c}$ and $\boldsymbol{x}_0$. We assume $\boldsymbol{x}_0 \sim \mathcal{N}(\boldsymbol{0}_d, \Sigma_{\boldsymbol{x}_0})$, and impose the following condition on the distributions of $\boldsymbol{A}$ and $\boldsymbol{c}$.

**Assumption 3.2** (LDS family). *The system matrix $\boldsymbol{A} \in \mathbb{R}^{d \times d}$ is sampled from a centrally symmetric distribution supported on $\{\boldsymbol{M} \in \mathbb{R}^{d \times d} \mid \rho(\boldsymbol{M}) \leq 1\}$, for which it holds that*

$$\mathbb{P}\left(\{\boldsymbol{A} \mid \exists i, j \in [d], \text{ s.t. } \lambda_i(\boldsymbol{A}) = \lambda_j(\boldsymbol{A})\}\right) = 0. \tag{9}$$

*In other words, $\boldsymbol{A}$ has a simple spectrum almost surely. The observation vector $\boldsymbol{c} \in \mathbb{R}^d$ is sampled independently, from a distribution that is absolutely continuous w.r.t. the Lebesgue measure over $\mathbb{R}^d$.*

Except for the central symmetry condition, Assumption 3.2 simply ensures that Assumption 3.1 holds w.p. 1 for every sampled LDS (proven in Appendix C.2). The central symmetry of $\boldsymbol{A}$'s distribution, on the other hand, is a technical requirement for proving our main result.

Data generation proceeds in two steps: we sample $\boldsymbol{A}$, $\boldsymbol{c}$, and $\boldsymbol{x}_0$ independently and observe the evolution of system (1) for $T$ steps. Note the noises $\boldsymbol{w}_t$ and $v_t$ are independent of $\boldsymbol{A}$, $\boldsymbol{c}$, and $\boldsymbol{x}_0$. We then construct $\boldsymbol{Y}_0$ (8) for a fixed $s$, and train our model to minimize the in-context loss

$$\mathcal{L}(\theta) = \mathbb{E}\left[ \frac{1}{2} \left( \mathcal{T}_\theta(\boldsymbol{Y}_0)_{s+1, T-s} - y_T \right)^2 \right], \tag{10}$$

where the expectation here and throughout is taken over $\boldsymbol{A}, \boldsymbol{c}, \boldsymbol{x}_0, \{\boldsymbol{w}_t\}, \{v_t\}$. The subscript $(s+1, T-s)$ isolates the rightmost element of the final output token.

## 4 OPTIMAL PARAMETER CONFIGURATIONS

This section presents our formal analysis and its implications relative to prior literature. Our theoretical contribution is two-fold. First, in Lemma 4.1 we reveal a salient structure within the first-order optimality condition, which plays an important role in finding optimum configurations relative to the in-context loss (10) for any $s \geq 1$. Second, in Theorem 4.1 we prove that the weight configuration implementing one step of GD is a global minimizer of loss (10) for AR(1)-type tokens.

Unlike the i.i.d. case, each token generated by the LDS depends on the entire history. This causes high-order data moments to appear in the in-context loss, which can only be handled by unrolling the process to its initial state $\boldsymbol{x}_0$. A general approach to dealing with this statistical dependence is given in Appendix D.3. We now describe the structure emerging within the first-order optimality condition.

Following Ahn et al. (2023), we use basic algebra (Appendix D.3) to rewrite loss (10) as

$$\mathcal{L}(\boldsymbol{b}, \boldsymbol{z}_1, \ldots \boldsymbol{z}_s) := \mathbb{E}\left[ \left( \frac{1}{T-s-1} \sum_{k=1}^{s} \langle \bar{\boldsymbol{Y}}, \boldsymbol{b}\boldsymbol{z}_k^\top \rangle y_{T-s-1+k} - y_T \right)^2 \right], \tag{11}$$

where $\boldsymbol{W}_V^\top = [\boldsymbol{0}_{(s+1)\times s}, \boldsymbol{b}]^\top$, $\boldsymbol{W}_{QK}^\top = [\boldsymbol{z}_1, \ldots \boldsymbol{z}_s, \boldsymbol{0}_{s+1}]$ and $\bar{\boldsymbol{Y}} := \boldsymbol{Y}_{0:, 1:T-s-1} \boldsymbol{Y}_{0:, 1:T-s-1}^\top$. The zero-padding of both matrices comes from predicting solely the last position of the final token. Consequently, parameters satisfying for all $j \in [s]$ that

$$\mathbb{E}\left[ \sum_{k=1}^{s} \langle \bar{\boldsymbol{Y}}, \boldsymbol{b}\boldsymbol{z}_k^\top \rangle y_{T-s-1+k} y_{T-s-1+j} \bar{\boldsymbol{Y}} \right] = (T-s-1)\mathbb{E}\left[ y_T y_{T-s-1+j} \bar{\boldsymbol{Y}} \right], \tag{12}$$

are critical points of the loss. Notably, the RHS of (12) has a banded structure induced by the distributional symmetry of $\boldsymbol{A}$, whose pattern alternates depending on the parity of $s+j$. In particular, RHS (12) $= \boldsymbol{B}_0(s,j)$ if $s+j \in 2\mathbb{Z}$, and RHS (12) $= \boldsymbol{B}_1(s,j)$ if $s+j \in 2\mathbb{Z}+1$, with

$$\boldsymbol{B}_0(s,j) := \begin{bmatrix} 0 & \star & 0 & & \\ \star & 0 & \star & \ddots & \\ 0 & \star & 0 & \ddots & 0 \\ & \ddots & \ddots & \ddots & \star \\ & & 0 & \star & 0 \end{bmatrix}, \text{ and } \boldsymbol{B}_1(s,j) := \begin{bmatrix} \star & 0 & \star & \cdots & \\ 0 & \star & 0 & \ddots & \\ \star & 0 & \star & \ddots & \star \\ \vdots & \ddots & \ddots & \ddots & 0 \\ & & \star & 0 & \star \end{bmatrix}, \tag{13}$$

where $\star$ is a placeholder for structurally unconstrained elements which can evaluate to arbitrary reals, not necessarily the same at different positions (see Appendix D.3). Note that $\boldsymbol{B}_0(s,j)$ is constrained to zero at positions $(r,l) \in [s+1] \times [s+1]$ such that $r+l \in 2\mathbb{N}$, and $\boldsymbol{B}_1(s,j)$ is constrained to zero whenever $r+l \in 2\mathbb{N}+1$. Lemma 4.1 below formalizes a class of parameters ensuring that LHS (12) respects the structural sparsity pattern of RHS (12).

**Lemma 4.1.** *For any $s \geq 1$, $j \in [s]$, the parameters $\boldsymbol{W}_{QK}$ and $\boldsymbol{W}_V$ having the structure*

$$\boldsymbol{W}_{QK} = \begin{bmatrix} \boldsymbol{R}_{1:s, 1:(s+1)} \\ \boldsymbol{0}_{s+1}^\top \end{bmatrix}, \qquad \boldsymbol{W}_V = \begin{bmatrix} \boldsymbol{0}_{(s+1)\times s} & \boldsymbol{r}_{2+s \bmod 2 : 2\lceil \frac{s+1}{2} \rceil} \end{bmatrix}, \tag{14}$$

*with* $\boldsymbol{R} := \mathbf{1}_{\lfloor \frac{s+1}{2} \rfloor \times \lceil \frac{s+1}{2} \rceil} \otimes \begin{bmatrix} \star & 0 \\ 0 & \star \end{bmatrix}$ *and* $\boldsymbol{r} := \mathbf{1}_{\lceil \frac{s+1}{2} \rceil} \otimes \begin{bmatrix} 0 \\ \star \end{bmatrix}$, *ensure that LHS* $(12)_{r,\ell} = 0$ *whenever* $r + l \in 2\mathbb{N}$ *and* $s + j \in 2\mathbb{Z}$, *or* $r + l \in 2\mathbb{N} + 1$ *and* $s + j \in 2\mathbb{Z} + 1$.

Note that these parameters recover the zero-structure of $\boldsymbol{B}_0(s, j)$ and $\boldsymbol{B}_1(s, j)$. Lemma 4.1 can be understood as a structure-based narrowing of the parameter class likely to hold minimizers of loss (10). The experiments in Section 5 give empirical support for this claim.

Our second step is to use structure (14) to identify a global minimizer of loss (10) for AR(1)-type tokens. The result is stated in Theorem 4.1, whose proof we defer to Appendix D.4.

**Theorem 4.1.** *Let* $\boldsymbol{Y}_0$ *encode the input tokens for* $s = 1$. *Then, the optimal parameters* $\theta^\star = \left( \boldsymbol{W}_{QK}^\star, \boldsymbol{W}_V^\star \right)$ *of a single linear self-attention layer with respect to loss* $\mathcal{L}(\theta)$ *are*

$$\boldsymbol{W}_{QK}^\star = \begin{bmatrix} \frac{(T-2)\mathbb{E}\left[y_{T-1}y_T \sum_{i=1}^{T-2} y_i y_{i+1}\right]}{\mathbb{E}\left[y_{T-1}^2 \left(\sum_{i=1}^{T-2} y_i y_{i+1}\right)^2\right]} & 0 \\ 0 & 0 \end{bmatrix}, \boldsymbol{W}_V^\star = \begin{bmatrix} 0 & 0 \\ 0 & 1 \end{bmatrix}, \tag{15}$$

*up to rescaling with a nonzero constant.*

Broadly, the proof of Theorem 4.1 encounters two difficulties compared to the i.i.d. case: the number of terms that need to be matched in satisfying the first-order optimality condition, and the full-history dependence of the data. We address the first obstacle by using the structural simplification of Lemma 4.1, and the second by invoking Isserlis' theorem (Isserlis, 1918), which provides a tractable decomposition of the higher-order moments of the data. Details are given in Appendix D.3.

Notably, a forward pass using the optimal parameters (15) amounts to the prediction given after one GD step on $\mathcal{L}_{AR(1)}(w)$ starting from $w_0 = 0$. We thus recover the ICL-as-optimization view upheld by works in the i.i.d. setting (Ahn et al., 2023; Mahankali et al., 2023) but for LDS-produced data.

## 4.1 DISCUSSION

To our knowledge, the only other architecture proposed for handling noisy observations $y_t$ of type (1) is given by Cole et al. (2025). Theirs is part of a proof of existence by construction and, as such, is not accompanied by confirming experimental evidence. Different from us, they propose an attention-only transformer that unrolls a *modified Richardson iteration* meant to estimate $\left( \frac{1}{T} \sum_{t=1}^T \boldsymbol{x}_{t+1} \boldsymbol{x}_t^\top \right) \left( \frac{1}{T} \sum_{t=1}^T \boldsymbol{x}_t \boldsymbol{x}_t^\top \right)^{-1}$ for a simpler LDS with direct state access. Their construction extends to the setting of objective (4) via the work of Tsiamis & Pappas (2019), who give a high probability result for the existence of $\left( \sum_{t=1}^{T-s-1} \bar{\boldsymbol{y}}_t \bar{\boldsymbol{y}}_t^\top \right)^{-1}$ under our assumptions. However, their transformer has a minimum of two layers, of which the first is fixed, therefore providing no guarantee that training will recover it. Our results take a first step towards filling this gap.

Tangentially, Akram & Vikalo (2024) construct a transformer emulating the KF, contingent on knowledge of the system parameters and an elaborate token augmentation scheme. While this architecture is capable of computing the forward KF observation $\hat{y}_T$, it relies on ideal knowledge of LDS (1) which is rarely encountered in practice.

Theorem 4.1 sets forth a plausible hypothesis for prior experiments (Du et al., 2023, Fig. 2) using a GPT-2 architecture trained autoregressively with data (1) for stable $\boldsymbol{A} \in \mathbb{S}_{++}^d$. Their results highlight the transformer's competitive performance relative to the KF for predicting the next observation of a previously unseen sequence, in-context. These experiments suggest an implicit form of system identification might be executed in the forward pass, though the mechanism remains unstudied. Through the ICL-as-optimization lens, we can interpret the high accuracy of GPT-2's in-context predictions as a possible consequence of Theorem 2 of Kozdoba et al. (2019). Importantly, the latter result implies that for an arbitrary, finite family $S$ of LDSs (1) and an $\varepsilon > 0$, there exists a window-length $s(\varepsilon)$ such that the optimal AR($s(\varepsilon)$) predictor incurs an average error that is at least as good, up to $\varepsilon$, as that of the forward observation prediction $\hat{y}_{t+1}$ of the best KF in $S$. Our results take the first step in formally exploring this hypothesis.

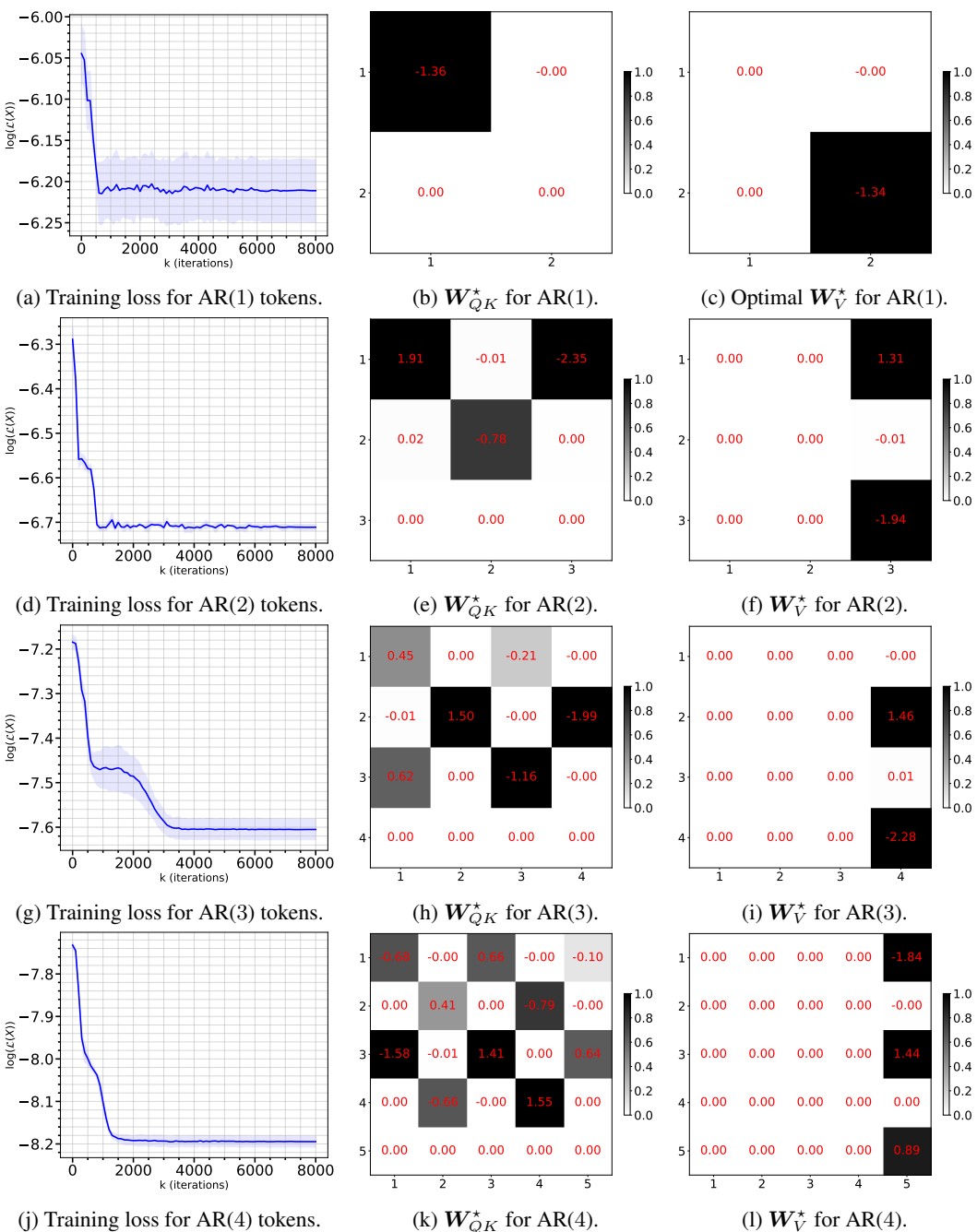

Figure 1: Experimental results for AR(1–4) tokens showing the optimally-trained attention parameters.

## 5 EXPERIMENTS

We now present numerical evidence supporting our theory. All experiments were implemented in Python 3.12 and run on a ThinkPad T14p with 32 GB RAM and a 22-core Intel Core™ Ultra 9 185H processor. The code is available at `https://github.com/XHZhang01/icl-for-lds-data`.

We train architecture (7) on sequences $\{y_t\}_{t=1}^{T}$, $T = 30$, each sampled from a different LDS of type (1) with a hidden state dimension $d = 5$. The number of training iterations is 8000 for all cases,

with an increase in the batch size for every increase in window size, starting from 3000 for AR(1). A fresh batch of LDSs is sampled at every iteration. The experiments cover the following four settings.

(a) For each sequence, we sample $A$'s diagonal entries uniformly in the interval $[-1, 1]$ and set $c = 1_d$. The noise covariances are set to $\Sigma_w = \mathtt{1e\text{-}2}I$ and $\sigma_v^2 = \mathtt{1e\text{-}2}$. The results are depicted in Figure 1 for AR(1–4) tokens.

(b) For each sequence, we sample a vector $v \sim \mathrm{Unif}([-1, 1]^d)$ and a matrix $Q \sim \mathrm{Haar}(O(d))$ independently, and set $A = Q^\top \mathrm{diag}(v)Q$. We further sample $c \sim \mathrm{Unif}([-2, 2]^d)$ independently. The noise covariances are set to $\Sigma_w = \mathtt{1e\text{-}2}I$ and $\sigma_v^2 = \mathtt{1e\text{-}2}$. Experiments for AR(1–4)-type tokens are provided in Figure 3 in Appendix B.3.

(c) For each sequence, we sample a vector $v \sim \mathrm{Unif}([-1, 1]^d)$ and a matrix $Q \sim \mathrm{Haar}(O(d))$ independently, and set $A = Q^\top \mathrm{diag}(v)Q$. We further sample $c \sim \mathrm{Unif}([-0.5, 0.5]^d)$ independently. We fix the covariance $\Sigma_w = \mathbf{Q_w}^\top \mathrm{diag}(\mathtt{1e\text{-}2} \cdot [0.9, 0.95, 1.0, 1.05, 1.1])\mathbf{Q_w}$ for all systems, for arbitrary $\mathbf{Q_w} \sim \mathrm{Haar}(O(d))$. We fix $\sigma_v^2 = \mathtt{1e\text{-}2}$. Experiments for AR(1–3)-type tokens are provided in Figure 4 in Appendix B.4.

(d) For each sequence, we independently sample a vector $v \sim \mathrm{Unif}([-1, 1]^d)$ and a matrix $\mathbf{P} = [p_{i,j}]_{i,j=1}^d$ with $p_{i,j} \overset{\mathrm{i.i.d.}}{\sim} \mathcal{U}([-1, 1])$, and set $A = \mathbf{P}^{-1}\mathrm{diag}(v)\mathbf{P}$. We further sample $c \sim \mathrm{Unif}([-1, 1]^d)$. The noise covariances are set to $\Sigma_w = \mathtt{1e\text{-}2}I$ and $\sigma_v^2 = \mathtt{1e\text{-}2}$. Experiments for AR(1–3)- tokens are provided in Figure 5 in Appendix B.5.

All the settings above have $x_0 \sim \mathcal{N}(0, \sigma_0^2 I)$, $\sigma_0^2 = \mathtt{1e\text{-}2}$. Note that we could have used any other centrally symmetric distribution with marginals supported on $[-1, 1]$ for the sampling of the diagonal $v$, e.g., $\mathrm{Unif}(\mathbb{S}^{d-1})$ — uniform on the unit sphere; $\mathrm{Unif}(\{x \in \mathbb{R}^d : \|x\|_2 \leq 1\})$ — uniform inside the unit ball, etc. These sampling schemes obey Assumption 3.2, as proven in Appendix E.1. All results are averaged over 3 random seeds. The weights are learned using AdamW (Loshchilov & Hutter, 2017) with gradient clipping and a learning rate schedule consisting of a linear warm-up phase followed by cosine annealing (Loshchilov & Hutter, 2016). A full list of hyperparameters is provided in Appendix B.

Subplots (b,c) in Figures 1, 3, 4 and 5 show optima conforming to Theorem 4.1 for AR(1)-type tokens. Moreover, subplots (e,f,h,i,k,l) in Figures 1 and 3, subplots (e,f,h,i) in Figure 4 and subplots (e,f) in Figure 5 confirm the pattern dictated by Lemma 4.1 for general $s \geq 2$. Finally, we empirically show that in setting (a), the weights converge to the sparsity pattern predicted by Lemma 4.1 in terms of the Jaccard distance of their supports. Setup details and results are given in Appendix B.2.

**Interpreting the sparsity pattern for AR(s)** $s \geq 2$. A quick calculation of the forward pass reveals that weights trained to optimality with AR($s$) tokens for $s \geq 2$ *do not* implement standard GD in the forward pass, but an iteration reminiscent of the Preconditioned Conjugate Gradient method (PCG) (Hestenes et al., 1952; Shewchuk et al., 1994), or more generally, to Krylov subspace methods (Saad, 2003, chapters 6 and 7).

We start by observing that the forward pass factor $\frac{1}{T-s-1}\bar{Y}$ (with $\bar{Y}$ defined in (11)) has a meaningful block structure involving the gradient at zero and the Hessian of $\mathcal{L}_{AR(s)}(w)$,

$$\frac{1}{T-s-1}\bar{Y} = \begin{bmatrix} \nabla^2\mathcal{L}_{AR(s)} & \nabla\mathcal{L}_{AR(s)}(0) \\ \nabla\mathcal{L}_{AR(s)}(0)^\top & \gamma \end{bmatrix}, \tag{16}$$

where $\gamma := \frac{1}{T-s-1}\sum_{t=1}^{T-s-1} y_{t+s}^2 \in \mathbb{R}$ and $\nabla^2\mathcal{L}_{AR(s)} \in \mathbb{R}^{s \times s}$ denotes the constant Hessian. Together with the parameter structure of Lemma 4.1 and the experiments, we use expression (16) to rewrite the transformer-induced predictor in a manner that highlights its resemblance to that obtained after two PCG steps on $\mathcal{L}_{AR(s)}(w)$ starting from $w_0 = 0$. We describe the case for even $s$, with the odd case following similarly. Let $s = 2k$, $k \in \mathbb{N}$ and $N := \frac{(s+1)^2+1}{2}$. Consider weights

$$W_{QK} = \begin{bmatrix} c_1 & 0 & c_2 & & \big| & c_{k+1} \\ 0 & c_{k+2} & 0 & \ddots & \big| & 0 \\ \vdots & \vdots & \ddots & \ddots & \big| & \vdots \\ 0 & c_{N-2k} & 0 & & \big| & 0 \\ \hline & \mathbf{0}_s^\top & & & \big| & 0 \end{bmatrix} \in \mathbb{R}^{s+1 \times s+1}, \qquad W_V = \begin{bmatrix} & & & \big| & c_{N-k} \\ & & & \big| & 0 \\ & \mathbf{0}_{s \times s} & & \big| & c_{N-k+1} \\ & & & \big| & \vdots \\ & & & \big| & 0 \\ \hline 0 \ldots 0 & & \big| & c_N \end{bmatrix} \in \mathbb{R}^{s+1 \times s+1}$$

where $c_i \in \mathbb{R}, \forall i \in [N]$. Renaming the top left $s \times s$ block of $\mathbf{W_{QK}}$ as $\mathbf{P}$, the top right $s \times 1$ block as $\mathbf{p}$, and the top right $s \times 1$ block of $\mathbf{W_V}$ as $\mathbf{q}$, the transformer-induced linear predictor $\frac{1}{T-s-1}\mathbf{W_{QK}}\bar{Y}\mathbf{W}_{\mathbf{V}_{:,s+1}}$ is

$$\mathbf{P}\nabla^2 \mathcal{L}_{AR(s)}\mathbf{q} \; + \; \xi_1 \mathbf{P}\nabla \mathcal{L}_{AR(s)}(\mathbf{0}) \; + \; \xi_2 \mathbf{p}, \tag{17}$$

where $\xi_1 = c_N \in \mathbb{R}$ and $\xi_2 = \mathbf{q}^\top \nabla \mathcal{L}_{AR(s)}(\mathbf{0}) + c_N \gamma \in \mathbb{R}$.

Comparatively, the PCG (Shewchuk et al., 1994, p. 51) predictor obtained after two steps on loss $\mathcal{L}_{AR(s)}$ with preconditioner $\mathbf{P}^{-1}$ starting from $\mathbf{w}_0 = \mathbf{0}$ is (see Appendix E.2)

$$\tau_1 \boldsymbol{P}\nabla^2 \mathcal{L}_{AR(s)}\boldsymbol{P}\nabla \mathcal{L}_{AR(s)}(\mathbf{0}) \; + \; \tau_2 \boldsymbol{P}\nabla \mathcal{L}_{AR(s)}(\mathbf{0}), \tag{18}$$

where $\tau_1, \tau_2 \in \mathbb{R}$ are iteration-dependent constants. Note that we have suspended the requirement for $\boldsymbol{P}$'s symmetry here — we will address this drawback shortly.

We make a few observations on the similarities and differences between the two predictors. The second terms of (17) and (18) coincide up to scaling. The first terms of these predictors represent directions mapped by $\boldsymbol{P}\nabla^2 \mathcal{L}_{AR(s)}$, which is typical of Krylov subspace methods. Moreover, if we initialize the conjugate direction of PCG to the vector $\mathbf{q}$, the first terms also coincide up to scaling. This prompts the natural question of whether directions $\boldsymbol{P}\nabla^2 \mathcal{L}_{AR(s)}\boldsymbol{P}\nabla \mathcal{L}_{AR(s)}(\mathbf{0})$ and $\boldsymbol{P}\nabla^2 \mathcal{L}_{AR(s)}\mathbf{q}$ are aligned, or at least significantly so. As a preliminary test, we compute this alignment in terms of cosine similarity when replacing $\nabla^2 \mathcal{L}_{AR(s)}$ and $\mathcal{L}_{AR(s)}(\mathbf{0})$ with empirical estimates of their expectations. Using the weights reported for the AR(4) case in settings a) and b) (Figures 1 and 3), batches of 16000 sequences for estimating the expectations and averaging the result over five random seeds, we obtain a cosine similarity of $0.88 \pm 0.05$ for setting a) and $0.93 \pm 0.01$ for setting b). The values suggest a strong alignment, which gives weight to this line of interpretation.

In terms of predictor dissimilarities, the set of directions whose linear combination yields (17) contains the additional vector $\boldsymbol{p}$. This structure is typical of so-called augmented Krylov methods, whereby "correction directions" are added to the standard Krylov search space to compensate for ill-behaved modes of $\mathbf{P}\nabla^2 \mathcal{L}_{AR(s)}$ (Carpenter et al., 2010, and references therein). Broadly, they can be seen as PCG generalizations, and have the added benefit of accommodating asymmetric $\boldsymbol{P}$s, thus resolving the earlier drawback. We test whether $\boldsymbol{p}$ acts as a correction by evaluating the residual obtained prior to and after incorporating $\boldsymbol{p}$. We use the same setup as above, and compute the cosine of the angle between the preconditioned residual evaluated at $\boldsymbol{q}$ (which coincides with $\boldsymbol{w}_1$ in PCG with non-standard initial conjugate direction), and the direction $\boldsymbol{P}\nabla^2 \mathcal{L}_{AR(s)}\boldsymbol{p}$. Anti-alignment would suggest that $\boldsymbol{p}$ moves the predictor in a residual reduction direction. Indeed, we observe a cosine of $-0.99 \pm 7e^{-5}$ for the AR(4) case in setting a), and $-0.99 \pm 3e^{-5}$ for the AR(4) case in setting b). Taken together, these preliminary results warrant further exploration of this interpretation.

Finally, we remark that our interpretation of the AR(s) case does not contradict the plain GD step observed for AR(1), since PCG variants collapse to GD for one-dimensional covariates.

## 6 LIMITATIONS & FUTURE DIRECTIONS

We have sketched a path to understanding how attention layers learn LDSs in context by leveraging results from the improper learning literature on system identification. Our study fully characterizes the learning of order-one autoregressive LDS approximations (Theorem 4.1) and narrows the class of weights that hold minimizers for higher-order approximations (Lemma 4.1), with all findings experimentally confirmed. These contributions provide the building blocks for closing further gaps, the most pressing of which is describing the optima of loss (10) for $s \geq 2$. A meaningful intermediate result here is quantifying the approximation power of optima obtained in the stationary LDS regime, where the problem structure further simplifies due to $\bar{Y}$ being Toeplitz. Such an assumption is reasonable, since systems with $\rho(\boldsymbol{A}) < 1$ reach stationarity exponentially fast. Further valuable directions concern bridging the theory-practice gap, such as extending the analysis to typical causal pretraining objectives, and empirically probing the parallel between Krylov subspace methods and the predictor modeled by a multi-layer transformer.

ACKNOWLEDGMENTS

The authors are grateful to Xiang Cheng and Anastasios Tsiamis for helpful discussions throughout the development of this work. The authors also sincerely thank the anonymous reviewers for their constructive feedback which helped improve and clarify this manuscript. All the authors of this work were generously supported by the Alexander von Humboldt foundation.

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

## A   LLM USAGE DISCLOSURE

LLMs were used in elaborating this paper as follows:

- Finding related work.

- Computing the result of polynomial multiplications.

- Generating LaTeX tables and tikz figures.

- Transferring proofs from pen-and-paper format into LaTeX automatically using the online tool Manus `https://manus.im/`.

# B    EXPERIMENTS — FURTHER DETAILS

## B.1    HYPERPARAMETERS

Table 1: Training hyperparameters of setting (a) in Section 5

| Hyperparameter | Value |
|---|---|
| Weight initialization | Xavier normal distribution (Glorot & Bengio, 2010) with gain = `1e-6` |
| Optimizer | AdamW (Loshchilov & Hutter, 2017) with $\beta_1 = 0.9$ and $\beta_2 = 0.98$ for all AR(1–4), $\epsilon = $ `1e-9` |
| Weight decay | `5e-3` for AR(2 & 4) and `1e-2` for AR(1 & 3) |
| Learning rate (i.e., max. val.) | `2e-2` for AR(1), `3e-2` for AR(2), `3e-3` for AR(3), `4e-3` for AR(4) |
| Min. learning rate | `1e-4` for AR(1 & 2) and `1e-5` for AR(3 & 4) |
| Linear warmup | 800 iter. |
| Decay schedule | Cosine annealing (Loshchilov & Hutter, 2016) |
| Max. decay steps | 7200 iter. |
| Max. grad norm (clipping) | 300 |
| Random seeds | $\{666013, 1, 0\}$ |
| Batch size / iter. | 3000 for AR(1), 4000 for AR(2), 10000 for AR(3), 18000 for AR(4) |
| Total iter. | 8001 |

Table 2: Training hyperparameters of setting (b) in Section 5

| Hyperparameter | Value |
|---|---|
| Weight initialization | Xavier normal distribution (Glorot & Bengio, 2010) with gain $= 1e-6$ |
| Optimizer | AdamW (Loshchilov & Hutter, 2017) with $\beta_1 = 0.98$ for AR(1), 0.96 for AR(2), 0.92 for AR(3), 0.88 for AR(4), $\beta_2 = 0.99$ for AR(1), 0.98 for AR(2), 0.96 for AR(3), 0.92 for AR(4), $\epsilon = 1e-9$ |
| Weight decay | $1e-2$ for AR(1), $1e-3$ for AR(2), $1e-4$ for AR(3 & 4) |
| Learning rate (i.e., max. val.) | $1e-3$ for AR(1), $4e-3$ for AR(2), $6e-3$ for AR(3), $8e-3$ for AR(4) |
| Min. learning rate | $1e-5$ |
| Linear warmup | 800 iter. |
| Decay schedule | Cosine annealing (Loshchilov & Hutter, 2016) |
| Max. decay steps | 7200 iter. |
| Max. grad norm (clipping) | 300 |
| Random seeds | $\{666013, 1, 0\}$ |
| Batch size / iter. | 4000 for AR(1), 8000 for AR(2), 14000 for AR(3), 18000 for AR(4) |
| Total iter. | 8001 |

Table 3: Training hyperparameters of setting (c) in Section 5

| Hyperparameter | Value |
|---|---|
| Weight initialization | Xavier normal distribution (Glorot & Bengio, 2010) with gain $= 1e-6$ |
| Optimizer | AdamW (Loshchilov & Hutter, 2017) with $\beta_1 = 0.98$ for AR(1), 0.96 for AR(2), 0.92 for AR(3), $\beta_2 = 0.99$ for AR(1), 0.98 for AR(2), 0.96 for AR(3), $\epsilon = 1e-9$ |
| Weight decay | $1e-2$ for AR(1), $5e-3$ for AR(2), $1e-3$ for AR(3) |
| Learning rate (i.e., max. val.) | $2e-3$ for AR(1), $3e-3$ for AR(2), $4e-3$ for AR(3) |
| Min. learning rate | $1e-5$ |
| Linear warmup | 800 iter. |
| Decay schedule | Cosine annealing (Loshchilov & Hutter, 2016) |
| Max. decay steps | 7200 iter. |
| Max. grad norm (clipping) | 300 |
| Random seeds | $\{666013, 1, 0\}$ |
| Batch size / iter. | 4000 for AR(1), 8000 for AR(2), 16000 for AR(3) |
| Total iter. | 8001 |

Table 4: Training hyperparameters of setting (d) in Section 5

| Hyperparameter | Value |
|---|---|
| Weight initialization | Xavier normal distribution (Glorot & Bengio, 2010) with gain $= \texttt{1e-6}$ |
| Optimizer | AdamW (Loshchilov & Hutter, 2017) with $\beta_1 = 0.92$ and $\beta_2 = 0.96$ for all AR(1 - 2), $\beta_1 = 0.88$ and $\beta_2 = 0.94$ for all AR(3), $\epsilon = \texttt{1e-9}$ |
| Weight decay | $\texttt{5e-3}$ for AR(1), $\texttt{1e-3}$ for AR(2), $\texttt{1e-3}$ for AR(3) |
| Learning rate (i.e., max. val.) | $\texttt{5e-3}$ for AR(1), $\texttt{5e-4}$ for AR(2), $\texttt{1e-4}$ for AR(3) |
| Min. learning rate | $\texttt{1e-5}$ |
| Linear warmup | 800 iter. |
| Decay schedule | Cosine annealing (Loshchilov & Hutter, 2016) |
| Max. decay steps | 7200 iter. |
| Max. grad norm (clipping) | 300 |
| Random seeds | $\{666013, 1, 0\}$ |
| Batch size / iter. | 4000 for AR(1), 10000 for AR(2), 18000 for AR(3) |
| Total iter. | 8001 |

## B.2 EXPERIMENTS SHOWING CONVERGENCE TO THE CHECKERBOARD PATTERN DURING TRAINING

This set of experiments serves to illustrate that parameters $\boldsymbol{W}_{QK}$ and $\boldsymbol{W}_V$ converge to the checkerboard pattern across iterations. Since the non-zero values of these parameters are of different magnitudes and we do not have their theoretical expressions for window-sizes greater than 1, we shall only consider their non-zero support, as follows.

**Definition B.1.** *For a matrix $\boldsymbol{M} \in \mathbb{R}^{d \times m}$, its support is defined as the collection of positions corresponding to non-zero values*

$$\mathrm{supp}(\boldsymbol{M}) \;:=\; \{(i,j) \in [d] \times [m] \mid a_{i,j} \neq 0\}. \tag{19}$$

*Additionally, the support-induced mask is a binary matrix with unit entries on the support*

$$\mathrm{mask}(\boldsymbol{M}) \;:=\; \left[\mathbf{1}_{(i,j)\in\mathrm{supp}(\boldsymbol{M})}\right]_{i,j=1}^{i=d,j=m} \tag{20}$$

*where $\mathbf{1}_C = 1$ if condition $C$ is true and $0$ otherwise, is the indicator function centered at $z$.*

We rely on the Jaccard distance (Jaccard, 1901) adapted to binary matrices $\boldsymbol{A}, \boldsymbol{B}$

$$\mathrm{d}_{\mathrm{Jac}}(\boldsymbol{A}, \boldsymbol{B}) \;:=\; 1 - \frac{\sum_{i,j} a_{i,j} b_{i,j}}{\sum_{i,j} \max\{a_{i,j}, b_{i,j}\}} \tag{21}$$

to track whether the support-induced masks of our parameters during training converge to the predicted (for AR(1)) or hypothesized (for AR($s$) $s \geq 2$) sparsity patterns of Lemma 4.1. Our experiments employ a tolerance level of $\texttt{1e-1}$ when computing the masks of $\boldsymbol{W}_V$ and $\boldsymbol{W}_{QK}$, meaning that any entry below this value is considered zero. The results are depicted in Figure 2 and its subplots for varying window sizes, where $\boldsymbol{M}_{QK}^{\mathrm{true}}$ and $\boldsymbol{M}_V^{\mathrm{true}}$ represent the masks posited in Lemma 4.1 for a null tolerance level. The illustrations empirically confirm that our parameters' supports converge to the ones identified in Lemma 4.1.

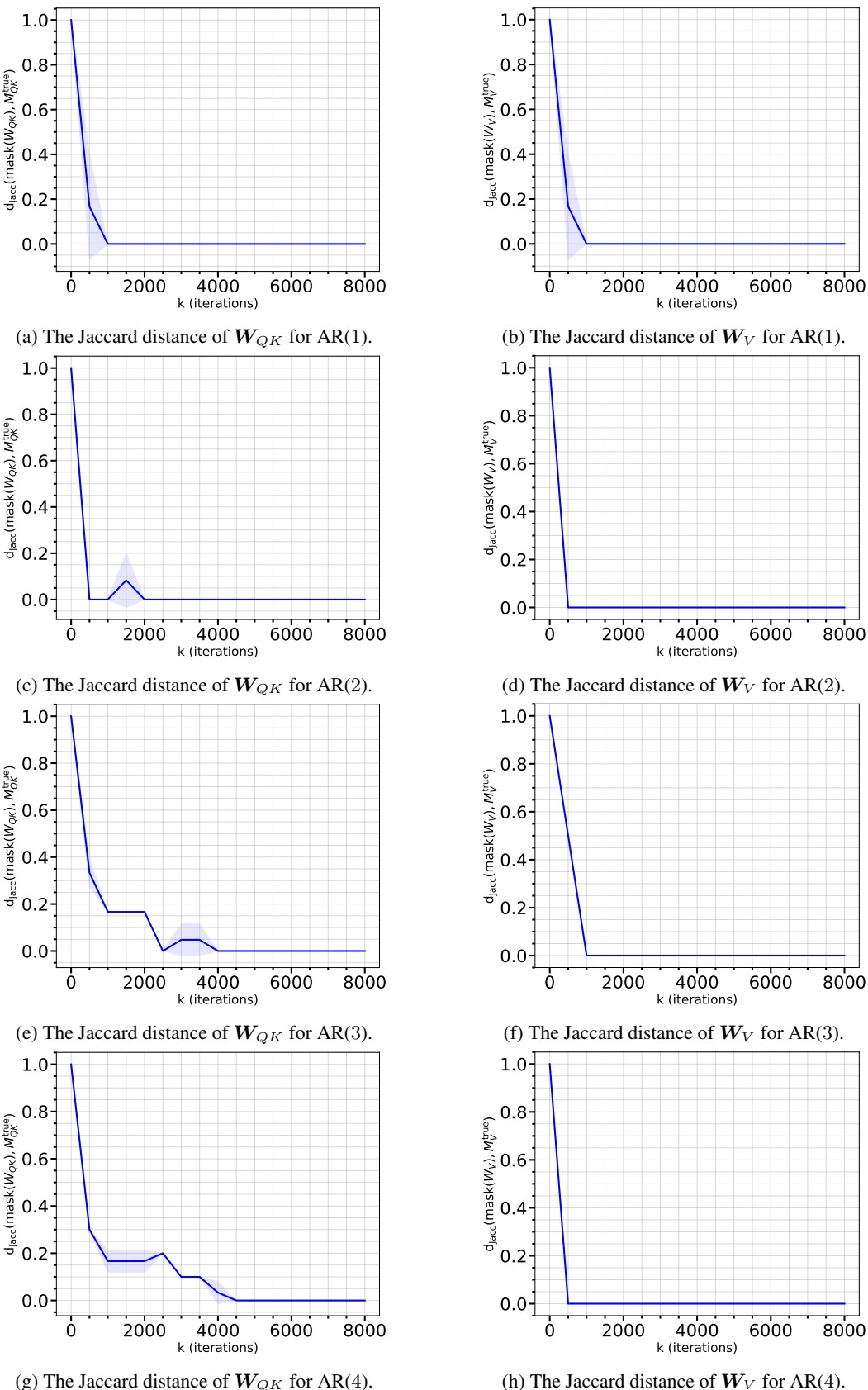

(a) The Jaccard distance of $\boldsymbol{W}_{QK}$ for AR(1).

(b) The Jaccard distance of $\boldsymbol{W}_V$ for AR(1).

(c) The Jaccard distance of $\boldsymbol{W}_{QK}$ for AR(2).

(d) The Jaccard distance of $\boldsymbol{W}_V$ for AR(2).

(e) The Jaccard distance of $\boldsymbol{W}_{QK}$ for AR(3).

(f) The Jaccard distance of $\boldsymbol{W}_V$ for AR(3).

(g) The Jaccard distance of $\boldsymbol{W}_{QK}$ for AR(4).

(h) The Jaccard distance of $\boldsymbol{W}_V$ for AR(4).

Figure 2: The experiment results of the Jaccard distance between the $\boldsymbol{M}_{QK}^{\text{true}}$ and $\boldsymbol{W}_{QK}$ and the Jaccard distance between the $\boldsymbol{M}_V^{\text{true}}$ and $\boldsymbol{W}_V$ for AR(1–4). Both converge to 0 at the end of the training.

## B.3 Experiments with non-diagonal, symmetric $\boldsymbol{A}$, random $\boldsymbol{c}$ and isotropic $\Sigma_w$

The LDS which generates the training data is as follows. For each sequence, sample $\boldsymbol{v} \sim$ $\text{Unif}([-1, 1]^d)$, sample $\boldsymbol{Q} \sim \text{Haar}(O(d))$ and set $\boldsymbol{A} = \boldsymbol{Q}^\top \text{diag}(\boldsymbol{v})\boldsymbol{Q}$. Sample $\boldsymbol{c} \sim \text{Unif}([-2, 2]^d)$. The noise covariances are set to $\Sigma_w = \texttt{1e-2}\boldsymbol{I}$ and $\sigma_v^2 = \texttt{1e-2}$.

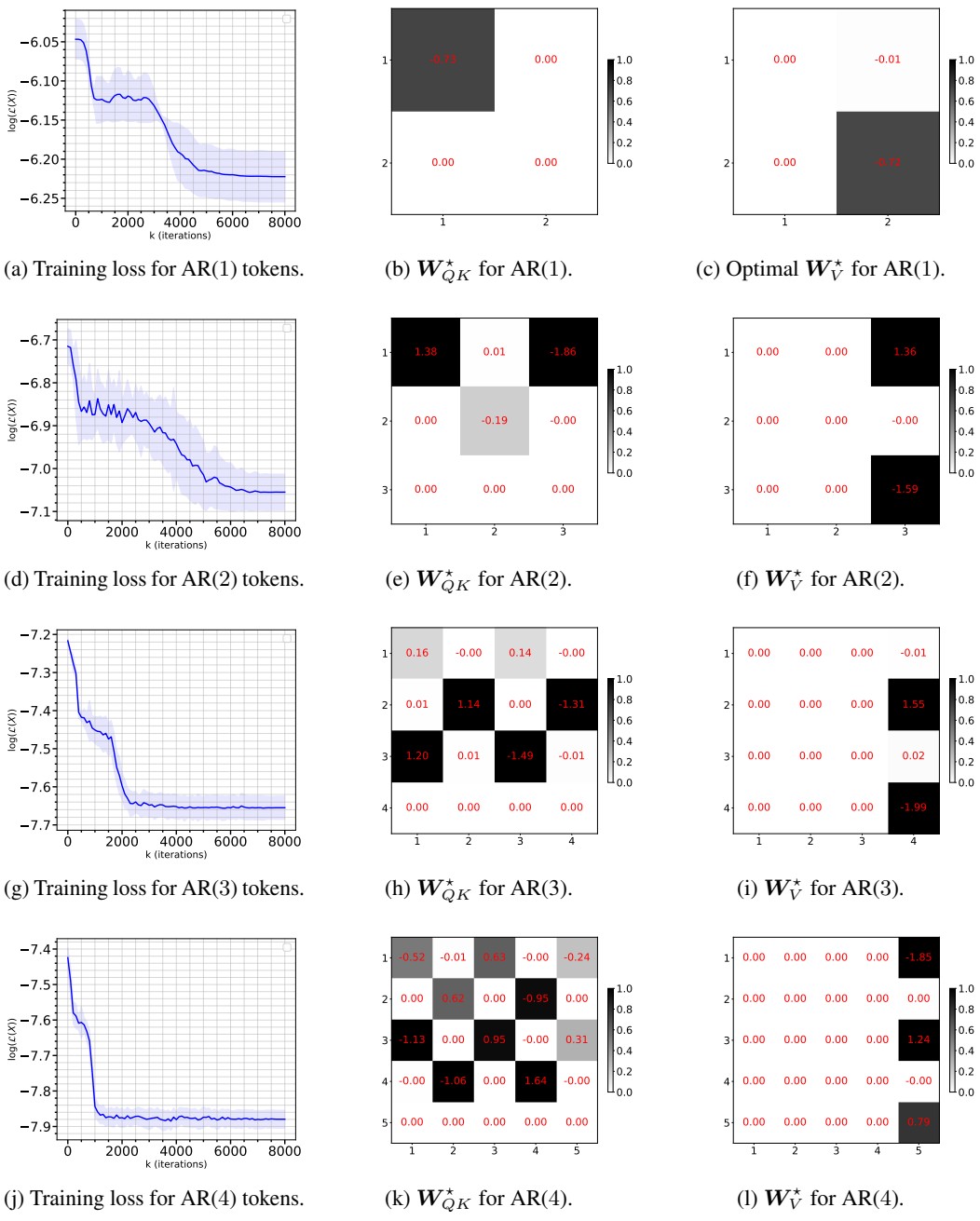

(a) Training loss for AR(1) tokens.     (b) $\boldsymbol{W}_{QK}^\star$ for AR(1).     (c) Optimal $\boldsymbol{W}_V^\star$ for AR(1).

(d) Training loss for AR(2) tokens.     (e) $\boldsymbol{W}_{QK}^\star$ for AR(2).     (f) $\boldsymbol{W}_V^\star$ for AR(2).

(g) Training loss for AR(3) tokens.     (h) $\boldsymbol{W}_{QK}^\star$ for AR(3).     (i) $\boldsymbol{W}_V^\star$ for AR(3).

(j) Training loss for AR(4) tokens.     (k) $\boldsymbol{W}_{QK}^\star$ for AR(4).     (l) $\boldsymbol{W}_V^\star$ for AR(4).

Figure 3: Experimental results for AR(1–4) with non-diagonal, symmetric $\boldsymbol{A}$, random $\boldsymbol{c}$ and isotropic $\Sigma_w$, setting (b) in Section 5, which align with the Lemma 4.1.

### B.4 EXPERIMENTS WITH NON-DIAGONAL, SYMMETRIC $\boldsymbol{A}$, RANDOM $\boldsymbol{c}$ AND NON-DIAGONAL, ANISOTROPIC $\Sigma_w$

The LDS which generates the training data is as follows. For each sequence, sample $\boldsymbol{v} \sim \text{Unif}([-1,1]^d)$; sample $\boldsymbol{Q} \sim \text{Haar}(O(d))$ and set $\boldsymbol{A} = \boldsymbol{Q}^\top \text{diag}(\boldsymbol{v})\boldsymbol{Q}$; sample $\boldsymbol{c} \sim \text{Unif}([-0.5, 0.5]^d)$. Set the process noise covariance $\Sigma_w = \mathbf{Q_w}^\top \text{diag}(\texttt{1e-2} \cdot [0.9, 0.95, 1.0, 1.05, 1.1])\mathbf{Q_w}$, where $\mathbf{Q_w} \sim \text{Haar}(O(d))$. Set $\sigma_v^2 = \texttt{1e-2}$.

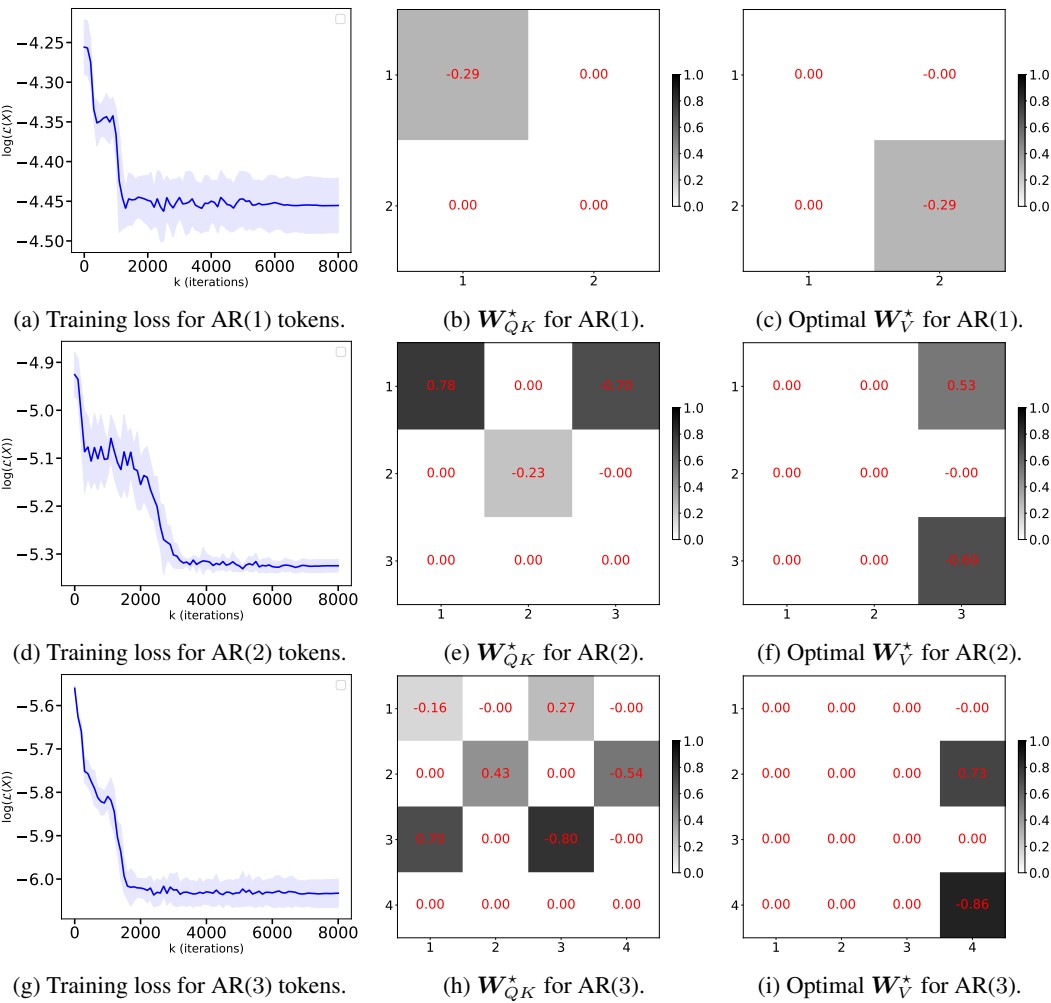

(a) Training loss for AR(1) tokens.     (b) $\boldsymbol{W}_{QK}^\star$ for AR(1).     (c) Optimal $\boldsymbol{W}_V^\star$ for AR(1).

(d) Training loss for AR(2) tokens.     (e) $\boldsymbol{W}_{QK}^\star$ for AR(2).     (f) Optimal $\boldsymbol{W}_V^\star$ for AR(2).

(g) Training loss for AR(3) tokens.     (h) $\boldsymbol{W}_{QK}^\star$ for AR(3).     (i) Optimal $\boldsymbol{W}_V^\star$ for AR(3).

Figure 4: Experimental results for AR(1–3) with non-diagonal, symmetric $\boldsymbol{A}$, random $\boldsymbol{c}$ and non-diagonal, anisotropic $\Sigma_w$, setting (c) in Section 5, which align with the Lemma 4.1.

### B.5 EXPERIMENTS WITH NON-DIAGONAL, NON-SYMMETRIC $\boldsymbol{A}$, RANDOM $\boldsymbol{c}$ AND ISOTROPIC $\Sigma_w$

The LDS which generates the training data is as follows.

For each sequence, sample $\boldsymbol{d} \sim \text{Unif}([-1, 1]^d)$, sample $\mathbf{P} = [p_{i,j}]_{i,j=1}^d$ by sampling $p_{i,j}$ i.i.d. from $\mathcal{U}([-1, 1])$, and set $\boldsymbol{A} = \mathbf{P}^{-1}\text{diag}(\boldsymbol{d})\mathbf{P}$. Sample $\boldsymbol{c} \sim \text{Unif}([-1, 1]^d)$. The noise covariances are set to $\Sigma_w = \texttt{1e-2}\boldsymbol{I}$ and $\sigma_v^2 = \texttt{1e-2}$.

In practice, we need to guarantee $\boldsymbol{P}$ is well conditioned. After sampling $p_{i,j}$ i.i.d. from $\mathcal{U}([-1, 1])$, we decompose $\boldsymbol{P} = \boldsymbol{QR}$, where $\boldsymbol{Q}$ is an orthogonal matrix and $\boldsymbol{R}$ is an upper-triangle matrix from the QR decomposition of $\boldsymbol{P}$. We modify the diagonals of $\boldsymbol{R}$ manually to make sure $\frac{\max_i \boldsymbol{R}_{ii}}{\min_i \boldsymbol{R}_{ii}} = 2$ and right multiply $\boldsymbol{Q}$ with the modified $\boldsymbol{R}$ to have the well conditioned $\boldsymbol{P}$.

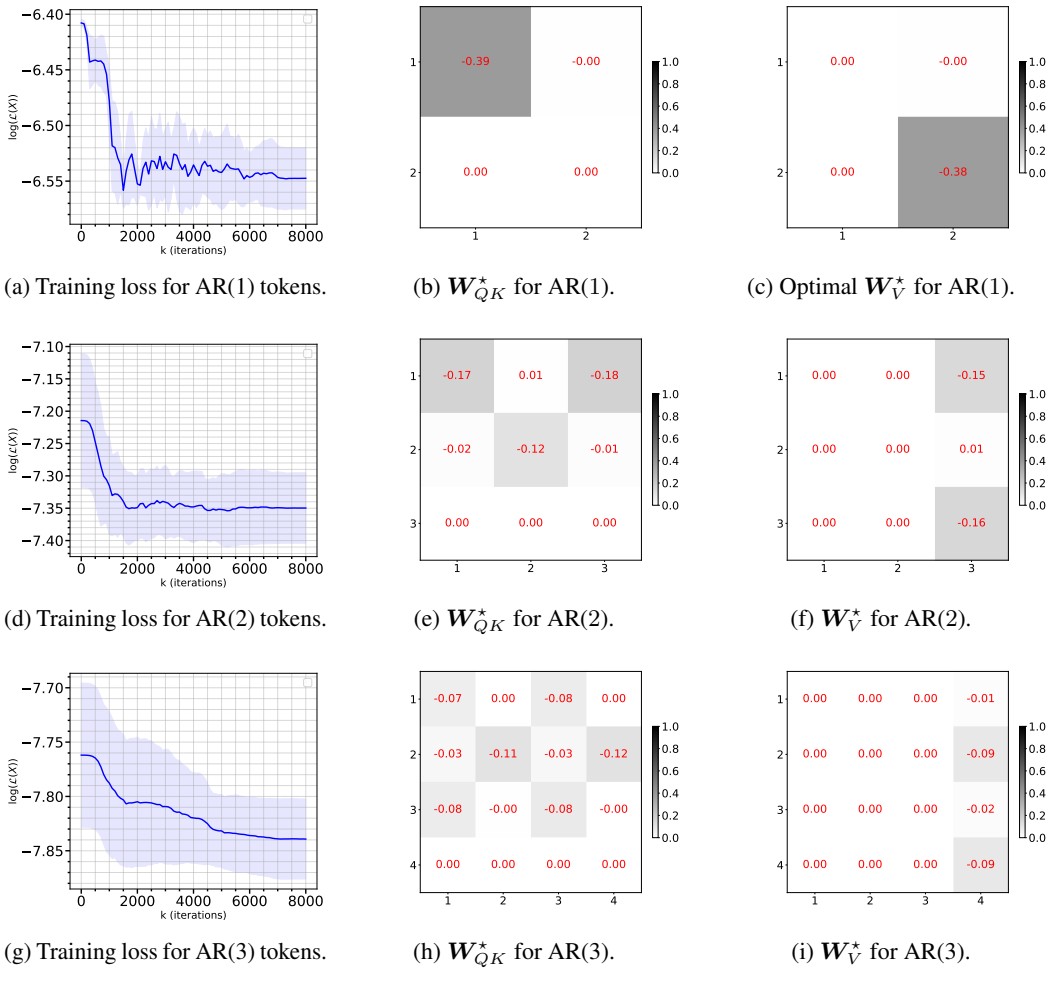

Figure 5: Experimental results for AR(1–3) with non-diagonal, non-symmetric $\boldsymbol{A}$, random $\boldsymbol{c}$ and isotropic $\Sigma_w$, setting (d) in Section 5, which align with the Lemma 4.1.

## C  SECTION 3 PROOFS

### C.1  PROOF OF TOKEN CONSTRUCTION LEMMA

**Lemma 3.1.** *For a given $s >= 1$, there exists an $s + 1$-headed linear attention layer with positional encoding which transforms input sequences $[y_1, y_2, \ldots, y_T]^\top$ into*

$$
\begin{bmatrix}
\bar{\boldsymbol{y}}_1 & \cdots & \bar{\boldsymbol{y}}_{T-s} & \boldsymbol{0}_{(s-1) \times (T-s-1)} \\
y_{s+1} & \cdots & 0 & \boldsymbol{0}_{T-s-1}^\top
\end{bmatrix}^\top .
$$

*The latter quantity is essentially equivalent to $\boldsymbol{Y}_0$ as defined in equation* (8).

**Proof.** We first define a matrix right-shift operator, which shifts each row one position to the right, padding the first column with zeros. Let $\gg: R^{m \times n} \to R^{m \times n}$ be $\gg(\boldsymbol{M}) = \boldsymbol{MR}$, where

$$
\boldsymbol{R} = \begin{bmatrix} 0 & \boldsymbol{0}_{n-1}^\top \\ \boldsymbol{0}_{n-1} & \boldsymbol{I}_{n-1} \end{bmatrix} . \tag{22}
$$

We follow Von Oswald et al. (2023a) in using the one-hot positional encodings, concatenated to the input sequence to obtain tokens $\{[y_t, \boldsymbol{e}_t]\}_{t=1}^T$. We define $s + 1$ attention heads given by

Define $\boldsymbol{W}_Q \in \mathbb{R}^{T+1 \times T}$, $\boldsymbol{W}_K \in \mathbb{R}^{T+1 \times T}$ and $\boldsymbol{W}_V \in \mathbb{R}^{T+1 \times s}$ as follows:

$$
\boldsymbol{W}_Q^h = \begin{bmatrix} \boldsymbol{0}_T^\top \\ \boldsymbol{I}_T \end{bmatrix}, \ \forall h \in [s+1]
$$

$$
(\boldsymbol{W}_K^h)^\top = \begin{bmatrix} \boldsymbol{0}_T, & \underbrace{\gg(\ldots \gg}_{h-1 \text{ times}}(\boldsymbol{I}_T)\ldots) \end{bmatrix}
$$

$$
\boldsymbol{W}_V^h = \begin{matrix} & 1 & \cdots & h & \cdots & s+1 \\ [\boldsymbol{0}_{T+1} & \cdots & \boldsymbol{e}_1 & \cdots & \boldsymbol{0}_{T+1}] \end{matrix}, \ \forall h \in [s+1] \tag{23}
$$

Each head then computes the following

$$
\underbrace{\begin{bmatrix} y_1 & 1 & 0 & \cdots & 0 \\ y_2 & 0 & 1 & \cdots & 0 \\ \vdots & \vdots & \vdots & \vdots & \vdots \\ y_T & 0 & 0 & \cdots & 1 \end{bmatrix}}_{=\boldsymbol{I}_T} \boldsymbol{W}_Q^k \ (\boldsymbol{W}_K^h)^\top \underbrace{\begin{bmatrix} y_1 & y_2 & y_3 & \cdots & y_T \\ 1 & 0 & 0 & \cdots & 0 \\ \vdots & \vdots & \vdots & \vdots & \vdots \\ 0 & 0 & 0 & \cdots & 1 \end{bmatrix}}_{=\begin{bmatrix} \boldsymbol{0}_{T-h+1 \times h-1} & \boldsymbol{I}_{T-h+1} \\ \boldsymbol{0}_{h-1 \times h-1} & \boldsymbol{0}_{h-1 \times T-h+1} \end{bmatrix}} \boldsymbol{W}_V \underbrace{\begin{bmatrix} y_1 & 1 & 0 & \cdots & 0 \\ y_2 & 0 & 1 & \cdots & 0 \\ \vdots & \vdots & \vdots & \vdots & \vdots \\ y_T & 0 & 0 & \cdots & 1 \end{bmatrix}}_{\begin{matrix} & 1 & \cdots & h & \cdots & s+1 \\ = \begin{bmatrix} 0 & \cdots & y_1 & \cdots & 0 \\ 0 & \cdots & y_2 & \cdots & 0 \\ \vdots & \vdots & \vdots & \vdots \\ 0 & \cdots & y_T & \cdots & 0 \end{bmatrix} \end{matrix}}
$$

$$
= \begin{matrix} & 1 & \cdots & h & \cdots & s+1 \\ & \begin{bmatrix} 0 & \cdots & y_h & \cdots & 0 \\ 0 & \cdots & y_{h+1} & \cdots & 0 \\ \vdots & \vdots & \vdots & \vdots \\ 0 & \cdots & y_T & \cdots & 0 \\ & & \boldsymbol{0}_{h \times s+1} & & \end{bmatrix} \end{matrix}
$$

Summing over the outputs of all heads, we get an equivalent representation to (8).  $\square$

## C.2 PROOF OF THE ALMOST SURE OBSERVABILITY OF THE LDS

We seek to show that Assumption 3.2 ensures LDS (1) observability w.p. 1. Note that the central symmetry of the distribution is irrelevant for this statement, and only relevant for the proofs in Section 4. We repeat Assumption 3.2 below for convenience.

**Assumption 3.2** (LDS family). *The system matrix $\boldsymbol{A} \in \mathbb{R}^{d \times d}$ is sampled from a centrally symmetric distribution supported on $\left\{ \boldsymbol{M} \in \mathbb{R}^{d \times d} \mid \rho(\boldsymbol{M}) \leq 1 \right\}$, for which it holds that*

$$\mathbb{P}\left( \left\{ \boldsymbol{A} \mid \exists i, j \in [d], \; s.t. \; \lambda_i(\boldsymbol{A}) = \lambda_j(\boldsymbol{A}) \right\} \right) = 0. \tag{9}$$

*In other words, $\boldsymbol{A}$ has a simple spectrum almost surely. The observation vector $\boldsymbol{c} \in \mathbb{R}^d$ is sampled independently, from a distribution that is absolutely continuous w.r.t. the Lebesgue measure over $\mathbb{R}^d$.*

**Lemma C.1.** *Assumption 3.2 ensures the pair $(\boldsymbol{A}, \boldsymbol{c})$ is observable w.p. 1.*

**Proof.** Since $\boldsymbol{A}$ has distinct eigenvalues w.p. 1 (the simple spectrum condition), it is (block) diagonalizable almost surely, and its eigenvectors $\{\boldsymbol{v}_1, \dots \boldsymbol{v}_d\}$ are linearly independent. Therefore, observability is ensured if $\boldsymbol{c}^\top \boldsymbol{v}_i \neq 0$ almost surely for all $i \in [d]$.

Since $\boldsymbol{c}$ is sampled from a distribution that is absolutely continuous w.r.t. the Lebesgue measure in $\mathbb{R}^d$, we want to prove that the set

$$\mathcal{U} = \bigcup_{i=1}^{d} \left\{ \boldsymbol{c} \in \boldsymbol{R}^d \mid \boldsymbol{c}^\top \boldsymbol{v}_i = 0 \right\}$$

has zero Lebesgue measure in the ambient $\mathbb{R}^d$. Each collection $\left\{ \boldsymbol{c} \in \boldsymbol{R}^d \mid \boldsymbol{c}^\top \boldsymbol{v}_i = 0 \right\}$ forms a proper subspace of $\mathbb{R}^d$ with dimension at most $d-1$ (it can be less, for complex $\boldsymbol{v}_i$). Therefore, its Lebesgue measure is null (see, e.g., (Royden & Fitzpatrick, 2010, pg. 435)).

Since $\mathcal{U}$ is a finite union of measure zero sets, it is itself measure zero. Hence, observability holds w.p. 1. □

# D    SECTION 4 PROOFS

## D.1    PRELIMINARIES

Since we're dealing with data generated from stochastic processes, our proofs will heavily rely on taking expectations conditioned on randomness up to a certain point in the process. In what follows, we formalize the natural filtrations with respect to process (1).

We denote the natural filtration associated with (1). as $\{\mathcal{F}_t\}_{t \geq 0}$, where

$$\mathcal{F}_t := \sigma(\boldsymbol{A}, \boldsymbol{c}, \boldsymbol{x}_0, \boldsymbol{w}_0, \ldots, \boldsymbol{w}_{t-1}, v_0, \ldots, v_{t-1}), \qquad t \geq 0. \tag{24}$$

By convention, when $t = 0$ the sets of noise variables are empty, and we define

$$\mathcal{F}_0 = \sigma(\boldsymbol{A}, \boldsymbol{c}, \boldsymbol{x}_0), \tag{25}$$

to illustrate that $\boldsymbol{A}$ and $\boldsymbol{c}$ are sampled once at time $0$ and then remain fixed.

It follows that

(a)  $\mathcal{F}_t \subseteq \mathcal{F}_{t+1}, \forall t \geq 0$

(b)  $\boldsymbol{x}_t$ is $\mathcal{F}_t$-measurable for all $t \geq 0$.

(c)  $y_t$ is $\mathcal{F}_{t+1}$-measurable (since $y_t$ depends on $v_t$)

(d)  The noise at time $t$ is independent on the respective filtration: $\boldsymbol{w}_t \perp\!\!\!\perp \mathcal{F}_t$, $v_t \perp\!\!\!\perp \mathcal{F}_t$, for all $t \geq 0$.

## D.2    AUXILIARY RESULTS AND TECHNICAL LEMMATA

**Theorem D.1** (Isserlis (1918)). *Let* $\boldsymbol{y} = [y_1, y_2, \ldots, y_n]^\top \sim \mathcal{N}_n(0, \Sigma)$ *be an $n$-dimensional, mean-zero multivariate normal vector. Then, for any even integer $n$,*

$$\mathbb{E}[y_1 y_2 \cdots y_n] = \sum_{p \in \mathrm{PP}(n)} \prod_{(\ell, r) \in p} \mathbb{E}[y_\ell y_r],$$

*where $\mathrm{PP}(n)$ denotes the set of all pairwise partitions of $[n]$ into disjoint pairs. If $n$ is odd, then* $\mathbb{E}[y_1 y_2 \cdots y_n] = 0$.

**Lemma D.1.** *Given random vectors $\boldsymbol{z}, \boldsymbol{w}, \boldsymbol{q} \in \mathbb{R}^d$ and assuming that $\boldsymbol{w}$ is independent of $\boldsymbol{z}, \boldsymbol{q}$ and the relevant integrability conditions hold, then*

$$\mathbb{E}\left[\boldsymbol{z}^\top \boldsymbol{w} \boldsymbol{w}^\top \boldsymbol{q}\right] = \mathbb{E}\left[\boldsymbol{z}^\top \mathbb{E}[\boldsymbol{w} \boldsymbol{w}^\top] \boldsymbol{q}\right] \tag{26}$$

**Proof.**  We use the towering property of expectations,

$$\mathbb{E}\left[\boldsymbol{z}^\top \boldsymbol{w} \boldsymbol{w}^\top \boldsymbol{z}\right] = \mathbb{E}\left[\boldsymbol{z}^\top \mathbb{E}\left[\boldsymbol{w} \boldsymbol{w}^\top \mid \boldsymbol{z}, \boldsymbol{q}\right] \boldsymbol{q}\right]$$
$$= \mathbb{E}\left[\boldsymbol{z}^\top \mathbb{E}\left[\boldsymbol{w} \boldsymbol{w}^\top\right] \boldsymbol{q}\right],$$

where the last line follows from the quantities' independence.  $\square$

**Lemma D.2.** *Let the sequence $\{y_i\}_{i \geq 0}$ be generated by an LDS (1) sampled according to Assumption 3.2. For time indices $0 \leq i \leq j$, it holds that*

$$\mathbb{E}[y_i y_j] = \mathbb{E}\left[\boldsymbol{c}^\top \boldsymbol{A}^i \Sigma_{\boldsymbol{x}_0} (\boldsymbol{A}^\top)^j \boldsymbol{c}\right] + \sum_{k=0}^{i-1} \mathbb{E}\left[\boldsymbol{c}^\top \boldsymbol{A}^{i-1-k} \Sigma_{\boldsymbol{w}} (\boldsymbol{A}^\top)^{j-1-k} \boldsymbol{c}\right] + \mathbf{1}_{\{i=j\}} \sigma_v^2, \tag{27}$$

*where $\mathbf{1}_{\{i=j\}}$ takes the value $1$ if $i = j$ and $0$ otherwise.*

**Proof.** For process (1) it holds that

$$\boldsymbol{x}_j = \boldsymbol{A}^{j-i}\boldsymbol{x}_i + \sum_{k=i}^{j-1}\boldsymbol{A}^{j-1-k}\boldsymbol{w}_k$$

and therefore

$$y_j = \boldsymbol{c}^\top\boldsymbol{A}^{j-i}\boldsymbol{x}_i + \sum_{k=i}^{j-1}\boldsymbol{c}^\top\boldsymbol{A}^{j-1-k}\boldsymbol{w}_k + v_j.$$

The product of scalars $y_i y_j$ therefore takes the form

$$y_i y_j = y_i y_j^\top$$

$$= (\boldsymbol{c}^\top\boldsymbol{x}_i + v_i)\left(\boldsymbol{c}^\top\boldsymbol{A}^{j-i}\boldsymbol{x}_i + \sum_{k=i}^{j-1}\boldsymbol{c}^\top\boldsymbol{A}^{j-1-k}\boldsymbol{w}_k + v_j\right)^\top$$

$$= \boldsymbol{c}^\top\boldsymbol{x}_i\boldsymbol{x}_i^\top(\boldsymbol{A}^\top)^{j-i}\boldsymbol{c} + \sum_{k=i}^{j-1}\boldsymbol{c}^\top\boldsymbol{x}_i\boldsymbol{w}_k^\top(\boldsymbol{A}^\top)^{j-1-k}\boldsymbol{c} + \boldsymbol{c}^\top\boldsymbol{x}_i v_j$$

$$+ v_i\boldsymbol{x}_i^\top(\boldsymbol{A}^\top)^{j-i}\boldsymbol{c} + \sum_{k=i}^{j-1}v_i\boldsymbol{w}_k^\top(\boldsymbol{A}^\top)^{j-1-k}\boldsymbol{c} + v_i v_j. \qquad (28)$$

Now, observing that $\mathbb{E}[y_i y_j] = \mathbb{E}\left[\mathbb{E}\left[y_i y_j \mid \mathcal{F}_i\right]\right]$ and remembering that $\boldsymbol{x}_i, \boldsymbol{A}, \boldsymbol{c}$ are $\mathcal{F}_i$-measurable, and that for all $i$ and $p \ge i$, $\boldsymbol{w}_p \perp\!\!\!\perp \mathcal{F}_i$ and $v_p \perp\!\!\!\perp \mathcal{F}_i$, and $\boldsymbol{w}_p \perp\!\!\!\perp v_q, \forall p, q \ge 0$, we have

$$\mathbb{E}\left[\boldsymbol{c}^\top\boldsymbol{x}_i\boldsymbol{x}_i^\top(\boldsymbol{A}^\top)^{j-i}\boldsymbol{c}\,\middle|\,\mathcal{F}_i\right] = \boldsymbol{c}^\top\boldsymbol{x}_i\boldsymbol{x}_i^\top(\boldsymbol{A}^\top)^{j-i}\boldsymbol{c},$$

$$\mathbb{E}\left[\sum_{k=i}^{j-1}\boldsymbol{c}^\top\boldsymbol{x}_i\boldsymbol{w}_k^\top(\boldsymbol{A}^\top)^{j-1-k}\boldsymbol{c}\,\middle|\,\mathcal{F}_i\right] = \sum_{k=i}^{j-1}\boldsymbol{c}^\top\boldsymbol{x}_i\mathbb{E}\left[\boldsymbol{w}_k^\top\right](\boldsymbol{A}^\top)^{j-1-k}\boldsymbol{c} = 0,$$

$$\mathbb{E}\left[\boldsymbol{c}^\top\boldsymbol{x}_i v_j\,\middle|\,\mathcal{F}_i\right] = \boldsymbol{c}^\top\boldsymbol{x}_i\mathbb{E}[v_j] = 0,$$

$$\mathbb{E}\left[v_i\boldsymbol{x}_i^\top(\boldsymbol{A}^\top)^{j-i}\boldsymbol{c}\,\middle|\,\mathcal{F}_i\right] = \mathbb{E}[v_i]\,\boldsymbol{x}_i^\top(\boldsymbol{A}^\top)^{j-i}\boldsymbol{c} = 0,$$

$$\mathbb{E}\left[\sum_{k=i}^{j-1}v_i\boldsymbol{w}_k^\top(\boldsymbol{A}^\top)^{j-1-k}\boldsymbol{c}\,\middle|\,\mathcal{F}_i\right] = \sum_{k=i}^{j-1}\mathbb{E}[v_i]\mathbb{E}[\boldsymbol{w}_k^\top](\boldsymbol{A}^\top)^{j-1-k}\boldsymbol{c} = 0,$$

$$\mathbb{E}\left[v_i v_j \mid \mathcal{F}_i\right] = \mathbb{E}[v_i v_j] = \mathbf{1}_{\{i=j\}}\sigma_v^2.$$

Therefore,

$$\mathbb{E}[y_i y_j] = \mathbb{E}[\mathbb{E}[y_i y_j \mid \mathcal{F}_i]] = \mathbb{E}\left[\boldsymbol{c}^\top\boldsymbol{x}_i\boldsymbol{x}_i^\top(\boldsymbol{A}^\top)^{j-i}\boldsymbol{c}\right] + \mathbf{1}_{\{i=j\}}\sigma_v^2. \qquad (29)$$

Noting that $\boldsymbol{x}_i = \boldsymbol{A}^i\boldsymbol{x}_0 + \sum_{k=0}^{i-1}\boldsymbol{A}^{i-1-k}\boldsymbol{w}_k$, we further unroll the first term inside the expectation in (29) and get

$$\boldsymbol{c}^\top\boldsymbol{x}_i\boldsymbol{x}_i^\top(\boldsymbol{A}^\top)^{j-i}\boldsymbol{c} = \left[\boldsymbol{c}^\top\boldsymbol{A}^i\boldsymbol{x}_0 + \sum_{k=0}^{i-1}\boldsymbol{c}^\top\boldsymbol{A}^{i-1-k}\boldsymbol{w}_k\right]\left[\boldsymbol{x}_0^\top(\boldsymbol{A}^\top)^j\boldsymbol{c} + \sum_{k=0}^{i-1}\boldsymbol{w}_k^\top(\boldsymbol{A}^\top)^{j-1-k}\boldsymbol{c}\right]$$

$$= \boldsymbol{c}^\top\boldsymbol{A}^i\boldsymbol{x}_0\boldsymbol{x}_0^\top(\boldsymbol{A}^\top)^j\boldsymbol{c} + \sum_{k=0}^{i-1}\boldsymbol{c}^\top\boldsymbol{A}^i\boldsymbol{x}_0\boldsymbol{w}_k^\top(\boldsymbol{A}^\top)^{j-1-k}\boldsymbol{c}$$

$$+ \sum_{k=0}^{i-1}\boldsymbol{c}^\top\boldsymbol{A}^{i-1-k}\boldsymbol{w}_k\boldsymbol{x}_0^\top(\boldsymbol{A}^\top)^j\boldsymbol{c} + \sum_{k,l=0}^{i-1}\boldsymbol{c}^\top\boldsymbol{A}^{i-1-k}\boldsymbol{w}_k\boldsymbol{w}_l^\top(\boldsymbol{A}^\top)^{j-1-l}\boldsymbol{c}.$$

$$(30)$$

Using $\boldsymbol{w}_p \perp\!\!\!\perp \mathcal{F}_0 \subset \mathcal{F}_i, \forall p \geq 0$ and $\boldsymbol{w}_p \perp\!\!\!\perp \boldsymbol{w}_q, \forall p \neq q$ in conjunction with (30) and Lemma D.1 we get

$$\mathbb{E}\left[\boldsymbol{c}^\top \boldsymbol{x}_i \boldsymbol{x}_i^\top (\boldsymbol{A}^\top)^{j-i} \boldsymbol{c} \,\middle|\, \mathcal{F}_0\right] = \boldsymbol{c}^\top \boldsymbol{A}^i \boldsymbol{x}_0 \boldsymbol{x}_0^\top (\boldsymbol{A}^\top)^j \boldsymbol{c} + \sum_{k=0}^{i-1} \boldsymbol{c}^\top \boldsymbol{A}^{i-1-k} \Sigma_{\boldsymbol{w}} (\boldsymbol{A}^\top)^{j-1-k} \boldsymbol{c} \quad (31)$$

Furthermore, noting that $\sigma(\boldsymbol{A}, \boldsymbol{c}) \subset \mathcal{F}_0$, we have that

$$\mathbb{E}\left[\boldsymbol{c}^\top \boldsymbol{x}_i \boldsymbol{x}_i^\top (\boldsymbol{A}^\top)^{j-i} \boldsymbol{c} \,\middle|\, \boldsymbol{A}, \boldsymbol{c}\right] = \boldsymbol{c}^\top \boldsymbol{A}^i \Sigma_{\boldsymbol{x}_0} (\boldsymbol{A}^\top)^j \boldsymbol{c} + \sum_{k=0}^{i-1} \boldsymbol{c}^\top \boldsymbol{A}^{i-1-k} \Sigma_{\boldsymbol{w}} (\boldsymbol{A}^\top)^{j-1-k} \boldsymbol{c}. \quad (32)$$

Taking full expectation in (32), and plugging everything back into (29), we get the stated result

$$\mathbb{E}\left[y_i y_j\right] = \mathbb{E}\left[\boldsymbol{c}^\top \boldsymbol{A}^i \Sigma_{\boldsymbol{x}_0} (\boldsymbol{A}^\top)^j \boldsymbol{c}\right] + \sum_{k=0}^{i-1} \mathbb{E}\left[\boldsymbol{c}^\top \boldsymbol{A}^{i-1-k} \Sigma_{\boldsymbol{w}} (\boldsymbol{A}^\top)^{j-1-k} \boldsymbol{c}\right] + \mathbf{1}_{\{i=j\}} \sigma_v^2. \qquad \square$$

**Lemma D.3.** *Let $\{y_i\}_{i \geq 0}$ be a sequence of observations generated by an LDS* (1) *sampled according to Assumption 3.2. Then,*

    *(a) if $i + j = 2p + 1$ for some $p \in \mathbb{N}_+$, $\mathbb{E}\left[y_i y_j\right] = 0$;*

    *(b) if $i + j + k + l = 2p + 1$ for some $p \in \mathbb{N}_+$, $\mathbb{E}\left[y_i y_j y_k y_l\right] = 0$;*

    *(c) if $i + j + k + l + m + n = 2p + 1$ for some $p \in \mathbb{N}_+$, $\mathbb{E}\left[y_i y_j y_k y_l y_m y_n\right] = 0$.*

*Note that there is no condition on the indices being pairwise distinct.*

**Proof.** To prove point (a), we start from the expression derived in Lemma D.2.

$$\mathbb{E}\left[y_i y_j\right] = \mathbb{E}\left[\boldsymbol{c}^\top \boldsymbol{A}^i \Sigma_{\boldsymbol{x}_0} (\boldsymbol{A}^\top)^j \boldsymbol{c}\right] + \sum_{k=0}^{i-1} \mathbb{E}\left[\boldsymbol{c}^\top \boldsymbol{A}^{i-1-k} \Sigma_{\boldsymbol{w}} (\boldsymbol{A}^\top)^{j-1-k} \boldsymbol{c}\right] + \mathbf{1}_{\{i=j\}} \sigma_v^2$$

Clearly, since $i + j$ is odd, it holds that $i \neq j$ and hence the third term is zero. Furthermore, since $\boldsymbol{A}$ has a centrally symmetric distribution, we have that

$$\begin{aligned}
\mathbb{E}\left[\boldsymbol{c}^\top \boldsymbol{A}^i \Sigma_{\boldsymbol{x}_0} (\boldsymbol{A}^\top)^j \boldsymbol{c}\right] &= \mathbb{E}\left[\boldsymbol{c}^\top (-\boldsymbol{A})^i \Sigma_{\boldsymbol{x}_0} (-\boldsymbol{A}^\top)^j \boldsymbol{c}\right] \\
&= (-1)^{i+j} \mathbb{E}\left[\boldsymbol{c}^\top \boldsymbol{A}^i \Sigma_{\boldsymbol{x}_0} (\boldsymbol{A}^\top)^j \boldsymbol{c}\right],
\end{aligned} \quad (33)$$

implying that $\mathbb{E}\left[\boldsymbol{c}^\top \boldsymbol{A}^i \Sigma_{\boldsymbol{x}_0} (\boldsymbol{A}^\top)^j \boldsymbol{c}\right] = 0$. We apply a similar reasoning for the other term and obtain that

$$\mathbb{E}\left[y_i y_j\right] = 0,$$

thus proving the first point.

For both points (b) and (c), we will rely on Isserlis's theorem, which we replicate in Theorem D.1 for convenience. Note that conditioned, on $\boldsymbol{A}$ and $\boldsymbol{c}$, the vectors $[y_i y_j y_k y_l | \boldsymbol{A}, \boldsymbol{c}]$ and $[y_i y_j y_k y_l y_m y_n | \boldsymbol{A}, \boldsymbol{c}]$ are jointly Gaussian since they are linear transformations of the jointly Gaussian vectors $\boldsymbol{r}_1 = \left[\boldsymbol{x}_0^\top, \boldsymbol{w}_0^\top, \ldots \boldsymbol{w}_{\max\{i,j,k,l\}}^\top, v_0, \ldots v_{\max\{i,j,k,l\}}\right]^\top$ and $\boldsymbol{r}_2 = \left[\boldsymbol{x}_0^\top, \boldsymbol{w}_0^\top, \ldots \boldsymbol{w}_{\max\{i,j,k,l,m,n\}}^\top, v_0, \ldots v_{\max\{i,j,k,l,m,n\}}\right]^\top$, respectively. We can therefore apply the towering property along with Isserlis's result to get

$$\begin{aligned}
\mathbb{E}\left[y_i y_j y_k y_l\right] &= \mathbb{E}\left[\mathbb{E}\left[y_i y_j y_k y_l \,\middle|\, \boldsymbol{A}, \boldsymbol{c}\right]\right] \\
&= \mathbb{E}\Big[\, \mathbb{E}\left[y_i y_j \,\middle|\, \boldsymbol{A}, \boldsymbol{c}\right] \mathbb{E}\left[y_k y_l \,\middle|\, \boldsymbol{A}, \boldsymbol{c}\right] + \mathbb{E}\left[y_i y_k \,\middle|\, \boldsymbol{A}, \boldsymbol{c}\right] \mathbb{E}\left[y_j y_l \,\middle|\, \boldsymbol{A}, \boldsymbol{c}\right] \\
&\qquad\qquad + \mathbb{E}\left[y_i y_l \,\middle|\, \boldsymbol{A}, \boldsymbol{c}\right] \mathbb{E}\left[y_j y_k \,\middle|\, \boldsymbol{A}, \boldsymbol{c}\right] \Big], \quad (34)
\end{aligned}$$

since $\mathrm{PP}(\{i,j,k,l\}) = \{\{(i,j),(k,l)\},\{(i,k),(j,l)\},\{(i,l),(j,k)\}\}$. Since $i+j+k+l$ is odd, the two pairs inside any $q \in \mathrm{PP}(\{i,j,k,l\})$ must have different parities (i.e., one even, one odd). W.l.o.g, we analyze the first term in (34), assuming $0 \leq i \leq j \leq k \leq l$. From (32), we know that

$$
\mathbb{E}\left[y_i y_j \mid \boldsymbol{A}, \boldsymbol{c}\right] \mathbb{E}\left[y_k y_l \mid \boldsymbol{A}, \boldsymbol{c}\right] = \left[\boldsymbol{c}^\top \boldsymbol{A}^i \Sigma_{\boldsymbol{x}_0}(\boldsymbol{A}^\top)^j \boldsymbol{c} + \sum_{t=0}^{i-1} \boldsymbol{c}^\top \boldsymbol{A}^{i-1-t}\Sigma_{\boldsymbol{w}}(\boldsymbol{A}^\top)^{j-1-t}\boldsymbol{c} + \mathbf{1}_{\{i=j\}}\sigma_v^2\right]
$$
$$
\left[\boldsymbol{c}^\top \boldsymbol{A}^k \Sigma_{\boldsymbol{x}_0}(\boldsymbol{A}^\top)^l \boldsymbol{c} + \sum_{t=0}^{k-1} \boldsymbol{c}^\top \boldsymbol{A}^{k-1-t}\Sigma_{\boldsymbol{w}}(\boldsymbol{A}^\top)^{l-1-t}\boldsymbol{c} + \mathbf{1}_{\{k=l\}}\sigma_v^2\right]
$$

(35)

Assume w.l.o.g that $i+j$ is even, and $k+l$ is odd. This implies that $\mathbf{1}_{\{k=l\}} = 0$. Taking full expectation on both sides and developing the product, we get

$$
\mathbb{E}\left[\,\mathbb{E}\left[y_i y_j \mid \boldsymbol{A}, \boldsymbol{c}\right] \mathbb{E}\left[y_k y_l \mid \boldsymbol{A}, \boldsymbol{c}\right]\,\right]
$$
$$
= \mathbb{E}\left[\,\boldsymbol{c}^\top \boldsymbol{A}^i \Sigma_{\boldsymbol{x}_0}(\boldsymbol{A}^\top)^j \boldsymbol{c}\boldsymbol{c}^\top \boldsymbol{A}^k \Sigma_{\boldsymbol{x}_0}(\boldsymbol{A}^\top)^l \boldsymbol{c}\,\right]
$$
$$
+ \sum_{t=0}^{k-1} \mathbb{E}\left[\,\boldsymbol{c}^\top \boldsymbol{A}^i \Sigma_{\boldsymbol{x}_0}(\boldsymbol{A}^\top)^j \boldsymbol{c}\boldsymbol{c}^\top \boldsymbol{A}^{k-1-t}\Sigma_{\boldsymbol{w}}(\boldsymbol{A}^\top)^{l-1-t}\boldsymbol{c}\,\right]
$$
$$
+ \sum_{t=0}^{i-1} \mathbb{E}\left[\,\boldsymbol{c}^\top \boldsymbol{A}^{i-1-t}\Sigma_{\boldsymbol{w}}(\boldsymbol{A}^\top)^{j-1-t}\boldsymbol{c}\boldsymbol{c}^\top \boldsymbol{A}^k \Sigma_{\boldsymbol{x}_0}(\boldsymbol{A}^\top)^l \boldsymbol{c}\,\right]
$$
$$
+ \sum_{t=0}^{i-1}\sum_{s=0}^{k-1} \mathbb{E}\left[\boldsymbol{c}^\top \boldsymbol{A}^{i-1-t}\Sigma_{\boldsymbol{w}}(\boldsymbol{A}^\top)^{j-1-t}\boldsymbol{c}\,\boldsymbol{c}^\top \boldsymbol{A}^{k-1-s}\Sigma_{\boldsymbol{w}}(\boldsymbol{A}^\top)^{l-1-s}\boldsymbol{c}\right]
$$
$$
+ \mathbf{1}_{\{i=j\}}\sigma_v^2\,\mathbb{E}\left[\,\boldsymbol{c}^\top \boldsymbol{A}^k \Sigma_{\boldsymbol{x}_0}(\boldsymbol{A}^\top)^l \boldsymbol{c}\right]
$$
$$
+ \mathbf{1}_{\{i=j\}}\sigma_v^2 \sum_{t=0}^{k-1} \mathbb{E}\left[\,\boldsymbol{c}^\top \boldsymbol{A}^{k-1-t}\Sigma_{\boldsymbol{w}}(\boldsymbol{A}^\top)^{l-1-t}\boldsymbol{c}\,\right] \quad (36)
$$

Using the index parity assumptions and the reasoning based on the central symmetry of $\boldsymbol{A}'s$ distribution from (33), we get that all the terms on the RHS of (36) are zero. We treat the remaining terms in (34) similarly to get the final result in (b).

Finally, point (c) follows a similar path. We have

$$
\mathrm{PP}(\{i,j,k,l,m,n\}) = \{\{(i,j),(k,l),(m,n)\},\{(i,j),(k,m),(l,n)\},\{(i,j),(k,n),(l,m)\},
$$
$$
\{(i,k),(j,l),(m,n)\},\{(i,k),(j,m),(l,n)\},\{(i,k),(j,n),(l,m)\},
$$
$$
\{(i,l),(j,k),(m,n)\},\{(i,l),(j,m),(k,n)\},\{(i,l),(j,n),(k,m)\},
$$
$$
\{(i,m),(j,k),(l,n)\},\{(i,m),(j,l),(k,n)\},\{(i,m),(j,n),(k,l)\},
$$
$$
\{(i,n),(j,k),(l,m)\},\{(i,n),(j,l),(k,m)\},\{(i,n),(j,m),(k,l)\}\}.
$$

For the parity hypothesis to be satisfied, not that inside a set $q \in \mathrm{PP}(\{i,j,k,l,m,n\})$, at least one pair must have an odd parity, while the other two must be of the same parity (either even or odd). W.o.l.g let $0 \leq i \leq j \leq k \leq l \leq m \leq n$, pick the first set in $\mathrm{PP}(\{i,j,k,l,m,n\})$ above (the rest follow the same logic) and assume that $m+n$ is odd. By the same logic as before, we have that $\mathbf{1}_{\{m=n\}} = 0$ and

$$
\mathbb{E}\left[\,\mathbb{E}\left[y_i y_j \mid \boldsymbol{A}, \boldsymbol{c}\right] \mathbb{E}\left[y_k y_l \mid \boldsymbol{A}, \boldsymbol{c}\right] \mathbb{E}\left[y_m y_n \mid \boldsymbol{A}, \boldsymbol{c}\right]\,\right]
$$
$$
= \mathbb{E}\left[\left[\boldsymbol{c}^\top \boldsymbol{A}^i \Sigma_{\boldsymbol{x}_0}(\boldsymbol{A}^\top)^j \boldsymbol{c} + \sum_{t=0}^{i-1} \boldsymbol{c}^\top \boldsymbol{A}^{i-1-t}\Sigma_{\boldsymbol{w}}(\boldsymbol{A}^\top)^{j-1-t}\boldsymbol{c} + \mathbf{1}_{\{i=j\}}\sigma_v^2\right]\right.
$$
$$
\left[\boldsymbol{c}^\top \boldsymbol{A}^k \Sigma_{\boldsymbol{x}_0}(\boldsymbol{A}^\top)^l \boldsymbol{c} + \sum_{t=0}^{k-1} \boldsymbol{c}^\top \boldsymbol{A}^{k-1-t}\Sigma_{\boldsymbol{w}}(\boldsymbol{A}^\top)^{l-1-t}\boldsymbol{c} + \mathbf{1}_{\{k=l\}}\sigma_v^2\right]
$$
$$
\left.\left[\boldsymbol{c}^\top \boldsymbol{A}^m \Sigma_{\boldsymbol{x}_0}(\boldsymbol{A}^\top)^n \boldsymbol{c} + \sum_{t=0}^{k-1} \boldsymbol{c}^\top \boldsymbol{A}^{m-1-t}\Sigma_{\boldsymbol{w}}(\boldsymbol{A}^\top)^{n-1-t}\boldsymbol{c}\right]\right]
$$

Without computing, one can see that every term in the expanded product will have powers of $\boldsymbol{A}$ whose sum is odd. Therefore, using the centrally symmetric property of $\boldsymbol{A}$'s distribution, all the terms evaluate to zero, and point (c) is proven. $\qquad\square$

### D.3 PROOF OF LEMMA 4.1

**Lemma 4.1.** *For any $s \geq 1$, $j \in [s]$, the parameters $\boldsymbol{W}_{QK}$ and $\boldsymbol{W}_V$ having the structure*

$$\boldsymbol{W}_{QK} \;=\; \begin{bmatrix} \boldsymbol{R}_{1:s,\,1:(s+1)} \\ \boldsymbol{0}_{s+1}^\top \end{bmatrix}, \qquad \boldsymbol{W}_V \;=\; \begin{bmatrix} \boldsymbol{0}_{(s+1)\times s} & \boldsymbol{r}_{2+s \bmod 2\,:\,2\lceil \frac{s+1}{2}\rceil} \end{bmatrix}, \tag{14}$$

*with $\boldsymbol{R} := \mathbf{1}_{\lfloor \frac{s+1}{2}\rfloor \times \lceil \frac{s+1}{2}\rceil} \otimes \begin{bmatrix} \star & 0 \\ 0 & \star \end{bmatrix}$ and $\boldsymbol{r} := \mathbf{1}_{\lceil \frac{s+1}{2}\rceil} \otimes \begin{bmatrix} 0 \\ \star \end{bmatrix}$, ensure that LHS $(12)_{r,\ell} = 0$ whenever $r + l \in 2\mathbb{N}$ and $s + j \in 2\mathbb{Z}$, or $r + l \in 2\mathbb{N} + 1$ and $s + j \in 2\mathbb{Z} + 1$.*

**Proof.** Recall the in-context loss in (10) with a general AR($s$)-constructed input token matrix $\boldsymbol{Y}_0 = \begin{bmatrix} \bar{\boldsymbol{y}}_1 & \bar{\boldsymbol{y}}_2 & \cdots & \bar{\boldsymbol{y}}_{T-s-1} & \bar{\boldsymbol{y}}_{T-s} \\ y_{s+1} & y_{s+2} & \cdots & y_{T-1} & 0 \end{bmatrix}$ is defined as

$$\mathcal{L}(\theta) \;:=\; \mathbb{E}\left[ \left( \mathcal{T}_\theta\left(\boldsymbol{Y}_0\right)_{s+1,T-s} - y_T \right)^2 \right]. \tag{37}$$

For equations (38) to (42) below, we use the same reformulations as Ahn et al. (2023). The last column of the transformer's output above can be written as

$$\begin{bmatrix} \bar{\boldsymbol{y}}_{T-1} \\ 0 \end{bmatrix} = \begin{bmatrix} \bar{\boldsymbol{y}}_{T-1} \\ 0 \end{bmatrix} + \frac{1}{T-s-1}\boldsymbol{W}_V^\top \left( \sum_{i=1}^{T-s-1} \begin{bmatrix} \bar{\boldsymbol{y}}_i \bar{\boldsymbol{y}}_i^\top & \bar{\boldsymbol{y}}_i y_{i+s} \\ \bar{\boldsymbol{y}}_i^\top y_{i+s} & y_{i+s}^2 \end{bmatrix} \right) \boldsymbol{W}_{QK}^\top \begin{bmatrix} \bar{\boldsymbol{y}}_{T-s} \\ 0 \end{bmatrix}, \tag{38}$$

where the summation comes from the causal mask. Therefore, the transformer's prediction of $y_T$, $\mathcal{T}_\theta\left(\boldsymbol{Y}_0\right)_{s+1,T-s}$ can be written as

$$\frac{1}{T-s-1}\boldsymbol{b}^\top \underbrace{\left( \sum_{i=1}^{T-s-1} \begin{bmatrix} \bar{\boldsymbol{y}}_i \bar{\boldsymbol{y}}_i^\top & \bar{\boldsymbol{y}}_i y_{i+s} \\ \bar{\boldsymbol{y}}_i^\top y_{i+s} & y_{i+s}^2 \end{bmatrix} \right)}_{:=\bar{\boldsymbol{Y}} \in \mathbb{R}^{(s+1)\times(s+1)}} [\boldsymbol{z}_1 \boldsymbol{z}_2 \cdots \boldsymbol{z}_s] \, \bar{\boldsymbol{y}}_{T-s}, \tag{39}$$

where $\boldsymbol{b}^\top \in \mathbb{R}^{1\times(s+1)}$ is the last row of $\boldsymbol{W}_V^\top$ and $\boldsymbol{z}_j \in \mathbb{R}^{(s+1)}$ is the $j^{th}$ column of $\boldsymbol{W}_{QK}^\top$. So the in-context loss $\mathcal{L}(\boldsymbol{W}_V, \boldsymbol{W}_{QK})$ can be rewritten as a function of $\boldsymbol{b}^\top$ and $\boldsymbol{ZZ} = [\boldsymbol{z}_j]_{j=1}^s$

$$\mathcal{L}(\boldsymbol{b}, \boldsymbol{ZZ}) \;:=\; \mathbb{E}\left[ \left( \frac{1}{T-s-1}\boldsymbol{b}^\top \bar{\boldsymbol{Y}} \boldsymbol{ZZ} \bar{\boldsymbol{y}}_{T-s} - y_T \right)^2 \right]. \tag{40}$$

Plugging in the expression of $\bar{\boldsymbol{y}}_{T-s}$, the in-context loss is

$$\mathcal{L}(\boldsymbol{b}, \boldsymbol{ZZ}) = \mathbb{E}\left[ \left( \frac{1}{T-s-1}\boldsymbol{b}^\top \bar{\boldsymbol{Y}} [\boldsymbol{z}_1 \boldsymbol{z}_2 \cdots \boldsymbol{z}_s] \begin{bmatrix} y_{T-s} \\ y_{T-s+1} \\ \vdots \\ y_{T-1} \end{bmatrix} - y_T \right)^2 \right]$$

$$= \mathbb{E}\left[\left(\frac{1}{T-s-1}\sum_{k=1}^{s}\boldsymbol{b}^{\top}\bar{\boldsymbol{Y}}\boldsymbol{z}_k y_{T-s-1+k} - y_T\right)^2\right]$$

$$= \mathbb{E}\left[\left(\frac{1}{T-s-1}\sum_{k=1}^{s}\mathrm{Tr}(\bar{\boldsymbol{Y}}\boldsymbol{z}_k\boldsymbol{b}^{\top}) y_{T-s-1+k} - y_T\right)^2\right]$$

$$= \mathbb{E}\left[\left(\frac{1}{T-s-1}\sum_{k=1}^{s}\langle\bar{\boldsymbol{Y}}, \boldsymbol{b}\boldsymbol{z}_k^{\top}\rangle y_{T-s-1+k} - y_T\right)^2\right]. \tag{41}$$

We reparametrize the in-context loss using $\boldsymbol{X}_k \coloneqq \boldsymbol{b}\boldsymbol{z}_k^{\top}$

$$\mathcal{L}(\boldsymbol{X}_{k\in[s]}) = \mathbb{E}\left[\left(\frac{1}{T-s-1}\sum_{k=1}^{s}\langle\bar{\boldsymbol{Y}}, \boldsymbol{X}_k\rangle y_{T-s-1+k} - y_T\right)^2\right]. \tag{42}$$

Note that the gradient of the in-context loss with respect to each $\boldsymbol{X}_j$ is

$$\nabla_{\boldsymbol{X}_j}\mathcal{L}(\boldsymbol{X}_{k\in[s]}) = 2\mathbb{E}\left[\left(\frac{1}{T-s-1}\sum_{k=1}^{s}\langle\bar{\boldsymbol{Y}}, \boldsymbol{X}_k\rangle y_{T-s-1+k} - y_T\right) y_{T-s-1+j}\bar{\boldsymbol{Y}}\right]. \tag{43}$$

The gradient $\nabla_{\boldsymbol{X}_j}\mathcal{L}(\boldsymbol{X}_{k\in[s]})$ is a sum of two terms, $\nabla_{\boldsymbol{X}_j}\mathcal{L}(\boldsymbol{X}_{k=1\cdots s}) = \mathbf{T}_{\boldsymbol{X}_j}^{(1)} + \mathbf{T}_{\boldsymbol{X}_j}^{(2)}$, where, replacing $\bar{\boldsymbol{Y}}$ we have

$$\mathbf{T}_{\boldsymbol{X}_j}^{(1)} \coloneqq \frac{2}{T-s-1}\mathbb{E}\left[\sum_{k=1}^{s}\langle\bar{\boldsymbol{Y}}, \boldsymbol{X}_k\rangle y_{T-s-1+k}y_{T-s-1+j}\sum_{i=1}^{T-s-1}\begin{bmatrix}\bar{\boldsymbol{y}}_i\bar{\boldsymbol{y}}_i^{\top} & \bar{\boldsymbol{y}}_i y_{i+s} \\ \bar{\boldsymbol{y}}_i^{\top} y_{i+s} & y_{i+s}^2\end{bmatrix}\right] \tag{44}$$

$$\mathbf{T}_{\boldsymbol{X}_j}^{(2)} \coloneqq -2\mathbb{E}\left[y_T y_{T-s-1+j}\sum_{i=1}^{T-s-1}\begin{bmatrix}\bar{\boldsymbol{y}}_i\bar{\boldsymbol{y}}_i^{\top} & \bar{\boldsymbol{y}}_i y_{i+s} \\ \bar{\boldsymbol{y}}_i^{\top} y_{i+s} & y_{i+s}^2\end{bmatrix}\right]. \tag{45}$$

Each matrix element of $\mathbf{T}_{\boldsymbol{X}_j}^{(2)}$ has the form

$$\sum_{i=1}^{T-s-1} 2\mathbb{E}\left[y_T y_{T-s-1+j}y_{i+m}y_{i+n}\right] \tag{46}$$

with $j \in [1, s]$, $m \in [0, s]$ and $n \in [0, s]$.

The sum of $y$'s indices in (46) for each term in the above sum is $2T + 2i + (m + n - s - 1 + j)$. The parity is determined by that of $m + n - s - 1 + j$ and is independent of the sum counter $i$ (i.e., the same for all terms). According to Lemma D.3, (46) is 0 if $(m + n - s - 1 + j)$ is odd, and of arbitrary value if it is even. So a general matrix element of $\mathbf{T}_{\boldsymbol{X}_j}^{(2)}$ is 0 if $(m + n - s - 1 + j)$ is odd and of arbitrary value if $(m + n - s - 1 + j)$ is even.

For a given AR($s$)-type token ($s$ is fixed) and a specific $j$, whether a matrix element of $\mathbf{T}^{(2)}_{\boldsymbol{X}_j}$ is 0 only depends on $m+n$ (its position in the matrix). So,

$$
\mathbf{T}^2_{\boldsymbol{X}_j} = \begin{cases}
\begin{bmatrix}
\star & 0 & \star & \cdots & \cdots & \\
0 & \star & 0 & \star & & \vdots \\
\star & 0 & \star & \ddots & \ddots & \vdots \\
\vdots & \star & \ddots & \ddots & \ddots & \star \\
\vdots & & \ddots & \ddots & \star & 0 \\
\cdots & \cdots & \star & 0 & \star &
\end{bmatrix} & , \text{if } |j-s-1| \text{ is even;} \\[2em]
\begin{bmatrix}
0 & \star & 0 & \cdots & \cdots & \\
\star & 0 & \star & 0 & & \vdots \\
0 & \star & 0 & \ddots & \ddots & \vdots \\
\vdots & 0 & \ddots & \ddots & \ddots & 0 \\
\vdots & & \ddots & \ddots & 0 & \star \\
\cdots & \cdots & 0 & \star & 0 &
\end{bmatrix} & , \text{if } |j-s-1| \text{ is odd.}
\end{cases}
$$

We now turn to $\mathbf{T}^{(1)}_{\boldsymbol{X}_j}$ with the end goal of finding a parameter configuration that matches the sparsity pattern of $\mathbf{T}^{(2)}_{\boldsymbol{X}_j}$. For this section, assume $s$ is odd (the other case follows similarly). First, let $\boldsymbol{X}_k := \left[ x^{(k)}_{i,j} \right]^{s+1}_{i,j=1}$ and unfold the expression of the matrix inner product

$$
\langle \bar{\boldsymbol{Y}}, \boldsymbol{X}_k \rangle = \sum_{r=0}^{s} \sum_{l=0}^{s} \sum_{p=0}^{T-s-1} x^{(k)}_{l+1,r+1} y_{p+r} y_{p+l}, \tag{47}
$$

where $r, l$ are the indices traversing $\bar{\boldsymbol{Y}}$.

Furthermore, each matrix element of $\mathbf{T}^{(1)}_{\boldsymbol{X}_j}$ inside the expectation has the form

$$
\frac{2}{T-s-1} \sum_{i=1}^{T-s-1} \sum_{k=1}^{s} \langle \bar{\boldsymbol{Y}}, \boldsymbol{X}_k \rangle \, y_{T-s-1+k} \, y_{T-s-1+j} \, y_{i+n} \, y_{i+m}, \tag{48}
$$

where $n, m \in \{0, 1, \ldots s\}$ are the indices traversing $\bar{\boldsymbol{Y}}$.

In what follows, we'll use the notion of a matrix's support, as follows.

**Definition D.1** (Matrix support). *Consider matrix $\boldsymbol{B} \in \mathbb{R}^{m \times n}$. The support of $\boldsymbol{B}$ is the set of index pairs corresponding to its nonzero entries,*

$$
\mathrm{supp}(\boldsymbol{B}) := \{ (r, \ell) \in [m] \times [n] \mid \boldsymbol{B}_{r\ell} \neq 0 \}. \tag{49}
$$

Assume that $j$ is fixed and odd (we discuss the even case afterwards). Note that the sparsity of each position in $\mathbf{T}^{(2)}_{\boldsymbol{X}_j}$ dictated by the parity of $(m+n-s-1+j)$ where, when $s, j$-odd, the respective element is zero whenever $m+n$ is even. Notice that except for the contribution of the matrix inner product, the sum of indices for the $y$-factors in (48) is $2(T-s-1+i)+k+j+n+m$ so the parity is determined by that of $k+j+n+m$. We distinguish two cases:

(a) when $k$ is even, $k+j$ is odd, and we wish that the term zeroes out for even $m+n$. This means that $\boldsymbol{X}_k$ must select in (47) only pairs $y_{p+r} y_{p+\ell}$ for which $r+\ell$ is even and zero-out the others. In other words, the support of $\boldsymbol{X}_k$ should satisfy

$$
\mathrm{supp}(\boldsymbol{X}_k) \subseteq \{ (r, \ell) \in [s+1] \times [s+1] \mid r+\ell \in 2\mathbb{Z} \}. \tag{50}
$$

Such an $\boldsymbol{X}_k$ may look like

$$\boldsymbol{X}_k = \begin{bmatrix} x_{11}^{(k)} & 0 & x_{13}^{(k)} & \cdots & x_{1,s}^{(k)} & 0 \\ 0 & x_{22}^{(k)} & 0 & \cdots & 0 & x_{2,s+1}^{(k)} \\ x_{31}^{(k)} & 0 & x_{33}^{(k)} & \cdots & x_{3,s}^{(k)} & 0 \\ \vdots & \vdots & \vdots & \vdots & \vdots & \vdots \\ x_{s,1}^{(k)} & 0 & x_{s,3}^{(k)} & \cdots & x_{s,s}^{(k)} & 0 \\ 0 & x_{s+1,2}^{(k)} & 0 & \cdots & 0 & x_{s+1,s+1}^{(k)} \end{bmatrix}, \tag{51}$$

with arbitrary (possibly also zero) values for constants $x_{i,j}^{(k)}$. Note that, as a consequence, the entries corresponding to odd $m+n$ are not forced to zero by the distributional symmetry of $\boldsymbol{A}$ and can generally take arbitrary values.

(b) when $k$ is odd, $k+j$ is even, and we wish that the term zeroes out for even $m+n$. This means that $\boldsymbol{X}_k$ must select in (47) only pairs $y_{p+r}y_{p+\ell}$ for which $r+\ell$ is odd and zero-out the others. In other words, the support of $\boldsymbol{X}_k$ should satisfy

$$\mathrm{supp}(\boldsymbol{X}_k) \subseteq \{(r,\ell) \in [s+1] \times [s+1] \mid r+\ell \in 2\mathbb{Z}+1\}. \tag{52}$$

Such an $\boldsymbol{X}_k$ may look like

$$\boldsymbol{X}_k = \begin{bmatrix} 0 & x_{12}^{(k)} & 0 & \cdots & 0 & x_{1,s+1}^{(k)} \\ x_{21}^{(k)} & 0 & x_{23}^{(k)} & \cdots & x_{2,s}^{(k)} & 0 \\ 0 & x_{32}^{(k)} & 0 & \cdots & 0 & x_{3,s+1}^{(k)} \\ \vdots & \vdots & \vdots & \vdots & \vdots & \vdots \\ 0 & x_{s,2}^{(k)} & 0 & \cdots & 0 & x_{s,s+1}^{(k)} \\ x_{s+1,1}^{(k)} & 0 & x_{s+1,3}^{(k)} & \cdots & x_{s+1,s}^{(k)} & 0 \end{bmatrix}, \tag{53}$$

with arbitrary (possibly also zero) values for constants $x_{i,j}^{(k)}$. Note that, as a consequence, the entries corresponding to odd $m+n$ are not forced to zero by the distributional symmetry of $\boldsymbol{A}$ and can generally take arbitrary values.

These patterns need to be coherent with the case of $j$-even. Note that in $\mathbf{T}_{\boldsymbol{X}_j}^{(2)}$, when $s$-odd, $j$-even, the respective element is zero whenever $m+n$ is odd. We again distinguish two cases:

(a) when $k$ is even, $k+j$ is even, and we wish that the term zeroes out for odd $m+n$. This means that $\boldsymbol{X}_k$ must select in (47) only pairs $y_{p+r}y_{p+\ell}$ for which $r+\ell$ is even and zero-out the others. Notice that the pattern of $\boldsymbol{X}_k$ in (51) for even $k$ satisfies this requirement and we have coherence.

(b) when $k$ is odd, $k+j$ is odd, and we wish that the term zeroes out for odd $m+n$. This means that $\boldsymbol{X}_k$ must select in (47) only pairs $y_{p+r}y_{p+\ell}$ for which $r+\ell$ is odd and zero-out the others. Notice that the pattern of $\boldsymbol{X}_k$ in (53) for odd $k$ satisfies this requirement and we have coherence.

The same approach goes through for even window size $s$. To sum up, weights satisfying equations (50) and (52) for the appropriate parity of triplet $(s,j,k)$ ensure the symmetry-induced sparsity pattern complies with that of $\mathbf{T}_{\boldsymbol{X}_j}^{(2)}$.

Finally, recall that $\boldsymbol{X}_k := \boldsymbol{b}\boldsymbol{z}_k^\top$. It can be easily shown that vectors $\boldsymbol{b}$ and $\boldsymbol{z}_k^\top$ yielding an $\boldsymbol{X}_k$ whose support has maximum cardinality are those whose support consists of indices of opposing parities (e.g., even indices for $\boldsymbol{b}$ and odd indices for $\boldsymbol{z}_k$, or vice versa) in the case of (52) or the same parities in the case of (50). For our case of odd window sizes $s$, the sparsity pattern of $\boldsymbol{b}$ and $\boldsymbol{z}_k^\top$ yielding a

complying $\boldsymbol{X}_k$ is

$$
\boldsymbol{b} = \begin{bmatrix} 0 \\ b_2 \\ \vdots \\ 0 \\ b_{s+1} \end{bmatrix} \qquad \boldsymbol{z}_k^\top = \begin{cases} \left[ 0, z_2^{(k)}, \dots, 0, z_{s+1}^{(k)} \right], \text{if } k \text{ is even} \\ \\ \left[ z_1^{(k)}, 0, \dots z_s^{(k)}, 0 \right], \text{if } k \text{ is odd} \end{cases} \tag{54}
$$

For even window size $s$, the patterns are

$$
\boldsymbol{b} = \begin{bmatrix} b_1 \\ 0 \\ b_2 \\ \vdots \\ 0 \\ b_{s+1} \end{bmatrix} \qquad \boldsymbol{z}_k^\top = \begin{cases} \left[ 0, z_2^{(k)}, \dots, 0, z_s^{(k)}, 0 \right], \text{if } k \text{ is even} \\ \\ \left[ z_1^{(k)}, 0, \dots z_{s-1}^{(k)}, 0, z_{s+1}^{(k)} \right], \text{if } k \text{ is odd} \end{cases} \tag{55}
$$

Arranging these vectors inside $\boldsymbol{W}_{QK}$ and $\boldsymbol{W}_V$ gives the stated result. $\qquad\square$

### D.4 PROOF OF THEOREM 4.1

**Theorem 4.1.** *Let $\boldsymbol{Y}_0$ encode the input tokens for $s = 1$. Then, the optimal parameters $\theta^\star = \left( \boldsymbol{W}_{QK}^\star, \boldsymbol{W}_V^\star \right)$ of a single linear self-attention layer with respect to loss $\mathcal{L}(\theta)$ are*

$$
\boldsymbol{W}_{QK}^\star = \begin{bmatrix} \frac{(T-2)\mathbb{E}\left[ y_{T-1} y_T \sum_{i=1}^{T-2} y_i y_{i+1} \right]}{\mathbb{E}\left[ y_{T-1}^2 \left( \sum_{i=1}^{T-2} y_i y_{i+1} \right)^2 \right]} & 0 \\ 0 & 0 \end{bmatrix}, \boldsymbol{W}_V^\star = \begin{bmatrix} 0 & 0 \\ 0 & 1 \end{bmatrix}, \tag{15}
$$

*up to rescaling with a nonzero constant.*

**Proof.** For the transformer parameters in (15), the corresponding $\boldsymbol{b}^\top = \begin{bmatrix} 0 & 1 \end{bmatrix}$ and the corresponding $\boldsymbol{F} = \begin{bmatrix} c & 0 \end{bmatrix}$, where $c := \frac{(T-2)\mathbb{E}_{\tilde{\boldsymbol{D}}}\left[ y_{T-1} y_T \sum_{i=1}^{T-2} y_i y_{i+1} \right]}{\mathbb{E}_{\tilde{\boldsymbol{D}}}\left[ y_{T-1}^2 \left( \sum_{i=1}^{T-2} y_i y_{i+1} \right)^2 \right]}$.

So $\boldsymbol{X} = \boldsymbol{X}_1 = \boldsymbol{b} \boldsymbol{f}_1^\top = \boldsymbol{b} \boldsymbol{F}^\top = \begin{bmatrix} 0 & 0 \\ c & 0 \end{bmatrix}$. The gradient of the in-context loss $\nabla_{\boldsymbol{X}} \mathcal{L}(\boldsymbol{X})$ is

$$
\begin{aligned}
\mathbf{T}_{\boldsymbol{X}_j}^{(1)} &= \frac{2}{T-2} \mathbb{E}_{\tilde{\boldsymbol{D}}} \left[ \langle \bar{\boldsymbol{Y}}, \boldsymbol{X} \rangle y_{T-1}^2 \bar{\boldsymbol{Y}} \right] \\
&= \frac{2}{T-2} \mathbb{E}_{\tilde{\boldsymbol{D}}} \left[ \langle \sum_{r=1}^{T-2} \begin{bmatrix} y_r^2 & y_r y_{r+1} \\ y_{r+1} y_r & y_{r+1}^2 \end{bmatrix}, \begin{bmatrix} 0 & 0 \\ c & 0 \end{bmatrix} \rangle y_{T-1}^2 \sum_{i=1}^{T-2} \begin{bmatrix} y_i^2 & y_i y_{i+1} \\ y_{i+1} y_i & y_{i+1}^2 \end{bmatrix} \right] \\
&= \frac{2}{T-2} \mathbb{E}_{\tilde{\boldsymbol{D}}} \left[ c \sum_{r=1}^{T-2} y_r y_{r+1} y_{T-1}^2 \sum_{i=1}^{T-2} \begin{bmatrix} y_i^2 & y_i y_{i+1} \\ y_{i+1} y_i & y_{i+1}^2 \end{bmatrix} \right] \\
&= \frac{2}{T-2} \mathbb{E}_{\tilde{\boldsymbol{D}}} \left[ c \sum_{r=1}^{T-2} y_r y_{r+1} y_{T-1}^2 \sum_{i=1}^{T-2} \begin{bmatrix} 0 & y_i y_{i+1} \\ y_{i+1} y_i & 0 \end{bmatrix} \right].
\end{aligned} \tag{56}
$$

According to Lemma D.3, the two diagonal elements in (56) $\mathbb{E}_{\tilde{\boldsymbol{D}}} \left[ c \sum_{r=1}^{T-2} y_r y_{r+1} y_{T-1}^2 \sum_{i=1}^{T-2} y_i^2 \right]$ and $\mathbb{E}_{\tilde{\boldsymbol{D}}} \left[ c \sum_{r=1}^{T-2} y_r y_{r+1} y_{T-1}^2 \sum_{i=1}^{T-2} y_{i+1}^2 \right]$ are 0, since the sums of $y$ indices are both odd.

$$
\begin{aligned}
\mathbf{T}_{\boldsymbol{X}_j}^{(2)} &= -2 \mathbb{E}_{\tilde{\boldsymbol{D}}} \left[ y_T y_{T-1} \sum_{i=1}^{T-2} \bar{\boldsymbol{Y}} \right] \\
&= -2 \mathbb{E}_{\tilde{\boldsymbol{D}}} \left[ y_T y_{T-1} \sum_{i=1}^{T-2} \begin{bmatrix} y_i^2 & y_i y_{i+1} \\ y_{i+1} y_i & y_{i+1}^2 \end{bmatrix} \right]
\end{aligned}
$$

$$= -2\mathbb{E}_{\tilde{D}}\left[y_T y_{T-1} \sum_{i=1}^{T-2} \begin{bmatrix} 0 & y_i y_{i+1} \\ y_{i+1} y_i & 0 \end{bmatrix}\right]. \tag{57}$$

According to Lemma D.3, the two diagonal elements in (57) $\mathbb{E}_{\tilde{D}}\left[y_T y_{T-1} \sum_{i=1}^{T-2} y_i^2\right]$ and $\mathbb{E}_{\tilde{D}}\left[y_T y_{T-1} \sum_{i=1}^{T-2} y_{i+1}^2\right]$ are 0, since the sums of $y$ indices are both odd.

Plugging in the expression of $c$, it can be easily found that

$$\nabla_{\boldsymbol{X}}\mathcal{L}(\boldsymbol{X}) = \mathbf{T}_{\boldsymbol{X}_j}^1 + \mathbf{T}_{\boldsymbol{X}_j}^2 = 0. \tag{58}$$

Since the in-context loss is convex in $\boldsymbol{X}$ and the $\boldsymbol{X}$ resulting from the $\boldsymbol{W}_V^\star$ and $\boldsymbol{W}_{QK}^\star$ above makes $\nabla_{\boldsymbol{X}}\mathcal{L}(\boldsymbol{X}) = 0$, the $\boldsymbol{W}_V^\star$ and $\boldsymbol{W}_{QK}^\star$ above is a global minimizer for the in-context loss. $\quad\square$

# E   PROOFS FOR SECTION 5

## E.1   PROOF THAT OUR EXPERIMENTS' SAMPLING SCHEMES OBEY ASSUMPTION 3.2

All our experiments use a sampling schemes whose generalization is the following:

(a) $\boldsymbol{A}$ constructed by sampling $\boldsymbol{v} \sim \mathcal{P}$, where $\mathcal{P}$ is centrally symmetric and absolutely continuous w.r.t. the Lebesgue measure on $\mathbb{R}^d$ with marginals supported on $[-1, 1]$, and independently sampling $\mathbf{P}$, whose every entry is drawn i.i.d. from any absolutely continuous distribution w.r.t. Lebesgue measure in $\mathbb{R}$. Matrix $\boldsymbol{A}$ is then formed as $\mathbf{P}\mathrm{diag}(\boldsymbol{v})\mathbf{P}^{-1}$.

(b) $\boldsymbol{c}$ is sampled from an absolutely continuous distribution w.r.t. Lebesgue measure in $\mathbb{R}^d$, or otherwise fixed with $\boldsymbol{c} \neq \mathbf{0}_d$.

We need to show that

(a) $\boldsymbol{A}$'s distribution is centrally symmetric, i.e., that $\boldsymbol{A} \stackrel{\mathrm{d}}{=} -\boldsymbol{A}$;

(b) $\boldsymbol{A}$'s spectrum is simple w.p. 1;

(c) observability still holds when $\boldsymbol{c}$ is fixed according to the above condition.

The first point is achieved since, by the central symmetry of $\boldsymbol{v}$'s distribution,

$$-\boldsymbol{A} = -\boldsymbol{P}^{-1}\mathrm{diag}(\boldsymbol{v})\boldsymbol{P} = -\boldsymbol{P}^{-1}\mathrm{diag}(-\boldsymbol{v})\boldsymbol{P} \stackrel{\mathrm{d}}{=} \boldsymbol{P}^{-1}\mathrm{diag}(\boldsymbol{v})\boldsymbol{P} = \boldsymbol{A}. \tag{59}$$

The second point is ensured by $\boldsymbol{v}$'s distribution being absolutely continuous w.r.t. the Lebesgue measure in $\mathbb{R}^d$, and hence the probability of $\boldsymbol{v}$ belonging to $(d-1)$-dimensional subspaces (and lower) such as $\{\boldsymbol{x} \in \mathbb{R}^d \mid \exists i, j \in [d] \text{ s.t. } x_i = x_j\}$ is null. In conjunction with the above, when we sample $\boldsymbol{c}$ from a continuous distribution in $\mathbb{R}^d$, Assumption (3.2) is satisfied.

However, our proofs and experiments go through even if $\boldsymbol{c}$ is fixed, as follows. First, the theoretical results rest on $\boldsymbol{A}$'s distributional symmetry and are invariant to the linear transformation induced by $\boldsymbol{c}$. Second, observability is ensured since $\det(\boldsymbol{O})$ in expression (5) is not zero w.p. 1, as follows.

We use $\det(\boldsymbol{OP}) \neq 0$ w.p. 1 $\iff \det(\boldsymbol{O}) \neq 0$ w.p. 1, since $\det(\boldsymbol{P}) \neq 0$ w.p. 1.

$$\det(\boldsymbol{OP}) \stackrel{\boldsymbol{z}:=\boldsymbol{c}^{\top}\boldsymbol{P}}{=} \det([\boldsymbol{z}; \mathrm{diag}(\boldsymbol{v})\boldsymbol{z}; \ldots \mathrm{diag}(\boldsymbol{v})^{d-1}\boldsymbol{z}]) \tag{60}$$

$$= \det(\mathrm{diag}(\boldsymbol{z}))\det\left(\begin{bmatrix} 1 & v_1 & \ldots & v_1^{d-1} \\ 1 & v_2 & \ldots & v_2^{d-1} \\ \ldots & \ldots & \ldots & \ldots \\ 1 & v_d & \ldots & v_d^{d-1} \end{bmatrix}\right). \tag{61}$$

Since $\boldsymbol{P}$'s entries are drawn i.i.d. from an absolutely continuous distribution w.r.t. Lebesgue measure in $\mathbb{R}$, it holds that $z_i \neq 0$ w.p. 1. The remaining matrix is Vandermonde with $v_i \neq v_j$, $\forall i, j \in [d]$ w.p. 1. Hence, the determinant is nonzero w.p. 1 and observability holds almost surely.

## E.2   RELATION OF TRANSFORMER FORWARD PASS WITH PCG

For convenience, we reproduce below the PCG iteration of Shewchuk et al. (1994) for minimizing an objective

$$f(\boldsymbol{w}) = \frac{1}{2}\boldsymbol{w}^{\top}\boldsymbol{A}\boldsymbol{w} + \boldsymbol{b}^{\top}\boldsymbol{w} + c$$

We compute the first two steps of the algorithm with respect to the loss (4), which can be rewritten as

$$\mathcal{L}_{AR(s)}(\boldsymbol{w}) := \frac{1}{2(T-s-1)} \sum_{t=1}^{T-s-1} (y_{t+s} - \boldsymbol{w}^{\top}\bar{\boldsymbol{y}}_t)^2 \tag{62}$$

---

**Algorithm 1** Preconditioned Conjugate Gradient

---

1: **Input:** preconditioner $\boldsymbol{H}$, $\boldsymbol{w}_0$, $\boldsymbol{r}_0 = \boldsymbol{b} - \boldsymbol{A}\boldsymbol{w}_0$, $\boldsymbol{d}_0 = \boldsymbol{H}^{-1}\boldsymbol{r}_0$, $\delta_{\text{new}} = \boldsymbol{r}_0^\top \boldsymbol{d}_0$
2: **for** $i = 0, 1, \dots$ **do**
3:    $\boldsymbol{z}_i = \boldsymbol{A}\boldsymbol{d}_i$
4:    $\alpha_i = \frac{\delta_{\text{new}}}{\boldsymbol{d}_i^\top \boldsymbol{z}_i}$
5:    $\boldsymbol{w}_{i+1} = \boldsymbol{w}_i + \alpha_i \boldsymbol{d}_i$
6:    $\boldsymbol{r}_{i+1} = \boldsymbol{r}_i - \alpha_i \boldsymbol{z}_i$
7:    $\boldsymbol{v}_{i+1} = \boldsymbol{H}^{-1}\boldsymbol{r}_{i+1}$
8:    $\delta_{\text{old}} = \delta_{\text{new}}$, $\delta_{\text{new}} = \boldsymbol{r}_{i+1}^\top \boldsymbol{v}_{i+1}$,
9:    $\beta_{i+1} = \frac{\delta_{\text{new}}}{\delta_{\text{old}}}$
10:   $\boldsymbol{d}_{i+1} = \boldsymbol{v}_{i+1} + \beta_{i+1}\boldsymbol{d}_i$
11: **end for**
12: **return** $\theta_T$

---

$$= \frac{1}{2(T-s-1)} \sum_{t=1}^{T-s-1} \boldsymbol{w}^\top \bar{\boldsymbol{y}}_t \bar{\boldsymbol{y}}_t^\top \boldsymbol{w} - 2y_{t+s}\boldsymbol{w}^\top \bar{\boldsymbol{y}}_t + y_{t+s}^2 \tag{63}$$

$$= \frac{1}{2}\boldsymbol{w}^\top \nabla^2 \mathcal{L}_{AR(s)} \boldsymbol{w} - \boldsymbol{w}^\top \nabla \mathcal{L}_{AR(s)}(0) + y_{t+s}^2 \tag{64}$$

Two iterations of Algorithm 1 starting from $\boldsymbol{w}_0 = \boldsymbol{0}$ and using $\boldsymbol{H} = \boldsymbol{P}^{-1}$ yield the following predictor.

$\boldsymbol{w}_1 = \alpha_0 \boldsymbol{d}_0 = \alpha_0 \boldsymbol{H}^{-1}\boldsymbol{r}_0$

$\boldsymbol{w}_2 = \boldsymbol{w}_1 + \alpha_1 \boldsymbol{d}_1$

$\quad = \alpha_0 \boldsymbol{d}_0 + \alpha_1 \left[ \boldsymbol{P}\boldsymbol{r}_1 + \beta_1 \boldsymbol{d}_0 \right]$

$\quad = \alpha_0 \boldsymbol{P}\nabla\mathcal{L}_{AR(s)}(\boldsymbol{0}) + \alpha_1 \left[ \boldsymbol{P}(\boldsymbol{r}_0 - \alpha_0 \boldsymbol{z}_0) + \beta_1 \boldsymbol{P}\nabla\mathcal{L}_{AR(s)}(\boldsymbol{0}) \right]$

$\quad = \alpha_0 \boldsymbol{P}\nabla\mathcal{L}_{AR(s)}(\boldsymbol{0}) + \alpha_1 \left[ \boldsymbol{P}(\nabla\mathcal{L}_{AR(s)}(0) - \alpha_0 \nabla^2\mathcal{L}_{AR(s)}\boldsymbol{d}_0) + \beta_1 \boldsymbol{P}\nabla\mathcal{L}_{AR(s)}(\boldsymbol{0}) \right]$

$\quad = \alpha_0 \boldsymbol{P}\nabla\mathcal{L}_{AR(s)}(\boldsymbol{0}) + \alpha_1 \left[ \boldsymbol{P}(\nabla\mathcal{L}_{AR(s)}(0) - \alpha_0 \nabla^2\mathcal{L}_{AR(s)}\boldsymbol{P}\nabla\mathcal{L}_{AR(s)}(0)) + \beta_1 \boldsymbol{P}\nabla\mathcal{L}_{AR(s)}(\boldsymbol{0}) \right]$

$\quad = (\alpha_0 + \alpha_1 + \alpha_1\beta_1)\boldsymbol{P}\nabla\mathcal{L}_{AR(s)}(\boldsymbol{0}) - \alpha_1\alpha_0 \boldsymbol{P}\nabla^2\mathcal{L}_{AR(s)}\boldsymbol{P}\nabla\mathcal{L}_{AR(s)}(0)$

