# OpenReview forum: "On learning linear dynamical systems in context with attention layers"
_ICLR.cc/2026/Conference — ICLR 2026 Poster_

### Official Review · Reviewer_yeiD · 2025-10-26

**Soundness:** 3
**Presentation:** 2
**Contribution:** 3
**Rating:** 4
**Confidence:** 2

**Summary:**

The paper addresses the task of learning dynamical models in-context with a linear transformer. The authors show how the transformer can be optimised to fit a linear, size-one, autoregressive model in-context. Under certain assumptions, they compute the attention parameters that guarantee one in-context step of gradient descent of a Least Squares objective.

**Strengths:**

- Learning a dynamical model in context is challenging and new.
- The state-of-the-art is well-presented, in the introduction and throughout the presentation.

**Weaknesses:**

- The performance of linear self-attention models does not match that of more expensive LLMs.
- After the reduction to improper learning, the problem resembles linear regression. The authors should comment on the specific challenges coming from the non-i.i.d. setup.
- The assumptions on the diagonality of the state transition matrix seem strong. The authors comment on the effects of relaxing it for the data-generating process. Is the assumption also crucial for Lemma and Theorem 4.1?
- The authors write *"However, a quick calculation of the forward pass reveals that weights trained to optimality with AR(s) tokens (7) for $s \geq 2$ do not implement GD in the forward pass"*. Does it mean the suggested procedure cannot be verified empirically?
- The experimental results focus on the attention parameters. Except for the *quick calculation of the forward pas* mentioned above, the authors do not provide any evidence that the learned model performs gradient descent in context, which, to my understanding, is the paper's main claim.
- The length of the sequences used in the experiments is 30. Is the same length used for in-context testing?

**Questions:**

- Why is the Gaussianity assumption necessary? Wouldn't requiring the noise to have a zero mean be enough?
- Once data matrices are formed, the problem reduces to standard linear regression. Is the extra challenge the sum over $k$ in Equation 9? Aren't $Y_t|Y_{t-1}$, $t=1, \dots, T$ i.i.d. when the model is stationary?
- What do you mean by *"temporal scaffold closer in nature to that of language-induced tokens, in stark contrast to the i.i.d."*?
- What happens if the state-transition matrix is not diagonal? Would $W_{QK}$ and $W_V$ still have the structure shown in Lemma 4.1?
- Does the idea generalise to i) multi-step GD and ii) larger sliding windows?
- In the discussion, you say that the model does not implement GD for $s\geq 2$. Does this falsify the theoretical equivalence claimed in the abstract? Is the *"recursive pattern of the optimal weights that builds on top of the GD inducing parameters recovered for AR(1)*" predicted by the theory?

---

> ### Author Response · Authors · 2025-11-23
> **Thank you for the feedback and suggestions (part 1)**
>
> We thank the reviewer for their time, thoughtful comments, and helpful suggestions. Since some concerns are shared among reviewers, we addressed them in a general comment, which we reference as needed. To help the reviewer locate the new parts in the amended manuscript (uploaded), we have marked them in blue. The line/equation references in the reply are all w.r.t the revised manuscript. When numeric references in the reply text are listed at the end of the reply.
>
> 1. **Regarding the simplified architecture.** We provide supporting arguments for our chosen architecture --- please see general response point 1.
>
> 2. **Regarding the specific challenges of the non-i.i.d. data setting.** We have already addressed the challenges in lines 296-300 and 351-355. Is there any particular point the reviewer wishes we further elaborate on?
>
> 3. **Regarding the restrictive assumptions on A and c.** We agree and have relaxed the assumptions --- please see general response points 3 and 4.
>
> 4. **Regarding the sequence length for training vs. testing.** The same length of $T=30$ was used for both training and testing. Context length generalization is a standalone line of work due to transformers' known poor performance in this regard [1, 2], and is not the focus of our paper.
>
> 5. **Regarding noise Gaussianity.** Please see general response point 4.
>
> 6. **Regarding the difficulty of characterizing the minimizers of loss (8).**
>     Finding the global minimizer of loss (8) is nontrivial, and different from standard linear regression because i) the $\mathcal{L}(\theta)$ is non-convex in $(W_{QK}, W_V)$ ii) it is highly non-linear in the inputs $y_i$ which are, moreover, correlated and hence difficult to tackle inside the expectation since they don't factorize as they do in the i.i.d case. The reformulation (9) does not eliminate these difficulties, but just ameliorates them via observing convexity w.r.t. the product reparameterization $ ba_i^\top$. Note that we require the individual values of $b$ and $a_i$, i.e., $W_V$ and $W_{QK}$, so the reparametrization does not give the final solution.
>
> 7. **Regarding stationarity and independence.** The LDS approaches stationarity as $T$ grows large, but this only means that the observations become identically distributed in the limit. In particular, if we sample $x_0$ from the stationary distribution $\mathcal{N}(0, \Sigma_{\infty})$, the process is stationary from the get-go. However, the observations do not become independent, since, for example, $ \mathrm{Cov}(y_t, y_{t+1}) = c^\top A \Sigma_\infty c\neq 0, \;\;\forall t \in \mathbb{N_+}$ (unless $A = 0$).
>
> 8. **Regarding the meaning of "temporal scaffold closer in nature to [...]".** We refer to the fact that language is a signal that unfolds over time, and therefore so are the tokens derived from it, which are processed by LLMs. For example, the typical English sentence structure is the sequence of subject-verb-object, with each element depending on the preceding ones both grammatically and semantically. These dependencies are more similar to LDS-produced observations than i.i.d. data, a fact also leveraged in prior works relying on LDSs for language modeling [3, 4, 5, 6].
>
> 9. **About generalizing to multi-step GD.** If we understand correctly, the reviewer asks whether multiple layers will translate to multiple GD steps. If so, the answer is no, not in general. For example, in the i.i.d. setting, [7] shows that multiple layers amount to a different algorithm they call GD++; complementarily, [8] show that multiple layers can perform multiple GD steps **if the weights are restricted to a certain sparsity pattern**. We expect this behaviour to extend to our case as well (i.e., multiple layers $\neq$ multiple steps of GD).
>
> 10. **Regarding generalizing to larger sliding windows.** We have improved our understanding since the submission --- please see the general response point 5.
>
> 11. **Regarding the evidence of GD steps being implemented by the optimal weights**
>     For AR(1), we recover the same weight structure as prior work [8, 9, 10] (albeit with a different value of the step size, since we use non-i.i.d. data). This structure is well-established to implement a GD step, and we omitted repeating the calculation to save space (computing the forward pass is done, e.g., in [9, pg.5], [7, Sec. 2]). We note that all prior works on LDS data use only AR(1)-type tokens in their analyses [10, 12], so this result addresses a relevant setting even without an optimality proof for AR(s), $s >= 2$.
>
>     Regarding the loss for AR(s)-type tokens, our understanding has improved our understanding since the submission --- please see the general response point 5.

---

> > ### Author Response · Authors · 2025-11-23
> > **Response part 2**
> >
> > 12. **Regarding the question: Does it mean the suggested procedure cannot be verified empirically?** We emphasize that for AR(1) the procedure is verified both empirically (see Fig. 1, a), b), c)) and theoretically (Theorem 4.1). For AR(s), $s \geq 2$, the weights recovered empirically seem to implement something more complex than GD, hence our statement "do not implement GD in the forward pass, or at least not in the formulaic manner of prior works". The recursive pattern highlighted in lines 489-491 in the old manuscript suggests an update direction that includes the GD direction as part of a linear combination. Since the submission, we've improved our understanding --- please see general response part 5.
> >
> > 13. **Regarding the question: "Is the "recursive pattern of the optimal weights that builds on top of the GD inducing parameters recovered for AR(1)" predicted by the theory?"**
> >
> >     Yes, partly. Lemma 4.1 demonstrates that the checkerboard pattern of the weights recovered by training satisfies a necessary structural condition for the critical points of the training loss. This checkerboard pattern is recursive.
> >
> > 14. **Regarding the question "Does this falsify the theoretical equivalence claimed in the abstract?"** The equivalence is not falsified. The abstract claims: "We provide [...] and show its equivalence to one step of Gradient Descent relative to an autoregression objective of window size one." --- therefore, the claim only refers to AR(1) type tokens, which is what we prove. We further say "Guided by experiments, we posit an extension to larger window size", which is what we do for AR(s), $s \geq 2$ in lines 017-018 and lines 453-461 of the old manuscript based on Fig. 1 and 2. We will improve the clarity of these statements.
> >
> > [1] (new) Anil, Cem, et al. Exploring length generalization in large language models. 2022
> >
> > [2] (new) Zhao, Liang, et al. Length extrapolation of transformers: A survey from the perspective of positional encoding. 2024
> >
> > [3] (new) Jeffrey L. Elman. Language as a dynamical system. 1995
> >
> > [4] (new) Whitney Tabor, Christopher Juliano, and Michael Tenenhaus. A dynamical system for language processing. 1996
> >
> > [5] (new) Peter Beim Graben, Bryan Jurish, Douglas Saddy, and Stefan Frisch. Language processing by dynamical systems. 2004
> >
> > [6] (new) David Belanger and Sham Kakade. A linear dynamical system model for text. 2015
> >
> > [7] (cited) Von Oswald, Johannes, et al. Transformers learn in-context by gradient descent. International Conference on Machine Learning. PMLR. 2023
> >
> > [8] (cited) Ahn, Kwangjun, et al. Transformers learn to implement preconditioned gradient descent for in-context learning. 2023
> >
> > [9] (cited) Mahankali, Arvind, Tatsunori B. Hashimoto, and Tengyu Ma. One step of gradient descent is provably the optimal in-context learner with one layer of linear self-attention. 2023
> >
> > [10] (cited) Michael E. Sander, Raja Giryes, Taiji Suzuki, Mathieu Blondel, and Gabriel Peyré. How do transformers perform in-context autoregressive learning. 2024
> >
> > [11] (cited) Frank Cole, Yulong Lu, Tianhao Zhang, and Yuxuan Zhao. In-context learning of linear dynamical systems with transformers: Error bounds and depth-separation. 2025

---

### Official Review · Reviewer_hQkJ · 2025-10-31

**Soundness:** 3
**Presentation:** 4
**Contribution:** 2
**Rating:** 4
**Confidence:** 2

**Summary:**

The paper investigates how a single linear self-attention layer can perform in-context learning (ICL) on data generated by linear dynamical systems (LDS)—a non-i.i.d. setting relevant to time-series and control problems. The authors derive theoretical results showing that the optimal linear-attention weights correspond to a gradient descent (GD) step on an autoregressive loss for first-order systems, extending prior work on ICL beyond the i.i.d. regime. They also identify structural patterns in optimal parameters for higher-order autoregressive models and support their findings with numerical experiments.

**Strengths:**

- This paper povides one of the first formal analyses of transformers’ in-context learning behavior on non-i.i.d. time-dependent data (LDS), bridging ICL theory with system identification literature.

- Experimental results confirm the theoretical findings and reveal recurring structural patterns in optimal weights for higher-order autoregressive tasks.

**Weaknesses:**

- The assumptions in this paper suggest that the non-i.i.d setting seems to be overly simplified, each sample seems to be just a sum from previous ones after some scalings. This setting leads to a significant mismatch with real-life data. Addition to this, a single linear attention layer is studied. This model is also entirely far from a transformer often used in practice. Those facts lead to concern about the significance of their result to understand transformer's expressive power.
- The optimality proof is limited to first-order systems; higher-order cases are only supported empirically without a full theoretical guarantee.

**Questions:**

N/A

---

> ### Author Response · Authors · 2025-11-23
> **Thank you for the feedback and suggestions**
>
> We thank the reviewer for their time, thoughtful comments, and helpful suggestions. Since some concerns are shared among reviewers, we addressed them in a general comment, which we reference as needed. To help the reviewer locate the new parts in the amended manuscript (uploaded), we have marked them in blue. The line/equation references are all w.r.t the revised manuscript.
>
> 1. **Regarding the restrictive assumptions on A, c and the noise.** We agree and we provide an extension to symmetric $A$ --- please see general response point 3.
>
>
> 2. **Regarding the simplistic architecture** We provide supporting arguments for our chosen architecture --- please see general response point 1
>
> 3. **Regarding the optimality proof being limited to AR(1) type tokens** We agree that this limitation is the most pressing one, and we provide further supporting arguments for the relevance of our setting. Please see the general response point 2.

---

### Official Review · Reviewer_jZgQ · 2025-10-31

**Soundness:** 3
**Presentation:** 2
**Contribution:** 3
**Rating:** 4
**Confidence:** 2

**Summary:**

The paper investigates the expressive power of a single linear attention layer for in-context learning of Linear Dynamical Systems (LDS) using non-i.i.d. data. The main result demonstrates that, for a first-order autoregressive approximation, the optimal linear attention layer effectively performs one step of gradient descent (GD) on the corresponding least-squares loss.

**Strengths:**

The main strength of the paper lies in its consideration of the challenging non-i.i.d. setting and its rigorous theoretical contribution. The authors formally prove that, for an order-one autoregressive approximation of a Linear Dynamical System, the optimal single linear attention layer corresponds to one step of gradient descent on the associated least-squares loss. Overall, the theoretical analysis is well-motivated and provides an interesting and valuable contribution to the understanding of the expressive power of linear attention mechanisms.

**Weaknesses:**

One of the main weaknesses of the paper lies in its presentation, which is at times convoluted and difficult to follow. Furthermore, since the paper’s primary contribution is theoretical, placing the key proofs in the appendix somewhat weakens its overall clarity and impact. Regarding the theoretical results, an additional limitation that reduces the paper’s significance is that the optimality result (Theorem 4.1), which demonstrates that the transformer performs one step of Gradient Descent on the least-squares loss, is restricted to the order-one autoregressive approximation of the Linear Dynamical System. Another substantial limitation lies in Assumption 3.2, which imposes a highly simplified structure on the linear dynamical system by restricting the matrix A to be diagonal with distinct, nonzero entries. While this choice facilitates analytical tractability, it eliminates interactions between state variables and therefore limits the applicability of the results to realistic systems exhibiting coupled dynamics. Moreover, the assumption that both the process and observation noise are Gaussian and isotropic further simplifies the analysis but fails to capture the correlated or non-Gaussian noise commonly encountered in real-world scenarios. The authors should also consider citing and discussing related recent work, such as Zhang, Yedi, et al. “Training dynamics of in-context learning in linear attention.” arXiv preprint arXiv:2501.16265 (2025), which similarly analyzes the expressive power and mechanistic basis of ICL within minimal transformer architectures.
Regarding the empirical study, the rationale behind the proposed experimental setup is not entirely convincing. It remains unclear how several hyperparameters (such as T, d, and the number of iterations) were selected and how their choice may have influenced the results. For instance, the noise margin, which appears to play a crucial role, is fixed, yet no explanation is provided for how its value was determined. The authors should clarify the sensitivity of the analysis and the observed recursive patterns to these parameter choices.

**Questions:**

- The optimality result in Theorem 4.1 is restricted to the order-one autoregressive approximation of the Linear Dynamical System. Do the authors expect their findings to extend to higher-order cases, and if so, under what assumptions?

- How sensitive are your results to relaxing Assumption 3.2, for example in systems with coupled (non-diagonal) dynamics or correlated, non-Gaussian noise?

- Could the authors elaborate on how the key hyperparameters were selected and discuss how sensitive the empirical results are to changes in these parameters?

---

> ### Author Response · Authors · 2025-11-23
> **Thank you for the feedback and suggestions**
>
> We thank the reviewer for their time, thoughtful comments, and helpful suggestions. Since some concerns are shared among reviewers, we addressed them in a general comment, which we reference as needed. To help the reviewer locate the new parts in the amended manuscript (uploaded), we have marked them in blue. The line/equation references in the reply are all w.r.t the revised manuscript. When numeric references in the reply text are listed at the end of the reply.
>
> 1.**Regarding proof placement in the appendix.** We appreciate the reviewer’s feedback. Our goal in the main text is to provide a clear picture of the broad technical approach and its implications --- we give a high-level outline of the proof in Section 4. Deferring full proofs to the appendix is standard in theoretical papers (see, e.g., the closely related works [3, 5, 6]). Nevertheless, we're happy to move the specific proof parts that the reviewer thinks would improve accessibility into the main text.
>
> 2.**Regarding the significance of our theoretical results.** We agree that this limitation is the most pressing one, and we provide further supporting arguments for the relevance of our setting. Please see general response point 2.
>
> 3.**Regarding the restrictive assumptions on A and c.** We agree, and provide an extension to symmetric $A$ --- please see general response point 3.
>
> 4.**Regarding noise Gaussianity/isotropy.** We relax the isotropy condition, but not the Gaussianity one -- please see general response point 4. In support of the model's applicability, we give several new references relying on Gaussian noise LDSs to model language [12, 13, 14, 15].
>
> 5.**Regarding the additional reference.** We thank the reviewer for the reference. The scope of this work differs from ours, as it studies the dynamics of linear attention training under Gradient Flow (whereas we are concerned solely with characterizing the global minima) and targets the i.i.d. data setting (whereas we study the more challenging setting of LDS-produced data). We can still include it along with the other papers on the i.i.d. setting.
>
> 6.**Regarding parameter choices in the experiments.** The parameter choice was guided by prior work, computational resources, and observed performance. Noise magnitudes were chosen of the same order as in the empirical work [7, Sec. III, A], where transformer ICL was first observed to compete with the KF. We emphasize that the theoretical results in Lemma 4.1 and Thm. 4.1 hold for any noise magnitudes, and we therefore expect the experiments to recover the theory --- of course, subject to tuning. We also expect that convergence becomes worse when observation noise ($v_t$) overpowers the signal magnitude ($x_t$) --- this also holds for e.g., the KF, whose convergence is slowed down in this circumstance. Regarding our choice of $T=30$ the value aligns with other works in this area (e.g., [3, p. 7] uses $T=20$, and [5, p. 8] uses $T=50$), and the same holds for $d=5$ (e.g., [10] uses $d\in \{1, 2\}$, and [11] uses $d\in\{2, 10\}$). The batchsize and number of iterations were set as follows: we start with small values for both and increase them until the training loss and the pattern of $W_{QK}$ and $W_V$ converge. Our final values are in the same ballpark as other works on ICL with i.i.d. data --- e.g., [3, pg. 7] run a total of $30000$ iterations and sample a fresh batch of $20000$ sequences every $100$ iterations. When we increase the batch size, we also increase the maximal learning rate as per Appendix B.1 Table 1 and  2, and decrease the $\beta_1$ and $\beta_2$ of AdamW (rule of thumb from [8] and [9]). Please see the revised manuscript for additional experiments on more complex empirical settings.
>
> 7.**Regarding the significance of the result in Thm. 4.1** We expect an optimality result similar to the one for AR(1) in Thm 4.1 to hold for AR(s) $s \geq  2$ under no further assumptions; however, we don't expect to get it using the same proof approach as for AR(1) since the algebra becomes intractable. We note that all prior works on LDS data use only AR(1)-type tokens in their analyses [7, 8], so our result still addresses a relevant setting. Regarding whether the GD interpretation still holds for AR(s) $s \geq  2$, our understanding has improved since the submission --- please see general response part 5.
>
> 8.**Regarding the extension of our results to non-diagonal A, non-isotropic noise and non-Gaussian noise** Please see general response pts. 3 and 4, and a new series of experiments in the revised manuscript.

---

> > ### Author Response · Authors · 2025-11-23
> > **References for the above reply**
> >
> > [1] (cited) Von Oswald, J., et al. Transformers learn in-context by gradient descent. 2023
> >
> > [2] (cited) Zhang, R. et al. Trained transformers learn linear models in-context. 2024
> >
> > [3] (cited) Ahn, K., et al. Transformers learn to implement preconditioned gradient descent for in-context learning. 2023
> >
> > [4] (cited) Mahankali, A. et al. One step of gradient descent is provably the optimal in-context learner with one layer of linear self-attention. 2023
> >
> > [5] (cited) M. E Sander, et al. How do transformers perform in-context autoregressive learning?
> >
> > [6] (cited) Cole, F., et al. In-context learning of linear dynamical systems with transformers: Error bounds and depth-separation. 2025
> >
> > [7] (cited) Balim, H, et al. Can transformers learn optimal filtering for unknown systems?. 2023
> >
> > [8] (new) Wang, Xi, and Laurence Aitchison. Batch size invariant Adam.2024
> >
> > [9] (new) Granziol, D. et al. Learning rates as a function of batch size: A random matrix theory approach to neural network training. 2022
> >
> > [10] (cited) M. Kozdoba, et al. On-line learning of linear dynamical systems: Exponential forgetting in kalman filters. 2019
> >
> > [11] (new) Hazan, E. et al. Learning linear dynamical systems via spectral filtering. 2017
> >
> > [12] (new) J. L Elman. Language as a dynamical system. 1995
> >
> > [13] (new) W. Tabor, et al. A dynamical system for language processing. 1996
> >
> > [14] (new) P. Graben, et al. Language processing by dynamical systems. 2004
> >
> > [15] (new) D. Belanger et al. A linear dynamical system model for text. 2015

---

> ### Author Response · Authors · 2025-12-01
> **Extension to the correlated noise**
>
> Regarding the correlated noise, we generalize our theory to $\Sigma_w$, the covariance of the transition noise as a non-diagonal matrix whose eigenvalues are different. Lemma D.3 is added to make our proof (Lemma D.1 and Lemma D.2) hold for such $\Sigma_w$. The experiment under this setting are included in Appendix B.5, Fig. 5 of the revision, which aligns with our theory.

---

### Official Review · Reviewer_2rfD · 2025-11-01

**Soundness:** 3
**Presentation:** 3
**Contribution:** 3
**Rating:** 6
**Confidence:** 4

**Summary:**

This paper investigates the theoretical behavior of linear attention layers when provided sequences generated by linear dynamical systems (LDS), aiming to understand how transformers can perform in-context learning under temporally correlated data. It was shown that for first-order autoregressive processes, the optimal weights of a linear attention layer perform a single gradient descent step on the least-squares estimation objective, essentially mirroring Kalman filtering updates. It is further shown which structured pattern the learned weights exhibit for higher-order AR models. Experiments on synthetic LDS data support these claims, suggesting that transformers can emulate filtering-like inference dynamics without explicit supervision.

**Strengths:**

The paper presents a clear theoretical analysis linking linear attention updates to gradient descent steps in linear dynamical systems. The extension of the understanding of in-context learning ICL beyond i.i.d. data to temporally correlated, non-i.i.d. sequences is conceptually
significant.

**Weaknesses:**

The scope is somewhat narrow: the main result holds only for AR(1) processes, with higher-order cases supported empirically rather than analytically, and the experimental validation remains limited to small synthetic settings.

Assumption 3.2 (lines 207-208) appears rather limiting -- diagonal A, combined with the assumptions on the noise covariance matrix stated as part of Assumption 3.1, implies that the dynamics of the components of the state vector in (1) are completely separable. The assumption c = 1_d (part of Assumption 3.2) is also limiting.

**Questions:**

It is stated in the paper that Assumption 3.2 can be "relaxed to symmetry at the expense of added complexity in the later data sampling step." This should be elaborated upon: what would be the path forward for more general A and c?

The core analytical result is for AR(1) / first-order LDS. Can you clarify precisely which parts of the argument fail for AR(p) with p > 1? Is it a technical difficulty (algebra gets intractable) or a fundamental obstacle (the correspondence to a single GD step is no longer true)? This matters for how general the claimed mechanism actually is.

Is the linear attention update / Kalman-style filtering analogy exact (i.e., the attention layer implements the same update under the assumptions), or qualitative (as in "both reduce prediction error using recent context")? It would help if this is stated explicitly, i.e., if  exact equivalence under stated assumptions is precisely established... otherwise, one might get an impression that attention simply learns to mimic the effect of an adaptive linear predictor under LDS-generated data. Either way, precision would be beneficial.

The Discussion section makes several statements about prior work that appear overstated. For instance, lines 358–359 claim this is the only work addressing noisy LDSs apart from Cole et al. '25, yet Goel and Bartlett '24 (already cited) analyze a similar and more general LDS model without assuming Assumption 3.2. Likewise, the statement in lines 368–371 that prior KF-emulating transformers require complete knowledge of system parameters seems inconsistent with published claims of robustness to partial parameter availability in that approach.

While the focus of the paper is on LDS + Gaussian inputs, an insight into what happens if the noise is not Gaussian but the system is still linear would be appreciated. Likewise, even a brief discussion on potential directions and/or expectation in non-linear + non-Gaussian case would be helpful.

---

> ### Author Response · Authors · 2025-11-23
> **Thank you for the feedback and suggestions**
>
> We thank the reviewer for their time, thoughtful comments, and helpful suggestions. Since some concerns are shared among reviewers, we addressed them in a general comment, which we reference as needed. To help the reviewer locate the new parts in the amended manuscript (uploaded), we have marked them in blue. The line/equation references in the reply are all w.r.t the revised manuscript. When numeric references in the reply text are listed at the end of the reply.
>
> 1. **Regarding the limitation of theoretical results to AR(1)-tokens.** We agree that this limitation is the most pressing one, and we provide supporting arguments for the relevance of our setting in general response point 2.
>
> 2. **Regarding the restrictive assumptions on A and c.** We agree, and we provide an extension to symmetric $A$ --- see general response point 3.
>
> 3. **Regarding the obstacle in proving optimality for AR(s), $s \geq 2$.** We state the ideal results (below) and highlight the parts that fail for AR(s) $s \geq 2$.
>
>     1. We observe a checkerboard pattern in the trained weights
>
>     2. We prove that this checkerboard pattern satisfies a necessary condition on the structure of critical points of loss (8) (Lemma 4.1)
>
>     3. We prove that the trained weights approximate a global minimum of loss (8)
>
>     4. We interpret the meaning of the trained weights by the algorithm they implement (if any)
>
>     For AR(1) points 1, 2, and 3 hold, and in point 4 we observe the weights implement a GD step, recovering the result of [6]. For general AR(s), points 1 and 2 hold, but proving point 3 with the approach used for AR(1) leads to intractable algebra (large systems of eqs.). For point 4, our understanding has improved since the submission --- see general response part 5.
>
> 4. **Regarding the analogy with the Kalman Filter (KF).**
> 	The analogy is not exact because: i) transformers are model-free learners and the KF is model-based; and ii) we operate on a truncation described in lines 185-187, which precludes an exact analogy (typical of improper learning approaches --- see lines 173-201). We emphasize that improper learning formulations don't aim to mirror KF, but to achieve provably competitive accuracy relative to it under the same data distribution/objective. E.g., [5] shows that online GD w.r.t. loss (4) gives predictors that are competitive with KF for sufficiently large $s$. Similarly, our goal is to provide a plausible hypothesis for why transformers are empirically observed [4] to be competitive with KF.
>
> 5. **Regarding overstatements on prior work.** We respectfully disagree with the reviewer on this point. Goel and Bartlett's [1] setting is fundamentally different from ours. Their Theorem 2 shows that there exists one layer of softmax attention that can approximate the KF corresponding to a  **given**, **fixed** LDS. Notably, the transformer's $\beta$ depends on this LDS via $\theta$ and $\kappa$. They do **not** study in-context learning, and their statement is a representation power result, whereas our study concerns expressive power --- i.e., after training to convergence on sequences of observations from distinct LDSs, a transformer can predict the next observation for a sequence of a newly sampled LDS.
>
>     Regarding Akram and Vikalo's work [2] (lines 372-375), their tokens contain at least a part of the LDS parameters **at all times**. In the ideal setting (their Eq. 20), $A$, $c^\top$, $\Sigma_w$, and $\sigma_v$ are included in the tokens. The robustness claim on page 9 (bottom) refers only to withholding $\Sigma_w$ and $\sigma_v$, and not $A$ and $c^\top$. Finally, their claim of withholding $A$, $\Sigma_w$, and $\sigma_v$ does **not** match their description in Appendix C, where the tokens still contain $\Sigma_w$ and $\sigma_v$. Finally, theirs is an existence result for an architecture implementing KF, but it does not show that training actually recovers this architecture. Our claim in lines 373-375, therefore, remains valid.
>
> 6. **Regarding non-Gaussian noise.** This was a general concern, and is addressed in the general response in pt. 4
>
> [1] (cited) G. Goel et al. Can a transformer represent a Kalman filter? 2024.
>
> [2] (cited) U. Akram et al. Can transformers in-context learn behavior of a linear dynamical system? 2024.
>
> [3] (cited) Akyürek, E., et al. What learning algorithm is in-context learning? Investigations with linear models. 2022.
>
> [4] (cited) Zhe Du et al. Can transformers learn optimal filtering for unknown systems? 2023.
>
> [5] (cited) M. Kozdoba, et al. On-line learning of linear dynamical systems: Exponential forgetting in Kalman filters. 2019
>
> [6] (cited) Ahn, K., et al. Transformers learn to implement preconditioned gradient descent for in-context learning. 2023
>
> [7] (cited) F.Cole, et al. In-context learning of linear dynamical systems with transformers: Error bounds and depth-separation. 2025.
>
> [8] (cited) M. Sander et al. How do transformers perform in-context autoregressive learning? 2024.

---

### Official Review · Reviewer_JMPY · 2025-11-10

**Soundness:** 3
**Presentation:** 4
**Contribution:** 3
**Rating:** 8
**Confidence:** 2

**Summary:**

This paper investigates in-context learning (ICL) for non iid sequential data, addressing a critical gap in the theoretical understanding of transformers. This paper takes the first step toward sequential data by studying Linear Dynamical Systems (LDS). The authors prove that for order-one autoregressive (AR(1)) approximations, the optimal linear attention layer implements exactly one step of Gradient Descent on the context-dependent least-squares loss. The key insight is that AR(s) models can closely approximate Kalman Filter predictions when the window size s is sufficiently large. The authors identify structural patterns for higher-order cases and provide experimental validation. The connection to language modeling is still distant, but this result represents the first optimality result for ICL with non-i.i.d. data extending the optimization framework from independence to temporal dependence.

**Strengths:**

* The paper is well written, well organized.I am far from being a theoretical specialist in the field, but the problem is very clearly understandable, as are the SOTA and the paper's positioning in the literature.
* Extends ICL theory to noise-corrupted dynamical systems with non iid data, more realistic scenario.
*  Provides Mechanistic Explanation for Transformer-Kalman Filter connection
* Solid theoretical contribution for AR(1) with Noise (theorem 4.1)
* structural insight for higher orders (banded structures etc).
* limitations are well discussed.

**Weaknesses:**

* in my opinion, some assumptions are highly restrictive (diagonal A for LDS, independance of the evolution of each dimension), with no clear insights how to generalize beyond the assumption.
* The  proof is only for AR(1), the results for s>=1 are only empirical.

**Questions:**

You state  that AR(s≥2) optimal weights do not
implement gradient descent. However, you observe an intriguing
recursive pattern  where AR(s+1) structure embeds
forward-shifted AR(s) structure. Can you clarify your current undestanding ?

---

> ### Author Response · Authors · 2025-11-23
> **Thank you for the feedback and suggestions**
>
> We thank the reviewer for their time, thoughtful comments, and helpful suggestions. Since a few of the concerns are shared between reviewers, we addressed them in the "General response" above, which we reference as needed. To help the reviewer locate the new parts in the amended manuscript (uploaded), we have marked them in blue. The line/equation references are all w.r.t the revised manuscript.
>
> 1.**Regarding the restrictive assumptions on A and c.** We agree and we provide an extension to symmetric $A$ --- please see general response point 3.
>
> 2.**Regarding the limitation of theoretical results to AR(1)-tokens.** We agree that this limitation is the most pressing one, and we provide some further supporting arguments for the relevance of our setting. Please see general response point 2.
>
> 3.**Regarding the interpretation of weight structures for AR(s), $s \geq 2$** Please see general response point 5.

---

### Author Response · Authors · 2025-11-23
**General response addressing common concerns (part 1)**

We thank the reviewers for their time, their thoughtful comments, and the helpful suggestions for improving our paper. We address a few of the shared concerns here and respond to the rest within the individual reply sections. All equation/line references are w.r.t. the revised manuscript (uploaded). The revised manuscript includes the modifications described in our comments, and they are marked in blue. All citations offered as part of the comments are marked with (cited), if they are already included in the paper, or (new) if they have been added during the rebuttal. The references for each reply are found at the end of the comment box.


1. **About using a single linear attention layer.** This type of simplified architecture is a mainstay in **theoretical** studies aimed at understanding transformers, as illustrated by prior works in this area [1, 2, 3, 4, 5]. This setting enables the precise characterization of optimal parameters (our goal), which are otherwise mathematically intractable for multiple layers without additional assumptions. Specifically, works studying multiple layers impose weight constraints such as precise sparsity patterns, weight symmetry, etc., see [1]; or disregard optimality altogether and instead construct architectures explicitly implementing algorithmic steps (e.g., Richardson iterations) that are not guaranteed to be minimizers of the training loss [7]; or operate in the regime of infinite architecture depth/context length [8]. Finally, the single layer of linear attention was shown to preserve significant landscape properties of deeper networks [6], thus serving as a valid proxy for more complex architectures.

2. **About the optimality proof being limited to AR(1)-type constructions.** We agree with the reviewers that this is the most pressing limitation of our paper. We emphasize a couple of points in support of our results' significance w.r.t. existing work: i) all prior works on non-i.i.d. data use strictly AR(1)-type tokens in their analyses, but for simpler data-generating processes than ours (e.g. noiseless or fully-observed state processes) [7, 8]. ; ii) we'd like to re-emphasize the result of Lemma 4.1, which proves that the structure of the trained weights (Fig.1 and Fig. 2, appendix) satisfies a necessary condition required by critical points of loss (eq. (8)) for general AR(s) cases. Taken together, Lemma 4.1 and the experiments significantly narrow down the space of parameters to be considered in global optimality proofs for AR(s)-type tokens, $s\geq2$. We provide new experiments in Appendix B.3, B.4, B.5, B.6 of the revised manuscript, which show that the non-zero support of the weights converges to that prescribed by Lemma 4.1, and that results extend to different choices of $\mathbf{A}$.

 3. **Regarding diagonal $A$ and fixed $c$.** The assumption of diagonal $A$ and fixed $c = 1_d$ was made for convenience, but the constraints were indeed unnecessarily tight. We relax to a symmetric $A$ and an arbitrary $c$ with non-zero entries. We believe the requirement on $A$ can be further relaxed, but we need to iron out the mathematical details. We will follow up with a comment in the next days. Our current requirements are:

    * $\rho(A) \leq 1$ with probability (w.p.) 1

    * $A^\top = A$ and $\mathbb{E}[A^r] = 0$ for all $r = 2k+1$ with $k \in \mathbb{N}_+$.

    * $A$ and $c$ are sampled independently of each other, the noises $\{w_t\}$, $\{v_t\}$ and $x_0$

    * the pair $(c, A)$ is observable w.p. 1, i.e., $\mathcal{O} \coloneqq [c; Ac; A^2 c; \ldots A^{d-1}c]^\top$ has rank $d$

    We sample $A$ as follows: sample $Q \sim \mathrm{Haar}(O(d))$ independently of $D = \mathrm{diag}(v)$ where $v \sim \mathcal{P}$ with $\mathcal{P}$ being any centrally symmetric distribution (e.g., uniform on sphere, uniform on $[-1, 1]^d$, etc). Furthermore, $c$ is sampled from an arbitrary, continuous distribution in $\mathbb{R}^d$. The first 3 points hold directly. We show the fourth by proving that  $\mathrm{det}(\mathcal{O}Q) \neq 0 \text{w.p. 1} \iff \mathrm{det}(\mathcal{O}) \neq 0 $ w.p. 1 since $\mathrm{det}(Q) \neq 0$ deterministically. Using the same notation for the decomposition of $A$, and letting $D_{ii}$ be the diagonal elements of $D$
    \begin{aligned}
    \det(\mathcal{O}Q)
    &\stackrel{z \coloneqq c^\top Q}{=}
    \det([z, Dz, \ldots, D^{d-1}z]) \\\\
    &= \det(\mathrm{diag}(z))\,
    \det\left(\begin{bmatrix}
    1 & D_{11} & \ldots & D_{11}^{d-1} \\\\
    1 & D_{22} & \ldots & D_{22}^{d-1} \\\\
    \vdots & \vdots & \ddots & \vdots \\\\
    1 & D_{dd} & \ldots & D_{dd}^{d-1}
    \end{bmatrix}\right).
    \end{aligned}
    Since $z_i \neq 0$ w.p. 1 and the other matrix is Vandermonde with $ D_{ii} \neq D_{jj}$ w.p. 1, the determinant is nonzero w.p. 1. Experiments under this new setting are included in Appendix B.4, Fig. 4; Appendix B.5, Fig. 5 of the revision.

---

### Author Response · Authors · 2025-11-23
**General response addressing common concerns (part 2)**

4. **On the noise being non-isotropic / non-Gaussian / correlated across time.**
    Our results extend without change to the noise being non-isotropic and correlated across dimensions, i.e., $w \sim \mathcal{N}(0, \Sigma_w)$ for some matrix $\Sigma_w \succ 0$. We amended the proof and main text to account for this.

    * **Regarding Gaussian vs. just zero-mean noise**: our proofs for Lemma 4.1 and Thm. 4.1 go through with only zero-mean noise. However, the results approximating filtering with solving a linear regression problem on successive windows $[y_i, \ldots y_{i+s}]$ rely on Gaussianity [10, 11] (discussion in lines 183 - 192). To our knowledge, equivalent results to [10, 11] for non-Gaussian noise are missing. Without this condition, our results don't imply anything about transformers' ability to learn the LDS, since the link to methods like the KF is absent.

    * **Regarding noise correlated **across time** (e.g., $w_{t+k}$ dependent on $w_t$)**: The same justification as above applies. Furthermore, it's likely that the proofs and results will change non-trivially.

5. **Interpretation of trained weights for AR(s)-type tokens, $s \geq 2$.**
    Our understanding at the time of the submission was that the trained weights implement a more sophisticated step than GD which contains the GD update direction in a linear combination --- this is visible in the recursive pattern in lines 489--491 of the old manuscript. We've since updated our understanding and remark that a step approximating Preconditioned Conjugate Gradients (PCG)[9] is taken.

    Due to $\rho(A) < 1$ w.p. 1, the process approaches stationarity exponentially fast, meaning that autocorrelations will be (almost) solely dependent on the lag $k$ and not on time $t$, i.e. $\mathbb{E}[y_t y_{t+k}] \approx \gamma(k), \; \forall t$. In particular, it holds approximately that $\frac{1}{T-s-1} \sum_{i=1}^{T-s-1}y_i y_{i+k} \approx \frac{1}{T-s-1} \sum_{i=1}^{T-s-1}y_{i+p} y_{i+p + k} \approx \hat{\gamma}(k)$. Therefore, we can approximate
    \\[\frac{1}{T-s-1} Y\_0 Y\_0^{\top} \approx
    \\begin{bmatrix}
    \\hat{\\gamma}(0) & \\hat{\\gamma}(1) & \\cdots & \\hat{\\gamma}(s) \\\\
    \\hat{\\gamma}(1) & \\hat{\\gamma}(0) & \\cdots & \\hat{\\gamma}(s-1) \\\\
    \\vdots & \\vdots & \\ddots & \\vdots \\\\
    \\hat{\\gamma}(s) & \\hat{\\gamma}(s-1) & \\cdots & \\hat{\\gamma}(0)
    \\end{bmatrix} =
    \\begin{bmatrix}
    \\nabla^{2}\\mathcal{L}\_{\\text{AR}(s)} &
    \\nabla\\mathcal{L}\_{\\text{AR}(s)}(0) \\\\[6pt]
    \\big(\\nabla\\mathcal{L}\_{\\text{AR}(s)}(0)\\big)^{\\!\\top} &
    \\hat{\\gamma}(0)
    \\end{bmatrix}.
    \\]
     where $\nabla^{2}\mathcal{L}_{\text{AR(s)}}$ is the Hessian (constant) and $\nabla\mathcal{L}\_{\text{AR(s)}}(0)$ is the gradient at $w= 0$. Using the parameter structure from Lemma 4.1/the experiments, the transformer's forward pass resembles two steps of the PCG method.  Let $s = 2k$ and $N =  \frac{(s+1)^2+1}{2}$ (the case $s=2k+1$ is similar). Then, the weight matrices are
    \\[
      {W\_{QK}} =
    \\begin{bmatrix}
    c\_1      & 0        & c\_2            & \\cdots & 0        & | & c\_{k+1} \\\\
    0        & c\_{k+2}  & 0           & \\cdots     & c\_{2k+1} & | & 0       \\\\
    \\vdots   & \\vdots   & \\ddots      & \\ddots & \\vdots   & | & \\vdots  \\\\
    0        & c\_{N-2k} & 0           & \\cdots      & c\_{N-k-1} & | & 0      \\\\
    --        & -- & --          & --          & --       & -- & --      \\\\
    0  & 0        & 0   & 0         & 0        & | & 0
    \\end{bmatrix}
    \\;\\;
    {W\_{V}} =
    \\begin{bmatrix}
    0      & \\cdots & 0      & | & c\_{N-k}     \\\\
    0      & \\cdots & 0      & | & 0           \\\\
    \\vdots &        & \\vdots & | & \\vdots      \\\\
    0      & \\cdots & 0      & | & 0           \\\\
    --      & \\cdots & --     &--& --           \\\\
    0      & \\cdots & 0      & |& c\_{N}
    \\end{bmatrix}.
    \\]
    Renaming the top left $s\times s$ block of $W\_{QK}$ as $P$, the top-right $s \times 1$ block as $p$, and the top right $s \times 1$ block of $W\_{V}$ as $q$, the transformer-induced linear predictor $\frac{1}{T-s-1} W\_{QK} Y\_0 Y\_0^\top{W\_{V}}\_{:, s+1}$  is
    \\[
    {P}\\nabla^{2}\\mathcal{L}\_{\\text{AR}(s)}{q}
    \;+\\;
    ({p}{q}^{\\top} + c\_N {P})\\nabla\\mathcal{L}\_{\\text{AR}(s)}({0})
    \;+\\;
    c\_N \\hat{\\gamma}\_0 {p}
    \\]
    Letting $ {P}^{\prime} \coloneqq \Gamma \nabla^{2}\mathcal{L}\_{AR(s)}$ with $\Gamma \coloneqq \frac{c\_N\hat{\gamma}\_0 {p}{q}^\top}{{q}^\top\nabla^{2}\mathcal{L}\_{AR(s)}^2{q}}$ and observing that $c\_N \hat{\gamma}\_0{p} = {P^{\prime}}\nabla^{2}\mathcal{L}\_{AR(s)}{q}$ (see Appendix E.2), the transformer-induced predictor finally rewrites as
    \\[
    \\left(\\Gamma \\nabla^{2}\\mathcal{L}\_{\\text{AR}(s)} + {P}\\right)
    \\, \\nabla^{2}\\mathcal{L}\_{\\text{AR}(s)}{q}
    \\;+\\;
    ({p}{q}^{\\top} + c\_N{P})
    \\, \\nabla\\mathcal{L}\_{\\text{AR}(s)}({0}).
    \\]

---

### Author Response · Authors · 2025-11-23
**General response addressing common concerns (part 3 -- final)**

This predictor resembles the predictor obtained after two PCG steps [9, p. 51] on loss $\mathcal{L}\_{AR(s)}$ with preconditioner ${P}^{-1}$ starting from ${w}\_0 = 0$ and initial conjugate direction ${d}\_0 = {q}$ (algorithm deferred to Appendix E.2). Note that ${P}$'s invertibility is assumed. The resulting predictor  is

\\[
{w}\_2 =
\\left(\\tau\_1 \\nabla^{2}\\mathcal{L}\_{\\text{AR}(s)}^{-1} - \\tau\_2 {P}\\right) \\nabla^{2}\\mathcal{L}\_{\\text{AR}(s)}{q} + \\tau\_3 {P}\\nabla\\mathcal{L}\_{\\text{AR}(s)}({0})
\\]

\\[
\\approx
\\left(2\\tau\_1 {I} - \\tau\_1 \\nabla^{2}\\mathcal{L}\_{\\text{AR}(s)} - \\tau\_2 {P}\\right)
\\nabla^{2}\\mathcal{L}\_{\\text{AR}(s)}{q} + \\tau\_3 {P}\\nabla\\mathcal{L}\_{\\text{AR}(s)}({0})
\\]

where $\tau\_1, \tau\_2, \tau\_3 \in \mathbb{R}$ are iteration-dependent constants (see Appendix E.2), and we used an order-one Neumann series approximation of the Hessian inverse. The latter was shown to exist with high probability for sufficiently large $T$ by [10]. Notably, this AR(s) analogy is congruent with the GD step observed for AR(1), since PCG collapses to GD when covariates belong to $\mathbb{R}$. What remains unexplained is the conjugate direction initialization $d\_0 = q$, as opposed to the typical initialization [9] $d\_0 = P\nabla\mathcal{L}\_{AR(s)}(0)$ for which we do not currently have an interpretation.

[1] (cited) Ahn, K., et al. Transformers learn to implement preconditioned gradient descent for in-context learning. 2023.

[2] (cited) Mahankali, A., et al. One step of gradient descent is provably the optimal in-context learner with one layer of linear self-attention. 2023.

[3] (cited) R. Zhang, et al. Trained transformers learn linear models in-context. 2024.

[4] (cited) Von Oswald, J., et al. Transformers learn in-context by gradient descent. 2023.

[5] (cited) M. E. Sander, et al. How do transformers perform in-context autoregressive learning? 2024.

[6] (new) Ahn, K., et al. Linear attention is (maybe) all you need (to understand transformer optimization). 2023.

[7] (cited) F. Cole, et. al. In-context learning of linear dynamical systems with transformers: Error bounds and depth-separation. 2025.

[8] (cited) Sander, M., and G. Peyré. Towards understanding the universality of transformers for next-token prediction. 2024.

[9] (new) Shewchuk, J. An introduction to the conjugate gradient method without the agonizing pain. (1994)

[10] (cited) Tsiamis, A. and G Pappas. Finite sample analysis of stochastic system identification. 2019.

[11] (cited) M. Kozdoba, et al. On-line learning of linear dynamical systems: Exponential forgetting in kalman filters, 2019.

---

### Author Response · Authors · 2025-12-03
**Rebuttal summary**

Dear reviewers, ACs,

We wrap up our rebuttal with this final comment, summarizing the changes/additions we've made over the past two weeks. All changes detailed here are marked in the last paper version, in blue text.

1. We've strongly relaxed our assumption on the LDS parameters $A$ and $c$ (Assumption 3.2). This relaxation is part of the latest revision, which complements this reply. $A$ was originally a diagonal matrix in $\mathbb{R}^{d\times d}$, with the diagonal sampled from a centrally symmetric distribution with marginals supported on $[-1, 1]$ and absolutely continuous w.r.t. the Lebesgue measure over $R^d$. We've relaxed the assumption on $A$'s distribution to any centrally symmetric distribution supported on $S = \\{M \in \mathbb{R}^{d\times d} \\, | \\, \rho(M) \leq 1\\}$ that can ensure with probability $1$ that $A$ has a simple spectrum and is therefore (block) diagonalizable (this assumption covers our prior sampling schemes and much more). Notably, the set of stable matrices with simple spectrum is dense in the above $S$, which speaks to the generality of our assumption. This relaxation is possible since Lemma 4.1 and Thm. 4.1 only require the central symmetry of $A$'s distribution, while the rest of the assumption is used to ensure $(A, c)$ observability. Observability is used only to ensure that predicting the next observation $y_T$ via minimizing $\mathcal{L}_{\text{AR}(s)}$ performs closely to the optimal Kalman Filter. Other than this, observability is not directly required by our proofs.


   Furthermore, we allow for $c$ to be sampled from a distribution that is absolutely continuous w.r.t. the Lebesgue measure in $\mathbb{R}^d$, whereas before, it was fixed. This extension was straightforward, since $c$ is sampled independently of all other randomness and is only involved in the observation step, not the hidden state dynamics. Similarly, we've updated our proofs to allow non-isotropic, spatially-correlated noise.
2. We've updated our understanding of the algorithm implemented by the transformer weights at optimality when using AR(s)-type tokens, with $s\geq 2$, and uncovered that a step similar to those of Preconditioned Conjugate Gradient (PCG) is taken. We detail the why and how in the main text. We note that this observation is coherent with our theoretical result stating that optimal weights implement a GD step in the AR(1) case, since PCG collapses to GD in one dimension. To our knowledge, this is the first algorithmic interpretation for the case of a past-observations window with a length larger than one.
3. We provide new experiments in settings with fully-dense $A$, as opposed to its previously diagonal structure. We empirically show that the structure of optima does not change. This is consistent with our theory (Lemma 4.1 and Theorem 4.1), which only requires the updated and very general Assumption 3.2.
4. Finally, our response from the 23rd of November addresses all the other major concerns raised for this paper: the setting's relevance, the treatment of related literature, and the hyperparameter choices in the experiments.
\end{enumerate}

We're grateful to the reviewers for their time and constructive feedback, which helped improve the strength of this paper's results. We have addressed all the major issues raised by reviewers, and hope that, in this light, the paper's scores can be reconsidered.

---

### Meta-Review · Area_Chair_tpgh · 2026-01-07

**Summary:**

This paper studies in-context learning in the setting of non-i.i.d. sequential data. The theoretical analysis focuses on an idealized regime and relies on relatively strong assumptions; however, the authors complement the theory with empirical results that support the main findings. Despite the simplified setting, the results are insightful and represent a meaningful step toward better understanding in-context learning for sequential data. I see no fundamental flaws in the proposed ideas.

Overall, reviewer sentiment is positive. The main concerns relate to the restrictiveness of the assumptions, the relatively narrow scope of the theory, and the presentation. The authors used the rebuttal phase effectively by clarifying the claims, revising the manuscript, and addressing several reviewer questions. While the theoretical setting remains somewhat narrow, I view this work as a starting point that can motivate future research. In particular, the paper provides relevant insights into the expressive power of linear attention layers for in-context learning, and the authors have already taken initial steps toward relaxing some assumptions.

Given the positive reviews, the quality of the rebuttal, and the relevance of the topic, I recommend accept.

**Reviewer Concerns:**

The main concerns relate to the restrictiveness of the assumptions, the relatively narrow scope of the theory, and the presentation. In response, the authors revised the manuscript and added additional discussion to improve clarity and accessibility. While the theoretical scope remains somewhat narrow, I do not view this as a major issue.

**Reviewer Scores:**

I feel the rebuttal addressed most concerns and could have potentially shifted the scores slightly up.

---

### Decision · Program_Chairs · 2026-01-26

Accept (Poster)